# Distributed Methods with Compressed Communication for Solving Variational Inequalities, with Theoretical Guarantees

**Aleksandr Beznosikov**
Innopolis University,* MIPT,† HSE University and Yandex, Russia
anbeznosikov@gmail.com

**Peter Richtárik**
KAUST,‡ Saudi Arabia
peter.richtarik@kaust.edu.sa

**Michael Diskin**
HSE University and Yandex, Russia
michael.s.diskin@gmail.com

**Max Ryabinin**
Yandex and HSE University, Russia
mryabinin0@gmail.com

**Alexander Gasnikov**
MIPT, HSE University and IITP RAS,§ Russia
gasnikov@yandex.ru

## Abstract

Variational inequalities in general and saddle point problems in particular are increasingly relevant in machine learning applications, including adversarial learning, GANs, transport and robust optimization. With increasing data and problem sizes necessary to train high performing models across various applications, we need to rely on parallel and distributed computing. However, in distributed training, communication among the compute nodes is a key bottleneck during training, and this problem is exacerbated for high dimensional and over-parameterized models. Due to these considerations, it is important to equip existing methods with strategies that would allow to reduce the volume of transmitted information during training while obtaining a model of comparable quality. In this paper, we present the first theoretically grounded distributed methods for solving variational inequalities and saddle point problems using compressed communication: MASHA1 and MASHA2. Our theory and methods allow for the use of both unbiased (such as Rand$k$; MASHA1) and contractive (such as Top$k$; MASHA2) compressors. New algorithms support bidirectional compressions, and also can be modified for stochastic setting with batches and for federated learning with partial participation of clients. We empirically validated our conclusions using two experimental setups: a standard bilinear min-max problem, and large-scale distributed adversarial training of transformers.

*Research Center for Artificial Intelligence, Innopolis University

†Moscow Institute of Physics and Technology

‡King Abdullah University of Science and Technology

§Institute for Information Transmission Problems RAS

36th Conference on Neural Information Processing Systems (NeurIPS 2022).

# 1 Introduction

## 1.1 The expressive power of variational inequalities

Due to their abstract mathematical nature and the associated flexibility they offer in modeling various practical problems of interests, *variational inequalities (VI)* have been an active area of research in applied mathematics for more than half a century [65, 31, 22]. It is well known that VIs can be used to formulate and study optimization problems, *saddle point problems (SPPs)*, games and fixed point problems, for example, in an elegant unifying mathematical framework [9].

Recently, a series of works by various authors [15, 26, 58, 13, 49] built a bridge between VIs/SPPs and GANs [28]. This allows to successfully transfer established insights and well-known techniques from the vast literature on VIs/SPPs, such as averaging and extrapolation, to the study of GANs. Besides their usefulness in studying GANs and alternative adversarial learning models [57], VIs/SPPs have recently attracted considerable attention of the machine learning community due to their ability to model other situations where the minimization of a single loss function does not suffice, such as auction theory [80], supervised learning with non-separable loss [39] or non-separable regularizer [7] and reinforcement learning [69, 66, 38].

In summary, VIs have recently become a potent tool enabling new advances in practical machine learning situations reaching beyond supervised learning where optimization problems and techniques, which can be seen as special instances of VIs and methods for solving them, reign supreme.

## 1.2 Training of supervised models via distributed optimization

On the other hand, for classical and much better understood *supervised machine learning*/minimization problems, researchers and practitioners face other challenges, which, until recently, have been outside of VI's research. Indeed, the training of modern supervised machine learning models in general, and deep neural networks in particular, is still extremely challenging. Due to their desire to improve the generalization of deployed models, machine learning engineers need to rely on training datasets of ever increasing sizes and on elaborate over-parametrized models [5]. Supporting workloads of such unprecedented magnitudes would be impossible without combining the latest advances in hardware acceleration, distributed systems and *distributed algorithm design* [83].

When training such modern supervised models in a distributed fashion, *communication cost* is often the bottleneck of the training system, and for this reason, a lot of effort was recently targeted at the design of communication efficient distributed optimization methods [45, 76, 25, 29]. A particularly successful technique for improving the communication efficiency of distributed first order optimization methods is *communication compression*. The idea behind this technique is rooted in the observation that in practical implementations it is often advantageous to communicate messages compressed via (often randomized) *lossy compression techniques* instead of communicating the full messages [75, 2]. If the number of parallel workers is large enough, the noise introduced by compression is reduced, and training with compressed communication will often lead to comparable test error while reducing the amount of communicated bits, which results in faster training, both in theory and practice [59, 29].

## 1.3 Two classes of compression operators

The paper focuses on compression methods for distributed VIs and SPPs. Let us give the main definitions. We say that a (possibly) stochastic mapping $Q : \mathbb{R}^d \to \mathbb{R}^d$ is an *unbiased compression operator* if there exists a constant $q \geq 1$ such that

$$\mathbb{E}Q(z) = z, \quad \mathbb{E}\|Q(z)\|^2 \leq q\|z\|^2, \quad \forall z \in \mathbb{R}^d. \tag{1}$$

Further, we say that a stochastic mapping $C : \mathbb{R}^d \to \mathbb{R}^d$ is a *contractive compression operator* if there exists a constant $\delta \geq 1$ such that

$$\mathbb{E}\|C(z) - z\|^2 \leq (1 - 1/\delta)\|z\|^2, \quad \forall z \in \mathbb{R}^d. \tag{2}$$

If $b$ is the number of bits needed to represent a single float (e.g., $b = 32$ or $b = 64$), then the number of bits needed to represent a generic vector $z \in \mathbb{R}^d$ is $\|z\|_{\text{bits}} := bd$. To describe how much a

compression operator reduces its input vector on average, we define the notion of expected density, denoted via $\beta^{-1} := \frac{1}{bd}\mathbb{E}\|Q(z)\|_{\text{bits}}$, where $\|Q(z)\|_{\text{bits}}$ is the number of bits needed to represent the quantized vector $Q(z)$. Note that $\beta \geq 1$. For the Rand$k$ operator [3, 10] we have $q = \beta = d/k$.

## 1.4 Towards communication-efficient distributed methods for VIs and SPPs

Classical VI/SPP algorithms such as the *Extra Gradient method* originally proposed by [46] and later studied by many authors [62, 41], including in a distributed environment [77, 52, 61, 73]. Among them, a number of works stand out trying to solve the communication bottleneck challenge using various approaches such as local steps, data-similarity etc.[88, 34, 16, 11, 12]. But despite the fact that the use of compression is one of the most popular communication-efficient approaches for distributed minimization problems, *no work has yet paid attention to the compression technique neither for distributed SPPs nor for VIs*, with the exception of the work [88], which relies on rounding to the nearest integer multiple of a certain quantity. This compression mechanism does not offer theoretical benefits and does not even lead to convergence to the solution since the errors introduced through rounding persist and prevent the method from solving the problem.

## 2 Summary of Contributions

In this paper, we investigate whether it is possible to design communication-efficient algorithms for solving distributed VI/SPP by borrowing generic communication compression techniques (1) and (2) from the optimization literature [75, 2, 59, 29, 72] and embedding them into established, efficient methods for solving VIs/SPPs [46, 62, 41, 1]. Whether or not this is possible is an open problem. In summary,

*we design the first algorithms with compression for solving general distributed VI/SPP (see Section 3, Equation 3) in the deterministic (see (4)), stochastic (see (44)) and federated (see (54)) regimes, supporting both unbiased (MASHA1 = Algorithms 1, 5, 7) and contractive (MASHA2 = Algorithms 2, 6, 8) compressors. Convergence of all our methods are analyzed in strongly-monotone (strongly convex - strongly concave), monotone (convex - concave) non-monotone/minty (non-convex-non-concave) cases.*

## 2.1 Two types of compressors

We develop two approaches for distributed VIs/SPPs depending on whether we use unbiased (1) or contractive (2) compressors, since each type of compressor demands a different algorithmic design and a different analysis. In particular, contractive compressors are notoriously hard to analyze even for optimization problems [44, 72]. Our method based on unbiased compressors is called MASHA1 (Algorithm 1), and our method based on contraction compressors is called MASHA2 (Algorithm 2).

## 2.2 Theoretical complexity results

We establish a number of theoretical complexity results for our methods, which we summarize in Table 1 (Appendix A). We consider the strongly monotone (strongly convex - strongly concave), monotone (convex - concave) regimes as well as the more general non-monotone/minty (non-convex-non-concave) regime. In the strongly monotone case we obtain linear convergence results ($O(\log 1/\epsilon)$) in terms of the distance to solution, in the monotone we obtain fast sublinear convergence results ($O(1/\epsilon)$) in terms of the *gap* function, and in the non-monotone case we have sublinear convergence results ($O(1/\epsilon^2)$) in terms of the Euclidean norm of the operator. To get an estimate for the number of information transmitted, one need to multiply the estimates from Table 1 by $1/\beta$. Then we get that from the point of view of the transmitted information (and also time for communications), MASHA1 is better by a factor $\sqrt{1/\beta + 1/M}$ ($M$ – number of workers) in comparison with the classical Extra Gradient. It means that we get an acceleration of $\min\{\sqrt{\beta}; \sqrt{M}\}$ times. For example, ADIANA from [48](the theoretical SOTA method with unbiased compressions for strongly convex minimization) has the same accelaration. The same situation is with MASHA2. The method has the same compression dependent multiplier as ECLK from [71] (the theoretical SOTA with contractive compression for minimization). Based on these facts, we hypothesize that MASHA1 and MASHA2 have unimprovable estimates (see Appendix B).

## 2.3 Stochastic case and variance reduction

MASHA1 and MASHA2 are designed to handle the *deterministic* setting. But often, in practice, the computation of the full operators/gradients is expensive, then we need to deal with *stochastic* realizations. In particular, a popular case is when each operator/gradient has a finite-sum structure on its own, e.g. , finite-sum of batches. For this issue, we consider two modifications: VR-MASHA1 (Algorithm 5) and VR-MASHA2 (Algorithm 6). Both are enhanced with bespoke *variance-reduction* techniques for better theoretical and practical performance. These results can be interesting in the non-distributed case. As far as we know, we are the first who consider variance reduction for non-monotone VIs. We found only one paper on non-convex-concave saddle point problems [87] under the PL condition. See Appendix F for details.

## 2.4 Federated learning and partial participation

*Federated learning* [45, 42] is an important and popular branch of distributed methods. Therefore, a good bonus for the algorithm is that it can be easily adapted for it. In a federated setup where the computing devices are mobile phones, tablets, personal computers etc, the importance of the communication bottleneck is even higher. In such circumstances, devices can have weak and slow connections, or they can even disconnect for a while. At such moments, it is not necessary to interrupt the learning process, and only available devices can be used. Therefore, we introduce two modifications: PP-MASHA1 (Algorithm 7) and PP-MASHA2 (Algorithm 8), that support the mode of *partial participation* of devices in the learning process. For minimization problems, a combination of quantization and partial participation occurs in [33, 68, 29]. The results are contained in Appendix G.

## 2.5 Bidirectional compression

Most methods, especially with contractive compressors, only use compression when transferring information from devices to the server. Meanwhile, quite often in practical situations, the transfer of information from the server to the device is also expensive [32, 81, 68]. In such situations it also makes sense to compress the information when sending it from the server to the agents. We can highlight some works on bidirectional unbiased [68] and contractive compressors [90, 81, 55, 23] for distributed minimization problems. But most of these methods have their small shortcomings in theoretical analysis such as deterministic setting only, homogeneity of local functions, etc. All our methods MASHA1, MASHA2 and their modifications support bidirectional compression. See Appendix D and E for details.

## 2.6 Experiments

Toy experiments on bilinear problems show that methods with compression for minimization problems may not work (diverge) for SPPs. Also we verify that MASHA1 and MASHA2 are much better than the classical Extra Gradient with added unbiased compression. Experiments on adversarial training of large-scale transformer (ALBERT) show the practical importance of compression in distributed methods for large SPPs.

# 3 Problem Formulation and Assumptions

## 3.1 Problem formulation

We study distributed variational inequality (VI) problem

$$\text{Find} \ \ z^* \in \mathbb{R}^d \ \ \text{such that} \ \ \langle F(z^*), z - z^* \rangle \geq 0, \ \ \forall z \in \mathbb{R}^d, \tag{3}$$

where $F : \mathbb{R}^d \rightarrow \mathbb{R}^d$ is an operator with certain favorable properties (e.g., Lipschitzness and monotonicity). We assume that the training data describing $F$ is *distributed* across $M$ workers/nodes/clients

$$F(z) := \frac{1}{M} \sum_{m=1}^{M} F_m(z), \tag{4}$$

where $F_m : \mathbb{R}^d \to \mathbb{R}^d$ for all $m \in \{1, 2, \ldots, M\}$. Next, we give main examples of VIs to show the breadth of this formalism.

**Example 3.1 (Minimization)** *Consider the minimization problem:*

$$\min_{z \in \mathbb{R}^d} f(z). \tag{5}$$

*Suppose that $F(z) := \nabla f(z)$. Then, if $f$ is convex, it can be proved that $z^* \in \mathbb{R}^d$ is a solution for (3) if and only if $z^* \in \mathbb{R}^d$ is a solution for (5). And if the function $f$ is non-convex, then $z^* \in \mathbb{R}^d$ is a solution for (3) if and only if $\nabla f(z^*) = 0$, i.e. $z^*$ is a stationary point.*

**Example 3.2 (Saddle point problem)** *Consider the saddle point problem:*

$$\min_{x \in \mathbb{R}^{d_x}} \max_{y \in \mathbb{R}^{d_y}} g(x, y). \tag{6}$$

*Suppose that $F(z) := F(x, y) = [\nabla_x g(x, y), -\nabla_y g(x, y)]$ and $\mathcal{Z} = \mathbb{R}^{d_x} \times \mathbb{R}^{d_y}$. Then, if $g$ is convex-concave, it can be proved that $z^* \in \mathcal{Z}$ is a solution for (3) if and only if $z^* \in \mathcal{Z}$ is a solution for (6). And if the function $g$ is non-convex-non-concave, then $z^* \in \mathcal{Z}$ is a solution for (3) if and only if $\nabla_x g(x^*, y^*) = 0$ and $\nabla_y g(x^*, y^*) = 0$, i.e. $z^*$ is a stationary point.*

If minimization problems are widely researched separately from variational inequalities. The study of saddle point problems often is associated with variational inequalities, therefore saddle point problems are strongly related to variational inequalities.

**Example 3.3 (Fixed point problem)** *Consider the fixed point problem:*

$$Find \ z^* \in \mathbb{R}^d \ such \ that \ T(z^*) = z^*, \tag{7}$$

*where $T : \mathbb{R}^d \to \mathbb{R}^d$ is an operator. With $F(z) = z - T(z)$, it can be proved that $z^* \in \mathbb{R}^d$ is a solution for (3) if and only if $F(z^*) = 0$, i.e. $z^* \in \mathbb{R}^d$ is a solution for (7).*

## 3.2 Assumptions

Next, we list two key assumptions - both are standard in the literature on VIs.

**Assumption 3.4 (Lipschitzness)** *The operator $F$ is $L$-Lipschitz continuous, i.e. for all $z_1, z_2 \in \mathbb{R}^d$ we have $\|F(z_1) - F(z_2)\| \leq L\|z_1 - z_2\|$.*

*Each operator $F_m$ is $L_m$-Lipschitz continuous, i.e. for all $z_1, z_2 \in \mathbb{R}^d$ it holds $\|F_m(z_1) - F_m(z_2)\| \leq L_m\|z_1 - z_2\|$. Let us define new constant $\tilde{L}$ as follows $\tilde{L}^2 = \frac{1}{M} \sum_{m=1}^{M} L_m^2$.*

For saddle point problems, these properties are equivalent to smoothness.

**Assumption 3.5 (Monotonicity)** *We need three cases of monotonicity*

*(SM) Strong monotonicity. The operator $F$ is $\mu$-strongly monotone, i.e. for all $z_1, z_2 \in \mathbb{R}^d$ we have $\langle F(z_1) - F(z_2), z_1 - z_2 \rangle \geq \mu\|z_1 - z_2\|^2$.*

*(M) Monotonicity. The operator $F$ is monotone, i.e. for all $z_1, z_2 \in \mathbb{R}^d$ we have $\langle F(z_1) - F(z_2), z_1 - z_2 \rangle \geq 0$.*

*(NM) Non-monotonicity. The operator $F$ is non-monotone (minty), if and only if there exists $z^* \in \mathbb{R}^d$ such that for all $z \in \mathbb{R}^d$ we have $\langle F(z), z - z^* \rangle \geq 0$.*

The last assumption is called the minty or variational stability condition. It is not a general non-monotonicity, but is already associated in the community with non-monotonicity [14, 37, 58, 53, 43, 36, 19], particularly with the setup, which is somewhat appropriate for GANS [51, 52, 21, 8].

## 4 MASHA

In this Section we present new algorithms and their convergence. Section 4.1 is devoted to the algorithm (MASHA1) with unbiased compression. Section 4.2 – to algorithm (MASHA2) with contractive

compression. Appendix gives modifications for the stochastic case – Section F, and for the federated learning – Section G. Appendix B is devoted to the hypothesis about optimality of MASHA1 and MASHA2.

## 4.1 MASHA1: Handling Unbiased Compressors

Before presenting our algorithm, let us discuss which approaches can be used to construct it. As discussed in Sections 1 and 2, compression methods play an important role in distributed minimization problems. All these methods are modifications of the classical GD. For instance, the authors of [2] compress stochastic gradients. Therefore, it is a natural idea to use GD-type methods for VIs as well. But it is a well-known fact that GD-type methods can give bad convergence estimates (see Section B.1 from [67]) or do not converge at all (see Section 7.2 and 8.2 from [27]) even on the simplest SPPs and VIs. From a practical point of view, this approach can also fail (see QSGD and EF in Section 5.1). In the non-distributed case, this problem has long been solved and the Extra Gradient method [46, 62, 41] is used instead of GD:

$$z^{k+1/2} = z^k - \gamma F(z^k), \quad z^{k+1} = z^k - \gamma F(z^{k+1/2}). \tag{8}$$

This method is optimal for both VIs and SPPs and has an estimate of convergence $\tilde{\mathcal{O}}(L/\mu)$ in the strongly monotone case. Therefore, the second idea for the compressed method is to add compression operators to the method (8), e.g. use $Q_k(F(z^k))$ and $Q_{k+1/2}(F(z^{k+1/2}))$ instead of $F(z^k)$ and $F(z^{k+1/2})$. In Section H we analyse this method, but it gives an estimate $\tilde{\mathcal{O}}\left(1 + q/M \right) \cdot L^2/\mu^2$, which is considerably worse in terms of $L/\mu$ than the original Extra Gradient method. The key problem is that in the analysis one has to deal with $\|Q_k(F(z^{k+1/2})) - Q_{k+1/2}(F(z^k))\|^2$. Without compression operators, such difference is easily evaluated using Assumption 3.4. But when the compression operators are different (in fact the same, but have different randomness) we cannot make a good estimate for this term. The idea arises to use the same randomness in both steps of the method (8), namely to substitute $Q_k(F(z^k))$ and $Q_k(F(z^{k+1/2}))$. But then $z^{k+1/2}$ depends on the randomness $Q_k$, and hence $Q_k(F(z^{k+1/2}))$ is biased, which further complicates the analysis. For exactly the same reasons, the various optimistic/single call modifications [70, 26, 35, 60] of the Extra Gradient method did not work for us either. We have also test the method (8) with compressions in practice (see CEG in Section 5.1), and it turns out to be worse than the method we will present below. In the end, the use of variance reduction and negative momentum techniques [1] is key in creating our algorithm. These tricks are not in themselves relevant to distributed problems, but, in our case, they help in creating MASHA1 and MASHA2.

---

**Algorithm 1** MASHA1

---

**Parameters:** Stepsize $\gamma > 0$, parameter $\tau \in (0; 1)$, number of iterations $K$.
**Initialization:** Choose $z^0 = w^0 \in \mathcal{Z}$.
Devices send $F_m(w^0)$ to server and get $F(w^0)$
**for** $k = 0, 1, 2, \ldots, K - 1$ **do**
    **for** each device $m$ in parallel **do**
        $z^{k+1/2} = \tau z^k + (1 - \tau)w^k - \gamma F(w^k)$
        Sends $g_m^k = Q_m^{\text{dev}}(F_m(z^{k+1/2}) - F_m(w^k))$ to server
    **end for**
    **for** server **do**
        Sends to devices $g^k = Q^{\text{serv}}\left[\frac{1}{M}\sum_{m=1}^M g_m^k\right]$
        Sends to devices one bit $b_k$ : 1 with probability $1 - \tau$, 0 with with probability $\tau$
    **end for**
    **for** each device $m$ in parallel **do**
        $z^{k+1} = z^{k+1/2} - \gamma g^k$
        If $b_k = 1$ then $w^{k+1} = z^k$, sends $F_m(w^{k+1})$ to server and gets $F(w^{k+1})$
        else $w^{k+1} = w^k$
    **end for**
**end for**

---

At the beginning of each MASHA1 iteration, all devices know the value of $F(w^k)$, hence they can calculate the value of $z^{k+1/2}$ locally without communications. Further, each device sends the com-

pressed version of the difference $F_m(z^{k+1/2}) - F_m(w^k)$ to the server. The compression on these transfers is done by their local $\{Q_m^{\text{dev}}\}$ operators. The server aggregates the information from devices, averages it, compresses by $Q^{\text{serv}}$ operator and makes a broadcast to all devices. As a result, an unbiased estimate of $F(z^{k+1/2}) - F(w^k)$ appears at each node. Also, the nodes receive one bit of information $b_k$. This bit is generated randomly on the server and is equal to 1 with probability $1 - \tau$ (where $1 - \tau$ is small). Note that $b_k$ can be generated locally, it is enough to use the same random generator and set the same seed on all devices. Next, the devices locally make a final update on $z^{k+1}$. The final step is an update of $w^{k+1}$: if $b_k = 1$, then $w^{k+1} = z^k$ or otherwise $w^{k+1} = w^k$. In the case when $w^{k+1} = z^k$, we need to exchange the uncompressed values of $F_m(w^{k+1})$ in order to ensure that at the beginning of the next iteration the value of $F(w^{k+1})$ is known to all agents. We use a possibly difference compressor on each device and also on the server. To distinguish between them, we denote the following notation: $Q_m^{\text{dev}}$, $q_m^{\text{dev}}$, $\beta_m^{\text{dev}}$ and $Q^{\text{serv}}$, $q^{\text{serv}}$, $\beta^{\text{serv}}$.

**Theorem 4.1** *Let Assumption 3.4 and one case of Assumption 3.5 are satisfied. Then for some step $\gamma$ the following estimates on* MASHA1 *number of iterations to achieve $\varepsilon$-solution holds*

- *in strongly monotone case (in terms of $\mathbb{E}[\|z^K - z^*\|^2] \sim \varepsilon$): $\mathcal{O}([\frac{1}{1-\tau} + \frac{C_q}{\mu\sqrt{1-\tau}}]\log\frac{1}{\varepsilon})$;*

- *in monotone case (in terms of $\mathbb{E}\max_{z\in\mathcal{C}}[\langle F(u), (\frac{1}{K}\sum_{k=0}^{K-1}z^{k+1/2}) - u\rangle] \sim \varepsilon$): $\mathcal{O}(\frac{C_q\|z^0 - z^*\|^2}{\varepsilon\sqrt{1-\tau}})$;*

- *in non-monotone case (in terms of $\mathbb{E}[\frac{1}{K}\sum_{k=0}^{K-1}\|F(w^k)\|^2] \sim \varepsilon^2$): $\mathcal{O}(\frac{C_q^2\|z^0 - z^*\|^2}{\varepsilon^2(1-\tau)})$;*

*where $C_q^2 = \frac{q^{\text{serv}}}{M^2}\sum_{m=1}^M(q_m^{\text{dev}}L_m^2 + (M-1)\tilde{L}^2)$.*

A full description of the algorithm, as well as a full statement of the theorem with proof, can be found in Appendix D.

The bounds in Theorem 4.1 are related to $\tau$. Let us find an optimal way to choose it. Note that (in average) once per $1/(1-\tau)$ iterations (when $b_k = 1$), we send uncompressed information. Based on this observation, we can find the best option for $\tau$. Let us analyze the case of compressions only on the devices' side ($q^{\text{serv}} = 1$). For simplicity, we put $Q_m^{\text{dev}} = Q$ with $q_m^{\text{dev}} = q$ and $\beta_m^{\text{dev}} = \beta$, also $L_m = \tilde{L} = L$. Since compression is done only on devices, we assume that the server's broadcast is cheap and we only care about devices. Then at each iteration the device sends $\mathcal{O}(1/\beta + 1 - \tau)$ bits – each time information compressed by $\beta$ times and with probability $1 - \tau$ we send the full package. From where we immediately get the optimal choice for $\tau$:

**Corollary 4.2** *Let Assumption 3.4 and one case of Assumption 3.5 are satisfied. Then for some step $\gamma$ and $1 - \tau = 1/\beta$ the following estimates on* MASHA1 *number of iterations to achieve $\varepsilon$-solution holds*

- *in strongly monotone case: $\mathcal{O}([\beta + \sqrt{\frac{q\beta}{M} + \beta} \cdot \frac{L}{\mu}]\log\frac{1}{\varepsilon})$;*
- *in monotone case: $\mathcal{O}(\sqrt{\frac{q\beta}{M} + \beta} \cdot \frac{L\|z^0 - z^*\|^2}{\varepsilon})$;*
- *in non-monotone case: $\mathcal{O}([\frac{q\beta}{M} + \beta]\frac{L^2\|z^0 - z^*\|^2}{\varepsilon^2})$.*

We can see that MASHA1 can outperform the uncompressed Extra Gradient method. Let us compare them in the strongly monotone case. The communication complexity of the Extra Gradient method is $\tilde{\mathcal{O}}(L/\mu)$. MASHA1 has communication complexity $\tilde{\mathcal{O}}(\sqrt{q/\beta M + 1/\beta} \cdot L/\mu)$. For practical compressors [10], $\beta \geq q$. Then, one can note that the communication complexity of MASHA1 differs from the complexity of the uncompressed method by an additional factor $(\sqrt{1/M + 1/\beta})$. It is easy to see that even for a small number of devices $M$ and expected density $\beta$, this factor is less than 1, hence MASHA1 outperforms the uncompressed method. We think that this factor $(\sqrt{1/M + 1/\beta})$ is theoretically unimprovable and optimal – see Section B for details.

One can also consider the case of bidirectional compression ($q^{\text{serv}} \neq 1$). Table 1 (line 3) shows the result for $q^{\text{serv}} = q_m^{\text{dev}} = q$, $\beta^{\text{serv}} = \beta_m^{\text{dev}} = \beta$ and $1 - \tau = 1/\beta$.

## 4.2 MASHA2: Handling Contractive Compressors

The use of contractive compressions is a more complex issue. In particular, it is known that if one simply put a contractive cospressor instead of an unbiased one, the method may diverge even for

quadratic problems [10]. To fix this, an error compensation technique [78, 44, 79] is used. The point of this approach is to keep untransmitted information and add it to a new package at the next iteration. This is the main difference between MASHA2 and MASHA1. MASHA2 introduces additional sequences $e^k$, $e_m^k$ for the server's and devices' error. To define contractive operators on devices and on the server, we introduce the following notation: $C_m^{\text{dev}}, \delta^{\text{dev}}, \beta^{\text{dev}}$ and $C_m^{\text{serv}}, \delta^{\text{serv}}, \beta^{\text{serv}}$.

---

**Algorithm 2** MASHA2

---

**Parameters:** Stepsize $\gamma > 0$, parameter $\tau$, number of iterations $K$.
**Initialization:** Choose $z^0 = w^0 \in \mathcal{Z}$, $e_m^0 = 0$, $e^0 = 0$.
Devices send $F_m(w^0)$ to server and get $F(w^0)$
**for** $k = 0, 1, 2, \ldots, K - 1$ **do**
    **for** each device $m$ in parallel **do**
        $z^{k+1/2} = \tau z^k + (1 - \tau) w^k - \gamma F(w^k)$
        Sends $g_m^k = C_m^{\text{dev}}(\gamma F_m(z^{k+1/2}) - \gamma F_m(w^k) + e_m^k)$ to server
        $e_m^{k+1} = e_m^k + \gamma F_m(z^{k+1/2}) - \gamma F_m(w^k) - g_m^k$
    **end for**
    **for** server **do**
        Sends to devices $g^k = C^{\text{serv}} \left[ \frac{1}{M} \sum_{m=1}^{M} g_m^k + e^k \right]$
        $e^{k+1} = e^k + \frac{1}{M} \sum_{m=1}^{M} g_m^k - g^k$
        Sends to devices one bit $b_k$ : 1 with probability $1 - \tau$, 0 with with probability $\tau$
    **end for**
    **for** each device $m$ in parallel **do**
        $z^{k+1} = z^{k+1/2} - \gamma g^k$
        If $b_k = 1$ then $w^{k+1} = z^k$, sends $F_m(w^{k+1})$ to server and gets $F(w^{k+1})$
        else $w^{k+1} = w^k$
    **end for**
**end for**

---

In the case of MASHA1, the key theoretical issue was the choice of a basic method (we discussed this at the beginning of Section 4.1). MASHA2 raises another problem for theoretical analysis, how to combine MASHA1 and the error feedback technique. The analysis of methods with error compensation for the minimization problem $\min_x f(x)$ is entirely tied to the existence of the function $f$ [79, 71, 72]. In particular, the differences $(f(\cdot) - f(x^*))$ appear in the whole analysis and is key in the technical lemmas. As a result $(f(\cdot) - f(x^*))$ is used as a convergence criterion even in the strongly convex case. But for VIs there is no function $f$, only the operator $F$ (the existence of $g(x, y)$ in SPP setup does not save the situation). This problem is solved in the proof of Theorem 4.3 by using an additional sequence $\|z^{k+1/2} - w^k\|$.

**Theorem 4.3** *Let Assumption 3.4 and one case of Assumption 3.5 are satisfied. Then for some step $\gamma$ the following estimates on* MASHA2 *number of iterations to achieve $\varepsilon$-solution holds*

- *in strongly monotone case (in terms of $\mathbb{E}[\|\hat{z}^K - z^*\|^2] \sim \varepsilon$):* $\mathcal{O}([\frac{1}{1-\tau} + \frac{\delta^{\text{dev}} \delta^{\text{serv}} \tilde{L}}{\mu \sqrt{1-\tau}}] \log \frac{1}{\varepsilon})$;

- *in monotone case ($\mathbb{E} \max_{z \in \mathcal{C}} [\langle F(u), (\frac{1}{K} \sum_{k=0}^{K-1} z^{k+1/2}) - u \rangle] \sim \varepsilon$):* $\mathcal{O}(\frac{\delta^{\text{dev}} \delta^{\text{serv}} \tilde{L} \|z^0 - z^*\|^2}{\varepsilon \sqrt{1-\tau}})$;

- *in non-monotone case (in terms of $\mathbb{E}[\frac{1}{K} \sum_{k=0}^{K-1} \|F(w^k)\|^2] \sim \varepsilon^2$):* $\mathcal{O}(\frac{(\delta^{\text{dev}} \delta^{\text{serv}})^2 \tilde{L}^2 \|z^0 - z^*\|^2}{\varepsilon^2 (1-\tau)})$.

A full listing of the algorithm, as well as a full statement of the theorem with proof, can be found in Appendix E.

The same way as in Section 4.1 we can consider only devices' or bidirectional compression. In particular, in the line 2 of Table 1 we put results for $\delta^{\text{serv}} = 1$, $\delta^{\text{dev}} = \delta$, $L_m = \tilde{L} = L$ and $1 - \tau = \beta$. In the line 4 of Table 1 there are results for $\delta^{\text{serv}} = \delta^{\text{dev}} = \delta$, $L_m = \tilde{L} = L$ and $1 - \tau = \beta$.

## 5 Experiments

### 5.1 Bilinear Saddle Point Problem

We start our experiments with a distributed bilinear problem, i.e. the problem (6) with

$$g_m(x,y) := x^\top A_m y + a_m^\top x + b_m^\top y + \frac{\lambda}{2}\|x\|^2 - \frac{\lambda}{2}\|y\|^2, \tag{9}$$

where $A_m \in \mathbb{R}^{d \times d}$, $a_m, b_m \in \mathbb{R}^d$. This problem is $\lambda$-strongly convex–strongly-concave and, moreover, all functions $g_m$ are $\|A_m\|_2$-smooth. Therefore, such a distributed problem is well suited for the primary comparison of our methods. We take $d = 100$ and generate positive definite matrices $A_m$ and vectors $a_m, b_m$ randomly, $\lambda$ is chosen as $\max_m \|A_m\|_2/10^5$.

The purpose of the experiment is to understand whether the MASHA1 and MASHA2 methods are superior to those in the literature. As a comparison, we take QGD [2] with Random 30%, classical Error Feedback [78] with Top 30% compression, as well as CEG (Section H) – Compressed Extra Gradient, each step of which we use Random 30%. In MASHA1 (Algorithm 1) we also used Random 30%, in MASHA2 (Algorithm 2) – Top 30%. See Figure 1. The stepsizes of all methods are chosen for best convergence.

We see on Figure 1 that methods based on gradient descent (QSGD and EF) converge slowly. This confirms that one needs to use method specifically designed for saddle point problems (for example, the extra-gradient method), and not classical optimization

Figure 1: Comparison MASHA1 (Algorithm 1) and MASHA2 (Algorithm 2) with Error Feedback, QGD and Compressed Extra Gradient (CEG) in iterations and in Mbytes for (9).

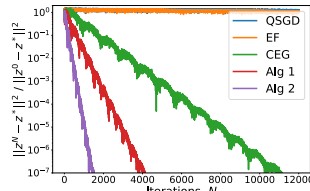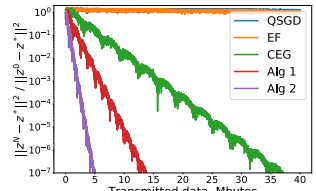

methods. The much slower convergence of CEG shows the efficiency of our approach in which we compress the differences $F_m(z^{k+1/2}) - F(w^k)$. MASHA2 wins MASHA1. This shows that in practice a contractive compressor can perform better than an unbiased one with the same parameters.

### 5.2 Adversarial Training of Transformers

We now evaluate how compression performs for variational inequalities (and for saddle point problems, as a special case) in a more practically motivated scenario. Indeed, saddle point problems (special case of variational inequalities) have sample applications in machine learning, including *adversarial training*. And our goal is to show that compression provides important improvements for such large-scale problems as well. We train a *transformer-based masked language model* [82, 18, 56] using a fleet of 16 low-cost preemptible workers with T4 GPU and low-bandwidth interconnect. For this task, we use the compute-efficient adversarial training regimen proposed for transformers by [91, 54]. Formally, the adversarial formulation of the problem is the min-max problem

$$\min_w \max_{\|\rho_n\| \le e} \frac{1}{N} \sum_{n=1}^N l(f(w, x_n + \rho_n, y_n)^2 + \frac{\lambda}{2}\|w\|^2,$$

where $w$ are the weights of the model, $\{(x_n, y_n)\}_{n=1}^N$ are pairs of the training data, $\rho$ is the so-called adversarial noise which introduces a perturbation in the data, and $\lambda$ are the regularization parameters. To make our setup more realistic, we train ALBERT-large with layer sharing [47], which was recently shown to be much more communication-efficient during training [74, 20]. We train our model on a combination of Bookcorpus and Wikipedia datasets with the same optimizer (LAMB) and parameters as in the original paper [47], use the adversarial training configuration of [91], and follow system design considerations for preemptible instances [74]. In LAMB optimizer we change the original positive momentum to negative momentum, as in MASHA. This means that we do not exactly use MASHA in these experiments, but a combination of MASHA and LAMB. In fact this approach is typical, e.g., in papers [15, 26, 58, 13, 49], the theoretical methods are combined with Adam.

In terms of communication, we consider 4 different setups for gradient compression: the "baseline" strategy with uncompressed gradients, full 8-bit quantization [17, 50], mixed 8-bit quantization, and Power compression [84] with rank $r=8$. For mixed 8-bit quantization and Power we only apply compression to gradient tensors with more than $2^{16}$ elements, sending smaller ones uncompressed. These small tensors represent layer biases and LayerNorm scales [6] that collectively amount to $\leq 1\%$ of the total gradient, but can be more difficult to compress than regular weight tensors. Finally, since Power is a biased compression algorithm, we use error feedback [44, 72] with a modified formulation proposed by [84]. For all experimental setups, we report learning curves in terms of the model training objective, similarly to [24, 74]. To quantify the differences in training loss better, we also evaluate the downstream performance for each model on several popular tasks from [85] after each model was trained on approximately 80 billion tokens. Finally, we measure the communication efficiency of each proposed strategy by measuring the average wall time per communication round when all 16 workers are active.

The learning curves in Figure 2 (upper) follow a predictable pattern, with more extreme compression techniques demonstrating slower per-iteration convergence. One curious exception to that is full 8-bit quantization, which was unable to achieve competitive training loss. The remaining three setups converge to similar loss values below 2. Both the baseline and mixed 8-bit compression show similar values in terms of downstream performance, with Power compression showing

Figure 2: **(upper left)** ALBERT training objective convergence rate with different compression algorithms; **(upper right)** ALBERT training objective convergence rate with different compression algorithms (zoomed); **(lower)** Average wall time per communication round with standard deviation over 5 repetitions and downstream evaluation scores on GLUE benchmark tasks after at 80 billion training tokens ($\approx 10^4$ optimizer steps).

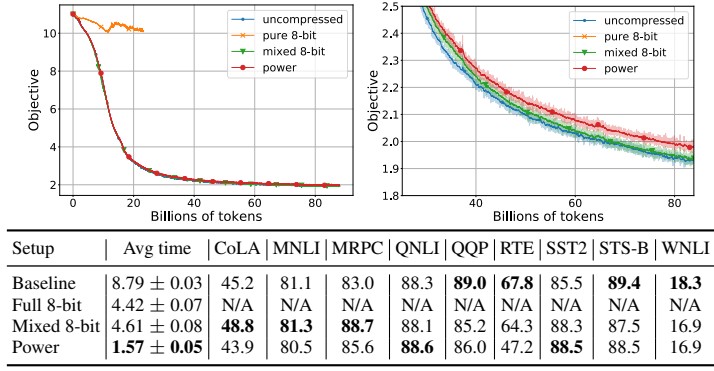

| Setup | Avg time | CoLA | MNLI | MRPC | QNLI | QQP | RTE | SST2 | STS-B | WNLI |
|---|---|---|---|---|---|---|---|---|---|---|
| Baseline | $8.79 \pm 0.03$ | 45.2 | 81.1 | 83.0 | 88.3 | **89.0** | **67.8** | 85.5 | **89.4** | **18.3** |
| Full 8-bit | $4.42 \pm 0.07$ | N/A | N/A | N/A | N/A | N/A | N/A | N/A | N/A | N/A |
| Mixed 8-bit | $4.61 \pm 0.08$ | **48.8** | **81.3** | **88.7** | 88.1 | 85.2 | 64.3 | 88.3 | 87.5 | 16.9 |
| Power | $\mathbf{1.57 \pm 0.05}$ | 43.9 | 80.5 | 85.6 | **88.6** | 86.0 | 47.2 | **88.5** | 88.5 | 16.9 |

mild degradation. But in terms of information transfer time, methods using compression (especially Power) are significantly superior to the method without compression. This makes it possible to use such techniques to increase the training time without sacrificing quality.

# 6 Conclusion

In this paper we present algorithms with unbiased and contractive compressions for solving distributed VIs and SPPs. Our algorithms are presented in deterministic, stochastic and federated versions. All basic algorithms and their modifications support bidirectional compression. Experiments confirm the efficiency of both our algorithms and the use of compression for solving large-scale VIs in general.

In future works it is important to address the issue of the necessity to forward uncompressed information in some iterations. Although full packages are rarely transmitted, this is a slight limitation of our approach. Lower bounds for compression methods are also an interesting area of research. At the moment there are neither such results for VIs and SPPs, nor for minimizations. In Appendix B we only hypothesize the optimality of our methods and back it up with analogies, provable lower estimates could complete the story with compressed methods.

# Acknowledgments

This research of A. Beznosikov has been supported by The Analytical Center for the Government of the Russian Federation (Agreement No. 70-2021-00143 dd. 01.11.2021, IGK 000000D730321P5Q0002).

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
