# APPENDIX

## Contents

# A    Table with summary our results

Table 1: Summary of our **iteration complexity** results for finding an $\varepsilon$-solution for problem (3) in the deterministic (i.e., (4)) with only device compression, deterministic with bidirectional (device-server) compression, stochastic (i.e., (4)+(44)) and federated learning/partial participation (i.e., (4)+(54)) setups. In the strongly-monotone (strongly convex - strongly convex) case, convergence is measured by the distance to the solution. In the monotone(convex-concave) case, convergence is measured in terms of the gap function (11). In non-monotone (non-convex-non-concave) case convergence is measured in terms of the norm of the operator. *Notation:* $\mu$ = constant of strong monotonicity of the operator $F$, $L$ = maximum of local Lipschitz constants $L_m$, $R$ = diameter (in Euclidean norm) of the optimization set, $R_0$ = initial distance to the solution, $q$ = the variance parameter associated with an unbiased compressor (see (1)); $\delta$ = the variance parameter associated with a contractive compressor (see (2)); $\beta, \beta$ = expected density (the number of times the operator compresses information); $M$ = the number of parallel clients/nodes; $r$ = the size of the local dataset (see (44)); $b$ = the number of clients in Partial Participation (FL) setup. We have results with bidirectional compression also in stochastic and federated setups, but to simplify the bounds, we present bidirectional results only in the deterministic setup.

| | | Strongly monotone | Monotone | Non monotone |
|---|---|---|---|---|
| Deter. (Device) (3)+(4) | MASHA1 Alg 1 Cor 4.2 | $\tilde{\mathcal{O}}\left(\beta + \sqrt{\beta + \frac{q\beta}{M}} \cdot \frac{L}{\mu}\right)$ | $\mathcal{O}\left(\sqrt{\beta + \frac{q\beta}{M}} \cdot \frac{LR^2}{\varepsilon}\right)$ | $\mathcal{O}\left(\left(\beta + \frac{q\beta}{M}\right) \cdot \frac{L^2R^2}{\varepsilon^2}\right)$ |
| | MASHA2 Alg 2 Cor E.2 | $\tilde{\mathcal{O}}\left(\beta + \delta\sqrt{\beta} \cdot \frac{L}{\mu}\right)$ | $\mathcal{O}\left(\delta\sqrt{\beta} \cdot \frac{LR^2}{\varepsilon}\right)$ | $\mathcal{O}\left(\delta^2\beta \cdot \frac{L^2R^2}{\varepsilon^2}\right)$ |
| Deter. (Bidirect.) (3)+(4) | MASHA1 Alg 1 Cor D.3 | $\tilde{\mathcal{O}}\left(\beta + \sqrt{q\beta + \frac{q^2\beta}{M}} \cdot \frac{L}{\mu}\right)$ | $\mathcal{O}\left(\sqrt{q\beta + \frac{q^2\beta}{M}} \cdot \frac{LR^2}{\varepsilon}\right)$ | $\mathcal{O}\left(\left(q\beta + \frac{q^2\beta}{M}\right) \cdot \frac{L^2R^2}{\varepsilon^2}\right)$ |
| | MASHA2 Alg 2 Cor E.3 | $\tilde{\mathcal{O}}\left(\beta + \delta^2\sqrt{\beta} \cdot \frac{L}{\mu}\right)$ | $\mathcal{O}\left(\delta^2\sqrt{\beta} \cdot \frac{LR^2}{\varepsilon}\right)$ | $\mathcal{O}\left(\delta^4\beta \cdot \frac{L^2R^2}{\varepsilon^2}\right)$ |
| Stoch. (F-S) (3)+(4)+(44) | VR-MASHA1 Alg 5 Cor F.3 | $\tilde{\mathcal{O}}\left(\beta + r + \max\{\sqrt{\beta}; \sqrt{r}\}\sqrt{1 + \frac{q}{M}} \cdot \frac{L}{\mu}\right)$ | $\mathcal{O}\left(\max\{\sqrt{\beta}; \sqrt{r}\}\sqrt{1 + \frac{q}{M}} \cdot \frac{LR^2}{\varepsilon}\right)$ | $\mathcal{O}\left(\left(\max\{\beta; r\}\left(1 + \frac{q}{M}\right)\right) \cdot \frac{L^2R^2}{\varepsilon^2}\right)$ |
| | VR-MASHA2 Alg 6 Cor F.5 | $\tilde{\mathcal{O}}\left(\beta + r + \max\{\sqrt{\beta}; \sqrt{r}\}\delta \cdot \frac{L}{\mu}\right)$ | $\mathcal{O}\left(\max\{\sqrt{\beta}; \sqrt{r}\}\delta \cdot \frac{LR^2}{\varepsilon}\right)$ | $\mathcal{O}\left(\max\{\beta; r\}\delta^2 \cdot \frac{L^2R^2}{\varepsilon^2}\right)$ |
| FL (PP) (3)+(4)+(54) | PP-MASHA1 Alg 7 Cor G.2 | $\tilde{\mathcal{O}}\left(\frac{\beta M}{b} + \sqrt{\frac{\beta M}{b} + \frac{q\beta M}{b}} \cdot \frac{L}{\mu}\right)$ | $\mathcal{O}\left(\sqrt{\frac{\beta M}{b} + \frac{q\beta M}{b}} \cdot \frac{LR^2}{\varepsilon}\right)$ | $\mathcal{O}\left(\left(\frac{\beta M}{b} + \frac{q\beta M}{b}\right) \cdot \frac{L^2R^2}{\varepsilon^2}\right)$ |
| | PP-MASHA2 Alg 8 Cor G.4 | $\tilde{\mathcal{O}}\left(\frac{\beta M}{b} + \delta\sqrt{\frac{\beta M^3}{b^3}} \cdot \frac{L}{\mu}\right)$ | $\mathcal{O}\left(\delta\sqrt{\frac{\beta M^3}{b^3}} \cdot \frac{LR^2}{\varepsilon}\right)$ | $\mathcal{O}\left(\delta^2\beta \frac{M^3}{b^3} \cdot \frac{L^2R^2}{\varepsilon^2}\right)$ |

# B  Optimality of MASHA1 and MASHA2

In this section, we discuss why the MASHA1 and MASHA2 convergence estimates cannot be improved (it means that the methods are optimal). We emphasise that this is only a hypothesis based on some analogies. As irrefutable proof we could use lower bounds, but there are no such lower bounds even for minimization problems (despite their wide research in the community). The following considerations are also outlined in Table 2.

We consider the strongly convex/strongly monotone case. For the deterministic minimization problem, lower and optimal upper bounds are given in [64]. These bounds are $\tilde{\mathcal{O}}\left(\sqrt{L/\mu}\right)$. Meanwhile, methods with unbiased (ADIANA [48]) and contractive (ECLK [71]) compression, but without compression, are also optimal for the deterministic minimization problem. Iteration complexity of ADIANA with compression is $\tilde{\mathcal{O}}\left(\sqrt{q\beta/M + \beta} \cdot \sqrt{L/\mu}\right)$. For ECLK complexities in iterations is $\tilde{\mathcal{O}}\left(\delta\sqrt{\beta} \cdot \sqrt{L/\mu}\right)$. Note the interesting feature that the compression dependent multiplier can be improved, but only with a loss in the $L/\mu$-multiplier. For example, DIANA [59] (unbiased) has $\tilde{\mathcal{O}}\left((q/M + 1) \cdot L/\mu\right)$ iteration complexity, or EF [79] (contractive) has $\tilde{\mathcal{O}}\left(\delta \cdot L/\mu\right)$ iteration complexity. MASHA1 and MASHA2 without compressions are optimal for Lipschitz continuous strongly monotone VIs [89] and have a deterministic bound $\tilde{\mathcal{O}}\left(L/\mu\right)$. MASHA1 and MASHA2 have the same compression dependency multipliers as ADIANA and ECLK. This suggests that the dependence of MASHA1 and MASHA2 on compression properties cannot be improved for variational inequalities without loss in $L/\mu$. In Section H, we prove the convergence of CEG with unbiased compression, which achieves $\tilde{\mathcal{O}}\left((q/M + 1) \cdot L^2/\mu^2\right)$ iteration complexity.

As another argument, let us give an example of the situation with the VR approach (finite sum problem) for minimization problems and for VIs. For minimization, the lower bounds in the smooth strongly convex case are $\tilde{\mathcal{O}}\left(r + \sqrt{rL/\mu}\right)$ [86]. The optimal method is [4]. SVRG [40] has estimates $\tilde{\mathcal{O}}\left(r + L/\mu\right)$ (better in $r$, worse in $L/\mu$). What about variational inequalities? The lower bounds in the Lipschitz continuous strongly convex case are $\tilde{\mathcal{O}}\left(r + \sqrt{r}L/\mu\right)$ [30]. The optimal methods are [1]. Methods from [67] have estimates $\tilde{\mathcal{O}}\left(r + L^2/\mu^2\right)$. Following this logic, estimates for ADIANA and ECLK are transformed into estimates for MASHA1 and MASHA2.

The same situation with estimates is in the convex/monotone case.

Table 2: Summary of iteration complexity results for minimization problems and variational inequalities in different setups: deterministic, stochastic, distributed with biased and contractive compressions. *Notation:* $\mu$ = constant of strong convexity/monotonicity, $L$ = Lipschitz constant of the gradient/operator, $q$ = the variance parameter associated with an unbiased compressor; $\delta$ = the variance parameter associated with a contractive compressor; $\beta$ = expected density (the number of times the operator compresses information); $M$ = the number of parallel clients/nodes; $r$ = the size of the local dataset.

| | Deterministic | | Stochastic (VR) | | | Unbiased compression | | |
|---|---|---|---|---|---|---|---|---|
| | Lower | Upper | Lower | Upper 1 | Upper 2 | Lower | Upper 1 | Upper 2 |
| Minimization | $\sqrt{\frac{L}{\mu}}$ [64] | $\sqrt{\frac{L}{\mu}}$ [64] | $\sqrt{r} \cdot \sqrt{\frac{L}{\mu}}$ [86] | $\sqrt{r} \cdot \sqrt{\frac{L}{\mu}}$ [4] | $r + \frac{L}{\mu}$ [40] | – | $\sqrt{\beta + \frac{q\beta}{M}} \cdot \sqrt{\frac{L}{\mu}}$ [48] | $\left(1 + \frac{q}{M}\right) \cdot \frac{L}{\mu}$ [59] |
| VI/SPP | $\frac{L}{\mu}$ [89] | $\frac{L}{\mu}$ [26] | $\sqrt{r} \cdot \frac{L}{\mu}$ [30] | $\sqrt{r} \cdot \frac{L}{\mu}$ [1] | $r + \frac{L^2}{\mu^2}$ [67] | – | $\sqrt{\beta + \frac{q\beta}{M}} \cdot \frac{L}{\mu}$ (**Ours**) | $\left(1 + \frac{q}{M}\right) \cdot \frac{L^2}{\mu^2}$ (**Ours**) |

| | | Contractive compression | |
|---|---|---|---|
| | Lower | Upper 1 | Upper 2 |
| | – | $\delta\sqrt{\beta} \cdot \sqrt{\frac{L}{\mu}}$ [71] | $\delta \cdot \frac{L}{\mu}$ [79] |
| | – | $\delta\sqrt{\beta} \cdot \frac{L}{\mu}$ (**Ours**) | – |

## C  Basic Facts

**Upper bound for a squared sum.** For arbitrary integer $n \geq 1$ and arbitrary set of vectors $a_1, \ldots, a_n$ we have

$$\left( \sum_{i=1}^{n} a_i \right)^2 \leq m \sum_{i=1}^{n} a_i^2 \tag{10}$$

## D  MASHA1: Handling Unbiased Compressors

In this section, we provide additional information about Algorithm 1 – MASHA1. We give a full form of MASHA1 – see Algorithm 3.

---
**Algorithm 3** (Algorithm 1) MASHA1

---
1: **Parameters:** Stepsize $\gamma > 0$, parameter $\tau$, number of iterations $K$.
2: **Initialization:** Choose $z^0 = w^0 \in \mathcal{Z}$.
3: Server sends to devices $z^0 = w^0$ and devices compute $F_m(w^0)$ and send to server and get $F(w^0)$
4: **for** $k = 0, 1, 2, \ldots, K-1$ **do**
5:     **for** each device $m$ in parallel **do**
6:         $\bar{z}^k = \tau z^k + (1-\tau)w^k$
7:         $z^{k+1/2} = \bar{z}_k - \gamma F(w^k)$
8:         Compute $F_m(z^{k+1/2})$ & send $Q_m^{\text{dev}}(F_m(z^{k+1/2}) - F_m(w^k))$ to server
9:     **end for**
10:     **for** server **do**
11:         Compute $Q^{\text{serv}}\left[ \frac{1}{M} \sum_{m=1}^{M} Q_m^{\text{dev}}(F_m(z^{k+1/2}) - F_m(w^k)) \right]$ & send to devices
12:         Sends to devices one bit $b_k$: 1 with probability $1-\tau$, 0 with with probability $\tau$
13:     **end for**
14:     **for** each device $m$ in parallel **do**
15:         $z^{k+1} = z^{k+1/2} - \gamma Q^{\text{serv}}\left[ \frac{1}{M} \sum_{m=1}^{M} Q_m^{\text{dev}}(F_m(z^{k+1/2}) - F_m(w^k)) \right]$
16:         **if** $b_k = 1$ **then**
17:             $w^{k+1} = z^k$
18:             Compute $F_m(w^{k+1})$ & send it to server; and get $F(w^{k+1})$ as a response from server
19:         **else**
20:             $w^{k+1} = w^k$
21:         **end if**
22:     **end for**
23: **end for**

---

The following theorem gives the convergence of MASHA1.

**Theorem D.1 (Theorem 4.1)** *Let distributed variational inequality* (3) + (4) *is solved by Algorithm 3 with unbiased compressor operators* (1): *on server with* $q^{serv}$ *parameter, on devices with* $\{q_m^{dev}\}$. *Let Assumption 3.4 and one case of Assumption 3.5 are satisfied. Then the following estimates holds*

• *in strongly-monotone case with* $\gamma \leq \min \left[ \frac{\sqrt{1-\tau}}{2C_q}; \frac{1-\tau}{2\mu} \right]$
*(where* $C_q = \sqrt{\frac{q^{serv}}{M^2} \sum_{m=1}^{M} (q_m^{dev} L_m^2 + (M-1)\tilde{L}^2)}$*):*

$$\mathbb{E}\left( \|z^K - z^*\|^2 + \|w^K - z^*\|^2 \right) \leq \left( 1 - \frac{\mu\gamma}{2} \right)^K \cdot 2\|z^0 - z^*\|^2;$$

• *in monotone case with* $\gamma \leq \frac{\sqrt{1-\tau}}{2C_q + 4\bar{L}}$*:*

$$\mathbb{E}\left[ \max_{z \in \mathcal{C}} \left[ \left\langle F(u), \left( \frac{1}{K} \sum_{k=0}^{K-1} z^{k+1/2} \right) - u \right\rangle \right] \right] \leq \frac{2 \max_{z \in \mathcal{C}} \left[ \|z^0 - z\|^2 \right] + 6\|z^0 - z^*\|^2}{\gamma K};$$

- *in non-monotone case with $\gamma \leq \frac{\sqrt{1-\tau}}{2C_q}$:*

$$\mathbb{E}\left(\frac{1}{K}\sum_{k=0}^{K-1}\|F(w^k)\|^2\right) \leq \frac{16\|z^0 - z^*\|^2}{\gamma^2 K}.$$

For the monotone case, we use the *gap function* as convergence criterion:

$$\text{Gap}(z) := \sup_{u \in \mathcal{C}}\left[\langle F(u), z - u\rangle\right]. \tag{11}$$

Here we do not take the maximum over the entire set $\mathbb{R}^d$ (as in the classical version), but over $\mathcal{C}$ – a compact subset of $\mathbb{R}^d$. Thus, we can also consider unbounded sets $\mathbb{R}^d$. This is permissible, since such a version of the criterion is valid if the solution $z^*$ lies in $\mathcal{C}$; for details see the work of [63].

Let us move on to the choice of $\tau$.

Let us start with the only devices compression, i.e. it is assumed that server-side compression is not required, because broadcasts from the server are cheap. As noted in the main part of the paper, then we consider only sendings from devices to the server. Note that the following expression $\sum_{m=1}^{M}(q_m^{\text{dev}}L_m^2 + (M-1)\tilde{L}^2)$ occurs in $C_q$. It means that we can choose $q_m^{\text{dev}}$ depending on $L_m$. Let us define $L_{\min} = \min_m L_m$ and $q_l = q$ for $l = \arg\min_m L_m$. If one put $q_m = qL_{\min}/L_m$, then we get $C_q = \sqrt{\frac{1}{M}(qL_{\min}^2 + (M-1)\tilde{L}^2)}$.

At each iteration, the device sends to the server $\mathcal{O}\left(\frac{1}{M}\sum_{m=1}^{M}\frac{1}{\beta_m} + (1-\tau)\right)$ bits – each time information compressed by $\beta_m$ (for device $m$) times and with probability $1-\tau$ the full package. Then the optimal choice $\tau$ is $1 - \frac{1}{\beta}$ with $\frac{1}{\beta} = \frac{1}{M}\sum_{m=1}^{M}\frac{1}{\beta_m}$.

**Corollary D.2 (Corollary 4.2)** *Let distributed variational inequality (3) + (4) is solved by Algorithm 3 without compression on server ($q^{serv} = 1$) and with unbiased compressor operators (1) on devices with $\{q_m^{dev}\}$ (as described in the previous paragraphs). Let Assumption 3.4 and one case of Assumption 3.5 are satisfied. Then the following estimates holds*

- *in strongly-monotone case with $\gamma \leq \min\left[\frac{1}{2}\cdot\left(\sqrt{\frac{q\beta L_{\min}^2}{M} + \beta\tilde{L}^2}\right)^{-1}; \frac{1}{2\mu\beta}\right]$:*

$$\mathbb{E}\left(\|z^K - z^*\|^2 + \|w^K - z^*\|^2\right) \leq \left(1 - \frac{\mu\gamma}{2}\right)^K \cdot 2\|z^0 - z^*\|^2;$$

- *in monotone case with $\gamma \leq \frac{1}{6}\cdot\left(\sqrt{\frac{q\beta L_{\min}^2}{M} + \beta\tilde{L}^2}\right)^{-1}$:*

$$\mathbb{E}\left[\max_{z\in\mathcal{C}}\left[\left\langle F(u), \left(\frac{1}{K}\sum_{k=0}^{K-1}z^{k+1/2}\right) - u\right\rangle\right]\right] \leq \frac{2\max_{z\in\mathcal{C}}\left[\|z^0 - z\|^2\right] + 6\|z^0 - z^*\|^2}{\gamma K};$$

- *in non-monotone case with $\gamma \leq \frac{1}{2}\cdot\left(\sqrt{\frac{q\beta L_{\min}^2}{M} + \beta\tilde{L}^2}\right)^{-1}$:*

$$\mathbb{E}\left(\frac{1}{K}\sum_{k=0}^{K-1}\|F(w^k)\|^2\right) \leq \frac{16\|z^0 - z^*\|^2}{\gamma^2 K}.$$

In the line 1 of Table 1 we put complexities to achieve $\varepsilon$-solution. For simplicity, we put $Q_m^{\text{dev}} = Q$ with $q_m^{\text{dev}} = q$ and $\beta_m^{\text{dev}} = \beta$, also $L_m = \tilde{L} = L$.

Next, we add server compression. Now the transfer from the server is important. Here and after, for simplicity, we put $Q^{\text{serv}} = Q_m^{\text{dev}} = Q$ with $q_m^{\text{dev}} = q$ and $\beta_m^{\text{dev}} = \beta$, also $L_m = \tilde{L} = L$. One can also analyze the case with different $q_m$ and $L_m$, as is done in Corollary D.2.

At each iteration, the device is sent to the server and the server to devices $\mathcal{O}\left(\frac{1}{\beta} + 1 - \tau\right)$ bits. Then the optimal choice $\tau$ is still $1 - \frac{1}{\beta}$.

**Corollary D.3** *Let distributed variational inequality (3) + (4) is solved by Algorithm 3 with unbiased compressor operators (1): on server with $q^{serv} = q$ parameter, on devices with $\{q_m^{dev} = q\}$. Let Assumption 3.4 and one case of Assumption 3.5 are satisfied. Then the following estimates holds*

- *in strongly-monotone case with $\gamma \leq \min\left[\frac{1}{2L} \cdot \left(\sqrt{\frac{q^2\beta}{M} + \beta}\right)^{-1}; \frac{1}{2\mu\beta}\right]$:*

$$\mathbb{E}\left(\|z^K - z^*\|^2 + \|w^K - z^*\|^2\right) \leq \left(1 - \frac{\mu\gamma}{2}\right)^K \cdot 2\|z^0 - z^*\|^2;$$

- *in monotone case with $\gamma \leq \frac{1}{6L} \cdot \left(\sqrt{\frac{q^2\beta}{M} + \beta}\right)^{-1}$:*

$$\mathbb{E}\left[\max_{z \in \mathcal{C}}\left[\left\langle F(u), \left(\frac{1}{K}\sum_{k=0}^{K-1} z^{k+1/2}\right) - u\right\rangle\right]\right] \leq \frac{2\max_{z \in \mathcal{C}}\left[\|z^0 - z\|^2\right] + 6\|z^0 - z^*\|^2}{\gamma K};$$

- *in non-monotone case with $\gamma \leq \frac{1}{2L} \cdot \left(\sqrt{\frac{q^2\beta}{M} + \beta}\right)^{-1}$:*

$$\mathbb{E}\left(\frac{1}{K}\sum_{k=0}^{K-1} \|F(w^k)\|^2\right) \leq \frac{16\|z^0 - z^*\|^2}{\gamma^2 K}.$$

In the line 3 of Table 1 we put complexities to achieve $\varepsilon$-solution.

### D.1 Proof of the convergence of MASHA1

**Proof of Theorem D.1:** We start from the following equalities for any $z$:

$$\|z^{k+1} - z\|^2 = \|z^{k+1/2} - z\|^2 + 2\langle z^{k+1} - z^{k+1/2}, z^{k+1/2} - z\rangle + \|z^{k+1} - z^{k+1/2}\|^2,$$

$$\|z^{k+1/2} - z\|^2 = \|z^k - z\|^2 + 2\langle z^{k+1/2} - z^k, z^{k+1/2} - z\rangle - \|z^{k+1/2} - z^k\|^2.$$

Then we sum two inequalities:

$$\|z^{k+1} - z\|^2 = \|z^k - z\|^2 + 2\langle z^{k+1} - z^k, z^{k+1/2} - z\rangle \\ + \|z^{k+1} - z^{k+1/2}\|^2 - \|z^{k+1/2} - z^k\|^2. \qquad (12)$$

Using lines 6, 7, 15, we get

$$\begin{aligned}
\|z^{k+1} - z\|^2 &= \|z^k - z\|^2 \\
&\quad - 2\left\langle \gamma Q^{\text{serv}}\left[\frac{1}{M}\sum_{m=1}^{M} Q_m^{\text{dev}}(F_m(z^{k+1/2}) - F_m(w^k))\right] + \gamma F(w^k), z^{k+1/2} - z\right\rangle \\
&\quad + 2\langle \tau z^k + (1-\tau)w^k - z^k, z^{k+1/2} - z\rangle \\
&\quad + \|z^{k+1} - z^{k+1/2}\|^2 - \|z^{k+1/2} - z^k\|^2 \\
&= \|z^k - z\|^2 \\
&\quad - 2\left\langle \gamma Q^{\text{serv}}\left[\frac{1}{M}\sum_{m=1}^{M} Q_m^{\text{dev}}(F_m(z^{k+1/2}) - F_m(w^k))\right] + \gamma F(w^k), z^{k+1/2} - z\right\rangle \\
&\quad + 2(1-\tau)\langle w^k - z^k, z^{k+1/2} - z\rangle \\
&\quad + \|z^{k+1} - z^{k+1/2}\|^2 - \|z^{k+1/2} - z^k\|^2.
\end{aligned}$$

The equality $2\langle a, b\rangle = \|a + b\|^2 - \|a\|^2 - \|b\|^2$ gives

$$\|z^{k+1} - z\|^2 = \|z^k - z\|^2$$

$$
\begin{aligned}
&\quad - 2\langle \gamma Q^{\text{serv}} \left[ \frac{1}{M} \sum_{m=1}^{M} Q_m^{\text{dev}}(F_m(z^{k+1/2}) - F_m(w^k)) \right] + \gamma F(w^k), z^{k+1/2} - z \rangle \\
&\quad + 2(1-\tau)\langle w^k - z^{k+1/2}, z^{k+1/2} - z \rangle + 2(1-\tau)\langle z^{k+1/2} - z^k, z^{k+1/2} - z \rangle \\
&\quad + \|z^{k+1} - z^{k+1/2}\|^2 - \|z^{k+1/2} - z^k\|^2 \\
&= \|z^k - z\|^2 \\
&\quad - 2\langle \gamma Q^{\text{serv}} \left[ \frac{1}{M} \sum_{m=1}^{M} Q_m^{\text{dev}}(F_m(z^{k+1/2}) - F_m(w^k)) \right] + \gamma F(w^k), z^{k+1/2} - z \rangle \\
&\quad + (1-\tau)\|w^k - z\|^2 - (1-\tau)\|w^k - z^{k+1/2}\|^2 - (1-\tau)\|z^{k+1/2} - z\|^2 \\
&\quad + (1-\tau)\|z^{k+1/2} - z^k\|^2 + (1-\tau)\|z^{k+1/2} - z\|^2 - (1-\tau)\|z^k - z\|^2 \\
&\quad + \|z^{k+1} - z^{k+1/2}\|^2 - \|z^{k+1/2} - z^k\|^2 \\
&= \tau\|z^k - z\|^2 + (1-\tau)\|w^k - z\|^2 \\
&\quad - 2\langle \gamma Q^{\text{serv}} \left[ \frac{1}{M} \sum_{m=1}^{M} Q_m^{\text{dev}}(F_m(z^{k+1/2}) - F_m(w^k)) \right] + \gamma F(w^k), z^{k+1/2} - z \rangle \\
&\quad + \|z^{k+1} - z^{k+1/2}\|^2 - (1-\tau)\|w^k - z^{k+1/2}\|^2 - \tau\|z^{k+1/2} - z^k\|^2. \tag{13}
\end{aligned}
$$

We now consider the three cases of monotonicity separately.

### D.1.1 Strongly-monotone case

Let substitute $z = z^*$, take full mathematical expectation and get

$$
\begin{aligned}
\mathbb{E}\|z^{k+1} - z^*\|^2 &= \tau\mathbb{E}\|z^k - z^*\|^2 + (1-\tau)\mathbb{E}\|w^k - z^*\|^2 \\
&\quad - 2\gamma\mathbb{E}\left[ \langle Q^{\text{serv}} \left[ \frac{1}{M} \sum_{m=1}^{M} Q_m^{\text{dev}}(F_m(z^{k+1/2}) - F_m(w^k)) \right] + F(w^k), z^{k+1/2} - z^* \rangle \right] \\
&\quad + \mathbb{E}\|z^{k+1} - z^{k+1/2}\|^2 - (1-\tau)\mathbb{E}\|w^k - z^{k+1/2}\|^2 - \tau\mathbb{E}\|z^{k+1/2} - z^k\|^2.
\end{aligned}
$$

With unbiasedness (1) we have

$$
\begin{aligned}
\mathbb{E}\|z^{k+1} - z^*\|^2 &= \tau\mathbb{E}\|z^k - z^*\|^2 + (1-\tau)\mathbb{E}\|w^k - z^*\|^2 \\
&\quad - 2\gamma\mathbb{E}\left[ \langle \mathbb{E}_{Q^{\text{serv}}, Q^{\text{dev}}} \left[ Q^{\text{serv}} \left[ \frac{1}{M} \sum_{m=1}^{M} Q_m^{\text{dev}}(F_m(z^{k+1/2}) - F_m(w^k)) \right] + F(w^k) \right], z^{k+1/2} - z^* \rangle \right] \\
&\quad + \mathbb{E}\|z^{k+1} - z^{k+1/2}\|^2 - (1-\tau)\mathbb{E}\|w^k - z^{k+1/2}\|^2 - \tau\mathbb{E}\|z^{k+1/2} - z^k\|^2 \\
&= \tau\mathbb{E}\|z^k - z^*\|^2 + (1-\tau)\mathbb{E}\|w^k - z^*\|^2 \\
&\quad - 2\gamma\mathbb{E}\left[ \langle F(z^{k+1/2}), z^{k+1/2} - z^* \rangle \right] \\
&\quad + \mathbb{E}\|z^{k+1} - z^{k+1/2}\|^2 - (1-\tau)\mathbb{E}\|w^k - z^{k+1/2}\|^2 - \tau\mathbb{E}\|z^{k+1/2} - z^k\|^2. \tag{14}
\end{aligned}
$$

Let us work with $\mathbb{E}\left[ \|z^{k+1} - z^{k+1/2}\|^2 \right]$, with (1) we get

$$
\begin{aligned}
\mathbb{E}\left[ \|z^{k+1} - z^{k+1/2}\|^2 \right] &= \gamma^2 \cdot \mathbb{E}\left[ \left\| Q^{\text{serv}} \left[ \frac{1}{M} \sum_{m=1}^{M} Q_m^{\text{dev}}(F_m(z^{k+1/2}) - F_m(w^k)) \right] \right\|^2 \right] \\
&\leq \gamma^2 \cdot \frac{q^{\text{serv}}}{M^2} \mathbb{E}\left[ \left\| \sum_{m=1}^{M} Q_m^{\text{dev}}(F_m(z^{k+1/2}) - F_m(w^k)) \right\|^2 \right] \\
&= \gamma^2 \cdot \frac{q^{\text{serv}}}{M^2} \sum_{m=1}^{M} \mathbb{E}\left[ \left\| Q_m^{\text{dev}}(F_m(z^{k+1/2}) - F_m(w^k)) \right\|^2 \right]
\end{aligned}
$$

$$+ \gamma^2 \cdot \frac{q^{\text{serv}}}{M^2} \sum_{m \neq l} \mathbb{E}\left[\langle Q_m^{\text{dev}}(F_m(z^{k+1/2}) - F_m(w^k)); Q_l^{\text{dev}}(F_l(z^{k+1/2}) - F_l(w^k))\rangle\right]$$

Next we apply (1) and Assumption 3.4 for the first term and independence and unbiasedness of $Q$ for the second term:

$$\mathbb{E}\left[\|z^{k+1} - z^{k+1/2}\|^2\right] \leq \gamma^2 \cdot \frac{q^{\text{serv}}}{M^2} \sum_{m=1}^{M} q_m^{\text{dev}} L_m^2 \mathbb{E}\left[\left\|z^{k+1/2} - w^k\right\|^2\right]$$
$$+ \gamma^2 \cdot \frac{q^{\text{serv}}}{M^2} \sum_{m \neq l} \mathbb{E}\left[\langle F_m(z^{k+1/2}) - F_m(w^k); F_l(z^{k+1/2}) - F_l(w^k)\rangle\right]$$
$$\leq \gamma^2 \cdot \frac{q^{\text{serv}}}{M^2} \sum_{m=1}^{M} q_m^{\text{dev}} L_m^2 \mathbb{E}\left[\left\|z^{k+1/2} - w^k\right\|^2\right]$$
$$+ \gamma^2 \cdot \frac{q^{\text{serv}}}{2M^2} \sum_{m \neq l} \mathbb{E}\left[\|F_m(z^{k+1/2}) - F_m(w^k)\|^2 + \|F_l(z^{k+1/2}) - F_l(w^k)\|^2\right]$$
$$\leq \gamma^2 \cdot \frac{q^{\text{serv}}}{M^2} \sum_{m=1}^{M} q_m^{\text{dev}} L_m^2 \mathbb{E}\left[\left\|z^{k+1/2} - w^k\right\|^2\right]$$
$$+ \gamma^2 \cdot \frac{q^{\text{serv}}}{2M^2} \sum_{m \neq l} \mathbb{E}\left[L_m^2 \|z^{k+1/2} - w^k\|^2 + L_l^2\|z^{k+1/2} - w^k\|^2\right]$$
$$= \gamma^2 \cdot \frac{q^{\text{serv}}}{M^2} \sum_{m=1}^{M} q_m^{\text{dev}} L_m^2 \mathbb{E}\left[\left\|z^{k+1/2} - w^k\right\|^2\right]$$
$$+ \gamma^2 \cdot \frac{q^{\text{serv}}(M-1)}{M} \tilde{L}^2 \mathbb{E}\left[\|z^{k+1/2} - w^k\|^2\right]$$
$$= \gamma^2 \cdot \frac{q^{\text{serv}}}{M^2} \mathbb{E}\left[\|z^{k+1/2} - w^k\|^2\right] \cdot \sum_{m=1}^{M} q_m^{\text{dev}} L_m^2 + (M-1)\tilde{L}^2 \quad (15)$$

Let us define new constant $C_q = \sqrt{\frac{q^{\text{serv}}}{M^2} \sum_{m=1}^{M}(q_m^{\text{dev}} L_m^2 + (M-1)\tilde{L}^2)}$ and then connect (14) and (15):

$$\mathbb{E}\|z^{k+1} - z^*\|^2 \leq \tau\mathbb{E}\|z^k - z^*\|^2 + (1-\tau)\mathbb{E}\|w^k - z^*\|^2$$
$$- 2\gamma\mathbb{E}\left[\langle F(z^{k+1/2}), z^{k+1/2} - z^*\rangle\right]$$
$$- (1 - \tau - \gamma^2 C_q^2)\mathbb{E}\|w^k - z^{k+1/2}\|^2 - \tau\mathbb{E}\|z^{k+1/2} - z^k\|^2. \quad (16)$$

Then we use choice of $w^{k+1}$ (lines 12, 17, 20) and get

$$\mathbb{E}\|w^{k+1} - z^*\|^2 = \mathbb{E}\left[\mathbb{E}_{w^{k+1}}\|w^{k+1} - z^*\|^2\right] = \tau\mathbb{E}\left\|w^k - z^*\right\|^2 + (1-\tau)\mathbb{E}\|z^k - z^*\|^2, \quad (17)$$

Summing (16) and (17), we obtain

$$\mathbb{E}\|z^{k+1} - z^*\|^2 + \mathbb{E}\|w^{k+1} - z^*\|^2$$
$$\leq \mathbb{E}\|z^k - z^*\|^2 + \mathbb{E}\|w^k - z^*\|^2$$
$$- 2\gamma\mathbb{E}\left[\langle F(z^{k+1/2}), z^{k+1/2} - z^*\rangle\right]$$
$$- (1 - \tau - \gamma^2 C_q^2)\mathbb{E}\|w^k - z^{k+1/2}\|^2 - \tau\mathbb{E}\|z^{k+1/2} - z^k\|^2. \quad (18)$$

The property of the solution (3) gives

$$\mathbb{E}\|z^{k+1} - z^*\|^2 + \mathbb{E}\|w^{k+1} - z^*\|^2$$
$$\leq \mathbb{E}\|z^k - z^*\|^2 + \mathbb{E}\|w^k - z^*\|^2$$
$$- 2\gamma\mathbb{E}\left[\langle F(z^{k+1/2}) - F(z^*), z^{k+1/2} - z^*\rangle\right]$$

$$- (1 - \tau - \gamma^2 C_q^2)\mathbb{E}\|w^k - z^{k+1/2}\|^2 - \tau\mathbb{E}\|z^{k+1/2} - z^k\|^2.$$

And by Assumption 3.5 in strong monotone case we have

$$\mathbb{E}\|z^{k+1} - z^*\|^2 + \mathbb{E}\|w^{k+1} - z^*\|^2$$
$$\leq \mathbb{E}\|z^k - z^*\|^2 + \mathbb{E}\|w^k - z^*\|^2 - 2\gamma\mu\mathbb{E}\|z^{k+1/2} - z^*\|^2$$
$$- (1 - \tau - \gamma^2 C_q^2)\mathbb{E}\|w^k - z^{k+1/2}\|^2 - \tau\mathbb{E}\|z^{k+1/2} - z^k\|^2.$$

With $-\|a\|^2 \leq -\frac{1}{2}\|a + b\|^2 + \|b\|^2$ we deduce:

$$\mathbb{E}(\|z^{k+1} - z^*\|^2 + \mathbb{E}\|w^{k+1} - z^*\|^2)$$
$$\leq \left(1 - \frac{\mu\gamma}{2}\right)\mathbb{E}\left(\|z^k - z^*\|^2 + \|w^k - z^*\|^2\right)$$
$$- (1 - \tau - \mu\gamma - \gamma^2 C_q^2)\mathbb{E}\|w^k - z^{k+1/2}\|^2 - (\tau - \mu\gamma)\mathbb{E}\|z^{k+1/2} - z^k\|^2. \tag{19}$$

It remains only to choose $\gamma \leq \min\left\{\frac{\sqrt{1-\tau}}{2C_q}; \frac{1-\tau}{2\mu}\right\}$ and get

$$\mathbb{E}(\|z^{k+1} - z^*\|^2 + \mathbb{E}\|w^{k+1} - z^*\|^2) \leq \left(1 - \frac{\mu\gamma}{2}\right) \cdot \mathbb{E}\left(\|z^k - z^*\|^2 + \|w^k - z^*\|^2\right).$$

Running the recursion completes the proof.

$\square$

### D.1.2 Monotone case

We start from (13):

$$2\gamma\langle F(z^{k+1/2}), z^{k+1/2} - z\rangle$$
$$= \tau\|z^k - z\|^2 - \|z^{k+1} - z\|^2 + (1-\tau)\|w^k - z\|^2$$
$$- 2\gamma\langle Q^{\mathrm{serv}}\left[\frac{1}{M}\sum_{m=1}^M Q_m^{\mathrm{dev}}(F_m(z^{k+1/2}) - F_m(w^k))\right] + F(w^k) - F(z^{k+1/2}), z^{k+1/2} - z\rangle$$
$$+ \|z^{k+1} - z^{k+1/2}\|^2 - (1-\tau)\|w^k - z^{k+1/2}\|^2 - \tau\|z^{k+1/2} - z^k\|^2.$$

Adding both sides $\|w^{k+1} - z\|^2$ and making small rearrangement we have

$$2\gamma\langle F(z^{k+1/2}), z^{k+1/2} - z\rangle$$
$$\leq \left[\|z^k - z\|^2 + \|w^k - z\|^2\right] - \left[\|z^{k+1} - z\|^2 + \|w^{k+1} - z\|^2\right]$$
$$- \tau\|w^k - z\|^2 - (1-\tau)\|z^k - z\|^2 + \|w^{k+1} - z\|^2$$
$$- 2\gamma\langle Q^{\mathrm{serv}}\left[\frac{1}{M}\sum_{m=1}^M Q_m^{\mathrm{dev}}(F_m(z^{k+1/2}) - F_m(w^k))\right] + F(w^k) - F(z^{k+1/2}), z^{k+1/2} - z\rangle$$
$$- \tau\|z^{k+1/2} - z^k\|^2 - (1-\tau)\|z^{k+1/2} - w^k\|^2 + \|z^{k+1} - z^{k+1/2}\|^2.$$

Then we sum up over $k = 0, \ldots, K - 1$, take maximum of both sides over $z \in \mathcal{C}$, after take expectation and get

$$2\gamma \cdot \mathbb{E}\left[\max_{z \in \mathcal{C}}\sum_{k=0}^{K-1}\langle F(z^{k+1/2}), z^{k+1/2} - z\rangle\right] \leq \max_{z \in \mathcal{C}}\left[\|z^0 - z\|^2 + \|w^0 - z\|^2\right]$$
$$+ \mathbb{E}\left[\max_{z \in \mathcal{C}}\sum_{k=0}^{K-1}\left[-\tau\|w^k - z\|^2 - (1-\tau)\|z^k - z\|^2 + \|w^{k+1} - z\|^2\right]\right]$$
$$- \sum_{k=0}^{K-1}\left[\tau\mathbb{E}\left[\|z^{k+1/2} - z^k\|^2\right] + (1-\tau)\mathbb{E}\left[\|z^{k+1/2} - w^k\|^2\right] - \mathbb{E}\left[\|z^{k+1} - z^{k+1/2}\|^2\right]\right]$$

$$
+ 2\gamma \mathbb{E}\left[\max_{z \in \mathcal{C}} \sum_{k=0}^{K-1}\left[\left\langle Q^{\text{serv}}\left[\frac{1}{M}\sum_{m=1}^{M} Q_m^{\text{dev}}(F_m(z^{k+1/2}) - F_m(w^k))\right] + F(w^k) - F(z^{k+1/2}), z - z^{k+1/2}\right\rangle\right]\right].
$$

Applying (15) for $\mathbb{E}\left[\|z^{k+1} - z^{k+1/2}\|\right]$, we get

$$
\begin{aligned}
2\gamma \cdot \mathbb{E}\left[\max_{z \in \mathcal{C}} \sum_{k=0}^{K-1}\langle F(z^{k+1/2}), z^{k+1/2} - z\rangle\right] &\leq \max_{z \in \mathcal{C}}\left[\|z^0 - z\|^2 + \|w^0 - z\|^2\right] \\
&+ \mathbb{E}\left[\max_{z \in \mathcal{C}} \sum_{k=0}^{K-1}\left[-\tau\|w^k - z\|^2 - (1-\tau)\|z^k - z\|^2 + \|w^{k+1} - z\|^2\right]\right] \\
&- \sum_{k=0}^{K-1}\left[\tau\mathbb{E}\left[\|z^{k+1/2} - z^k\|^2\right] + (1 - \tau - \gamma^2 C_q^2)\mathbb{E}\left[\|z^{k+1/2} - w^k\|^2\right]\right] \\
&+ 2\gamma\mathbb{E}\left[\max_{z \in \mathcal{C}} \sum_{k=0}^{K-1}\left[\left\langle Q^{\text{serv}}\left[\frac{1}{M}\sum_{m=1}^{M} Q_m^{\text{dev}}(F_m(z^{k+1/2}) - F_m(w^k))\right] + F(w^k) - F(z^{k+1/2}), z - z^{k+1/2}\right\rangle\right]\right].
\end{aligned}
$$

With $\gamma \leq \frac{\sqrt{1-\tau}}{2C_q}$ we get

$$
\begin{aligned}
2\gamma \cdot \mathbb{E}\left[\max_{z \in \mathcal{C}} \sum_{k=0}^{K-1}\langle F(z^{k+1/2}), z^{k+1/2} - z\rangle\right] &\leq \max_{z \in \mathcal{C}}\left[\|z^0 - z\|^2 + \|w^0 - z\|^2\right] \\
&+ \mathbb{E}\left[\max_{z \in \mathcal{C}} \sum_{k=0}^{K-1}\left[-\tau\|w^k - z\|^2 - (1-\tau)\|z^k - z\|^2 + \|w^{k+1} - z\|^2\right]\right] \\
&+ 2\gamma\mathbb{E}\left[\max_{z \in \mathcal{C}} \sum_{k=0}^{K-1}\left[\left\langle Q^{\text{serv}}\left[\frac{1}{M}\sum_{m=1}^{M} Q_m^{\text{dev}}(F_m(z^{k+1/2}) - F_m(w^k))\right] + F(w^k) - F(z^{k+1/2}), z - z^{k+1/2}\right\rangle\right]\right].
\end{aligned}
\tag{20}
$$

To finish the proof we need to estimate terms in two last lines. We begin with
$\mathbb{E}\left[\max_{z \in \mathcal{C}} \sum_{k=0}^{K-1}\langle F(z^{k+1/2}) - Q^{\text{serv}}\left[\frac{1}{M}\sum_{m=1}^{M} Q_m^{\text{dev}}(F_m(z^{k+1/2}) - F_m(w^k))\right] - F(w^k), z^{k+1/2} - z\rangle\right]$.
Let define sequence $v$: $v^0 = z^0$, $v^{k+1} = v^k - \gamma\delta_k$ with $\delta^k = F(z^{k+1/2}) - Q^{\text{serv}}\left[\frac{1}{M}\sum_{m=1}^{M} Q_m^{\text{dev}}(F_m(z^{k+1/2}) - F_m(w^k))\right] - F(w^k)$. Then we have

$$
\sum_{k=0}^{K-1}\langle \delta^k, z^{k+1/2} - u\rangle = \sum_{k=0}^{K-1}\langle \delta^k, z^{k+1/2} - v^k\rangle + \sum_{k=0}^{K-1}\langle \delta^k, v^k - z\rangle. \tag{21}
$$

By the definition of $v^{k+1}$, we have

$$
\begin{aligned}
\langle \gamma\delta^k, v^k - z\rangle &= \langle \gamma\delta^k, v^k - v^{k+1}\rangle + \langle v^{k+1} - v^k, z - v^{k+1}\rangle \\
&= \langle \gamma\delta^k, v^k - v^{k+1}\rangle + \frac{1}{2}\|v^k - z\|^2 - \frac{1}{2}\|v^{k+1} - z\|^2 - \frac{1}{2}\|v^k - v^{k+1}\|^2 \\
&= \frac{\gamma^2}{2}\|\delta^k\|^2 + \frac{1}{2}\|v^k - v^{k+1}\|^2 + \frac{1}{2}\|v^k - z\|^2 - \frac{1}{2}\|v^{k+1} - z\|^2 - \frac{1}{2}\|v^k - v^{k+1}\|^2 \\
&= \frac{\gamma^2}{2}\|\delta^k\|^2 + \frac{1}{2}\|v^k - z\|^2 - \frac{1}{2}\|v^{k+1} - z\|^2.
\end{aligned}
$$

With (21) it gives

$$
\begin{aligned}
\sum_{k=0}^{K-1}\langle \delta^k, z^{k+1/2} - z\rangle &\leq \sum_{k=0}^{K-1}\langle \delta^k, z^{k+1/2} - v^k\rangle + \frac{1}{\gamma}\sum_{k=0}^{K-1}\left(\frac{\gamma^2}{2}\|\delta^k\|^2 + \frac{1}{2}\|v^k - z\|^2 - \frac{1}{2}\|v^{k+1} - z\|^2\right) \\
&\leq \sum_{k=0}^{K-1}\langle \delta^k, z^{k+1/2} - v^k\rangle + \frac{\gamma}{2}\sum_{k=0}^{K-1}\|\delta^k\|^2 + \frac{1}{2\gamma}\|v^0 - z\|^2.
\end{aligned}
$$

We take the maximum on $z$ and get

$$\max_{z \in \mathcal{C}} \sum_{k=0}^{K-1} \langle \delta^k, z^{k+1/2} - z \rangle \leq \sum_{k=0}^{K-1} \langle \delta^k, z^{k+1/2} - v^k \rangle + \frac{1}{2\gamma} \max_{z \in \mathcal{C}} \|v^0 - z\|^2$$

$$+ \frac{\gamma}{2} \sum_{k=0}^{K-1} \|F(z^{k+1/2}) - Q^{\mathrm{serv}}\left[\frac{1}{M}\sum_{m=1}^{M} Q_m^{\mathrm{dev}}(F_m(z^{k+1/2}) - F_m(w^k))\right] - F(w^k)\|^2.$$

Taking the full expectation, we get

$$\mathbb{E}\left[\max_{z \in \mathcal{C}} \sum_{k=0}^{K-1} \langle \delta^k, z^{k+1/2} - z \rangle\right] \leq \mathbb{E}\left[\sum_{k=0}^{K-1} \langle \delta^k, z^{k+1/2} - v^k \rangle\right] + \frac{1}{2\gamma} \max_{z \in \mathcal{C}} \|v^0 - z\|^2$$

$$+ \frac{\gamma}{2} \sum_{k=0}^{K-1} \mathbb{E}\left[\|F(z^{k+1/2}) - Q^{\mathrm{serv}}\left[\frac{1}{M}\sum_{m=1}^{M} Q_m^{\mathrm{dev}}(F_m(z^{k+1/2}) - F_m(w^k))\right] - F(w^k)\|^2\right]$$

$$= \mathbb{E}\left[\sum_{k=0}^{K-1} \langle \mathbb{E}\left[F(z^{k+1/2}) - Q^{\mathrm{serv}}\left[\frac{1}{M}\sum_{m=1}^{M} Q_m^{\mathrm{dev}}(F_m(z^{k+1/2}) - F_m(w^k))\right] - F(w^k) \mid z^{k+1/2} - v^k\right], z^{k+1/2} - v^k \rangle\right]$$

$$+ \frac{\gamma}{2} \sum_{k=0}^{K-1} \mathbb{E}\left[\|F(z^{k+1/2}) - Q^{\mathrm{serv}}\left[\frac{1}{M}\sum_{m=1}^{M} Q_m^{\mathrm{dev}}(F_m(z^{k+1/2}) - F_m(w^k))\right] - F(w^k)\|^2\right]$$

$$+ \frac{1}{2\gamma} \max_{z \in \mathcal{C}} \|v^0 - z\|^2$$

$$= \frac{\gamma}{2} \sum_{k=0}^{K-1} \mathbb{E}\left[\|F(z^{k+1/2}) - Q^{\mathrm{serv}}\left[\frac{1}{M}\sum_{m=1}^{M} Q_m^{\mathrm{dev}}(F_m(z^{k+1/2}) - F_m(w^k))\right] - F(w^k)\|^2\right]$$

$$+ \frac{1}{2\gamma} \max_{z \in \mathcal{C}} \|v^0 - z\|^2. \tag{22}$$

Now let us estimate $\mathbb{E}\left[\max_{z \in \mathcal{C}} \sum_{k=0}^{K-1} \left[-\tau\|w^k - z\|^2 - (1-\tau)\|z^k + z\|^2 + \|w^{k+1} - z\|^2\right]\right]$, for this we note that

$$\mathbb{E}\left[\max_{z \in \mathcal{C}} \sum_{k=0}^{K-1} \left[-\tau\|w^k - z\|^2 - (1-\tau)\|z^k - z\|^2 + \|w^{k+1} - z\|^2\right]\right]$$

$$= \mathbb{E}\left[\max_{z \in \mathcal{C}} \sum_{k=0}^{K-1} \left[-2\langle(1-\tau)z^k + \tau w^k - w^{k+1}, z\rangle - (1-\tau)\|z^k\|^2 - \tau\|w^k\|^2 + \|w^{k+1}\|^2\right]\right]$$

$$= \mathbb{E}\left[\max_{z \in \mathcal{C}} \sum_{k=0}^{K-1} \left[-2\langle(1-\tau)z^k + \tau w^k - w^{k+1}, z\rangle\right]\right]$$

$$+ \mathbb{E}\left[\sum_{k=0}^{K-1} -(1-\tau)\|z^k\|^2 - \tau\|w^k\|^2 + \|w^{k+1}\|^2\right].$$

One can note that by definition $w^{k+1}$: $\mathbb{E}\left[(1-\tau)\|z^k\|^2 + \tau\|w^k\|^2 - \|w^{k+1}\|^2\right] = 0$, then

$$\mathbb{E}\left[\max_{z \in \mathcal{C}} \sum_{k=0}^{K-1} \left[-\tau\|w^k - z\|^2 - (1-\tau)\|z^k - z\|^2 + \|w^{k+1} - z\|^2\right]\right]$$

$$= 2\mathbb{E}\left[\max_{z \in \mathcal{C}} \sum_{k=0}^{K-1} \langle(1-\tau)z^k + \tau w^k - w^{k+1}, -z\rangle\right]$$

$$= 2\mathbb{E}\left[\max_{z \in \mathcal{C}} \sum_{k=0}^{K-1} \langle(1-\tau)z^k + \tau w^k - w^{k+1}, z\rangle\right].$$

Further, one can carry out the reasoning similarly to chain for (22):

$$\mathbb{E}\left[\max_{z\in\mathcal{C}}\sum_{k=0}^{K-1}\left[\tau\|w^k-z\|^2+(1-\tau)\|z^k-z\|^2-\|w^{k+1}-z\|^2\right]\right]$$

$$\leq\sum_{k=0}^{K-1}\mathbb{E}\left[\|(1-\tau)z^{k+1}+\tau w^k-w^{k+1}\|^2\right]+\max_{z\in\mathcal{C}}\|v^0-z\|^2$$

$$=\sum_{k=0}^{K-1}\mathbb{E}\left[\|\mathbb{E}_{w^{k+1}}[w^{k+1}]-w^{k+1}\|^2\right]+\max_{z\in\mathcal{C}}\|v^0-z\|^2$$

$$=\sum_{k=0}^{K-1}\mathbb{E}\left[-\|\mathbb{E}_{w^{k+1}}[w^{k+1}]\|^2+\mathbb{E}_{w^{k+1}}\|w^{k+1}\|^2\right]+\max_{z\in\mathcal{C}}\|v^0-z\|^2$$

$$=\sum_{k=0}^{K-1}\mathbb{E}\left[-\|(1-\tau)z^k+\tau w^k\|^2+(1-\tau)\|z^k\|^2+\tau\|w^k\|^2\right]+\max_{z\in\mathcal{C}}\|v^0-z\|^2$$

$$=\sum_{k=0}^{K-1}\tau(1-\tau)\mathbb{E}\left[\|z^k-w^k\|^2\right]+\max_{z\in\mathcal{C}}\|v^0-z\|^2. \tag{23}$$

Substituting (22) and (23) in (20) we get

$$2\gamma\mathbb{E}\left[\max_{z\in\mathcal{C}}\sum_{k=0}^{K-1}\langle F(z^{k+1/2}),z^{k+1/2}-z\rangle\right]\leq\max_{z\in\mathcal{C}}\left[3\|z^0-z\|^2+\|w^0-z\|^2\right]$$

$$+\sum_{k=0}^{K-1}\tau(1-\tau)\mathbb{E}\left[\|z^k-w^k\|^2\right]$$

$$+\gamma^2\mathbb{E}\left[\|F(z^{k+1/2})-Q^{\mathrm{serv}}\left[\frac{1}{M}\sum_{m=1}^{M}Q_m^{\mathrm{dev}}(F_m(z^{k+1/2})-F_m(w^k))\right]-F(w^k)\|^2\right]. \tag{24}$$

Next we work separately with $\mathbb{E}\left[\|F(z^{k+1/2})-Q^{\mathrm{serv}}\left[\frac{1}{M}\sum\limits_{m=1}^{M}Q_m^{\mathrm{dev}}(F_m(z^{k+1/2})-F_m(w^k))\right]-F(w^k)\|^2\right]$:

$$\mathbb{E}\left[\left\|Q^{\mathrm{serv}}\left[\frac{1}{M}\sum_{m=1}^{M}Q_m^{\mathrm{dev}}(F_m(z^{k+1/2})-F_m(w^k))\right]+F(w^k)-F(z^{k+1/2})\right\|^2\right]$$

$$=\mathbb{E}\left[\left\|Q^{\mathrm{serv}}\left[\frac{1}{M}\sum_{m=1}^{M}Q_m^{\mathrm{dev}}(F_m(z^{k+1/2})-F_m(w^k))\right]\right\|^2\right]+\mathbb{E}\left[\|F(z^{k+1/2})-F(w^k)\|^2\right]$$

$$+\mathbb{E}\left[\langle Q^{\mathrm{serv}}\left[\frac{1}{M}\sum_{m=1}^{M}Q_m^{\mathrm{dev}}(F_m(z^{k+1/2})-F_m(w^k))\right];F(z^{k+1/2})-F(w^k)\rangle\right].$$

With (15) we get

$$\mathbb{E}\left[\left\|Q^{\mathrm{serv}}\left[\frac{1}{M}\sum_{m=1}^{M}Q_m^{\mathrm{dev}}(F_m(z^{k+1/2})-F_m(w^k))\right]+F(w^k)-F(z^{k+1/2})\right\|^2\right]$$

$$\leq C_q^2\mathbb{E}\left[\left\|z^{k+1/2}-w^k\right\|^2\right]+\mathbb{E}\left[\|F(z^{k+1/2})-F(w^k)\|^2\right]$$

$$+\mathbb{E}\left[\langle\frac{1}{M}\sum_{m=1}^{M}Q_m^{\mathrm{dev}}(F_m(z^{k+1/2})-F_m(w^k));F(z^{k+1/2})-F(w^k)\rangle\right]$$

$$=C_q^2\mathbb{E}\left[\left\|z^{k+1/2}-w^k\right\|^2\right]+2\mathbb{E}\left[\|F(z^{k+1/2})-F(w^k)\|^2\right]$$

$$\leq C_q^2 \mathbb{E}\left[\left\|z^{k+1/2} - w^k\right\|^2\right] + \frac{2}{M}\sum_{m=1}^{M} L_m^2 \cdot \mathbb{E}\left[\left\|z^{k+1/2} - w^k\right\|^2\right]. \tag{25}$$

With Assumption 3.4 and notation $\tilde{L}^2 = \frac{1}{M}\sum_{m=1}^{M} L_m^2$ from (24) and (25) we have

$$2\gamma\mathbb{E}\left[\max_{z\in\mathcal{C}}\sum_{k=0}^{K-1}\langle F(z^{k+1/2}), z^{k+1/2} - z\rangle\right] \leq \max_{z\in\mathcal{C}}\left[3\|z^0 - z\|^2 + \|w^0 - z\|^2\right]$$

$$+ \sum_{k=0}^{K-1}\left[\tau(1-\tau)\mathbb{E}\left[\|z^k - w^k\|^2\right] + \gamma^2(C_q^2 + 2\tilde{L}^2)\mathbb{E}\left[\|z^{k+1/2} - w^k\|^2\right]\right].$$

With $\gamma \leq \frac{\sqrt{1-\tau}}{2\sqrt{C_q^2 + 2\tilde{L}^2}}$ we deduce to

$$2\gamma\cdot\mathbb{E}\left[\max_{z\in\mathcal{C}}\sum_{k=0}^{K-1}\langle F(z^{k+1/2}), z^{k+1/2} - z\rangle\right] \leq \max_{z\in\mathcal{C}}\left[3\|z^0 - z\|^2 + \|w^0 - z\|^2\right]$$

$$+ (1-\tau)\sum_{k=0}^{K-1}\left[\mathbb{E}\left[\|z^{k+1} - w^k\|^2\right] + \mathbb{E}\left[\|z^{k+1/2} - w^k\|^2\right]\right]$$

$$\leq \max_{z\in\mathcal{C}}\left[3\|z^0 - z\|^2 + \|w^0 - z\|^2\right]$$

$$+ 3(1-\tau)\sum_{k=0}^{K-1}\left[\mathbb{E}\left[\|z^k - z^{k+1/2}\|^2\right] + \mathbb{E}\left[\|z^{k+1/2} - w^k\|^2\right]\right].$$

Let us go back to (19) with $\mu = 0$, $\gamma \leq \frac{\sqrt{1-\tau}}{2C_q}$ and get that

$$\mathbb{E}(\|z^{k+1} - z^*\|^2 + \mathbb{E}\|w^{k+1} - z^*\|^2)$$

$$\leq \mathbb{E}\left(\|z^k - z^*\|^2 + \|w^k - z^*\|^2\right)$$

$$- \frac{1-\tau}{2}\left(\mathbb{E}\|w^k - z^{k+1/2}\|^2 + \mathbb{E}\|z^{k+1/2} - z^k\|^2\right).$$

Hence substituting this we go to the end of the proof:

$$2\gamma\cdot\mathbb{E}\left[\max_{z\in\mathcal{C}}\sum_{k=0}^{K-1}\langle F(z^{k+1/2}), z^{k+1/2} - z\rangle\right] \leq \max_{z\in\mathcal{C}}\left[3\|z^0 - z\|^2 + \|w^0 - z\|^2\right]$$

$$+ 6\sum_{k=0}^{K-1}\left[\mathbb{E}\left(\|z^k - z^*\|^2 + \|w^k - z^*\|^2\right) - \mathbb{E}(\|z^{k+1} - z^*\|^2 + \mathbb{E}\|w^{k+1} - z^*\|^2)\right]$$

$$\leq \max_{z\in\mathcal{C}}\left[3\|z^0 - u\|^2 + \|w^0 - z\|^2\right] + 6\left(\|z^0 - z^*\|^2 + \|w^0 - z^*\|^2\right)$$

$$\leq \max_{z\in\mathcal{C}}\left[4\|z^0 - z\|^2\right] + 12\|z^0 - z^*\|^2.$$

It remains to slightly correct the convergence criterion by monotonicity of $F$:

$$\mathbb{E}\left[\max_{z\in\mathcal{C}}\sum_{k=0}^{K-1}\left[\langle F(z^{k+1/2}), z^{k+1/2} - z\rangle\right]\right]$$

$$\geq \mathbb{E}\left[\max_{z\in\mathcal{C}}\sum_{k=0}^{K-1}\left[\langle F(u), z^{k+1/2} - u\rangle\right]\right].$$

where we additionally use $\bar{z}^K = \frac{1}{K}\sum_{k=0}^{K-1} z^{k+1/2}$. This brings us to

$$\mathbb{E}\left[\max_{z\in\mathcal{C}}\left[\langle F(u), \left(\frac{1}{K}\sum_{k=0}^{K-1} z^{k+1/2}\right) - u\rangle\right]\right] \leq \frac{2\max_{z\in\mathcal{C}}\left[\|z^0 - z\|^2\right] + 6\|z^0 - z^*\|^2}{\gamma K}.$$

$\square$

### D.1.3 Non-monotone case

We start from (18)

$$\mathbb{E}\|z^{k+1} - z^*\|^2 + \mathbb{E}\|w^{k+1} - z^*\|^2$$
$$\leq \mathbb{E}\|z^k - z^*\|^2 + \mathbb{E}\|w^k - z^*\|^2$$
$$- 2\gamma\mathbb{E}\left[\langle F(z^{k+1/2}), z^{k+1/2} - z^*\rangle\right]$$
$$- (1 - \tau - \gamma^2 C_q^2)\mathbb{E}\|w^k - z^{k+1/2}\|^2 - \tau\mathbb{E}\|z^{k+1/2} - z^k\|^2.$$

And then use non-monotone case of Assumption 3.5:

$$\mathbb{E}\|z^{k+1} - z^*\|^2 + \mathbb{E}\|w^{k+1} - z^*\|^2$$
$$\leq \mathbb{E}\|z^k - z^*\|^2 + \mathbb{E}\|w^k - z^*\|^2$$
$$- (1 - \tau - \gamma^2 C_q^2)\mathbb{E}\|w^k - z^{k+1/2}\|^2 - \tau\mathbb{E}\|z^{k+1/2} - z^k\|^2.$$

With $\tau \geq \frac{1}{2}$ we get

$$\mathbb{E}\|z^{k+1} - z^*\|^2 + \mathbb{E}\|w^{k+1} - z^*\|^2$$
$$\leq \mathbb{E}\|z^k - z^*\|^2 + \mathbb{E}\|w^k - z^*\|^2 - (1 - \tau - \gamma^2 C_q^2)\mathbb{E}\|w^k - z^{k+1/2}\|^2$$
$$- \frac{1}{4}\mathbb{E}\|z^{k+1/2} - z^k\|^2 - \frac{1}{4}\mathbb{E}\|z^{k+1/2} - z^k\|^2$$
$$= \mathbb{E}\|z^k - z^*\|^2 + \mathbb{E}\|w^k - z^*\|^2 - (1 - \tau - \gamma^2 C_q^2)\mathbb{E}\|w^k - z^{k+1/2}\|^2$$
$$- \frac{1}{4}\mathbb{E}\|z^{k+1/2} - z^k\|^2 - \frac{1}{4}\mathbb{E}\|(1 - \tau)(w^k - z^k) - \gamma F(w^k)\|^2.$$

Using $-\|a\|^2 \leq -\frac{1}{2}\|a + b\|^2 + \|b\|^2$ gives

$$\mathbb{E}\|z^{k+1} - z^*\|^2 + \mathbb{E}\|w^{k+1} - z^*\|^2$$
$$\leq \mathbb{E}\|z^k - z^*\|^2 + \mathbb{E}\|w^k - z^*\|^2 - (1 - \tau - \gamma^2 C_q^2)\mathbb{E}\|w^k - z^{k+1/2}\|^2$$
$$- \frac{1}{4}\mathbb{E}\|z^{k+1/2} - z^k\|^2 - \frac{\gamma^2}{8}\mathbb{E}\|F(w^k)\|^2 + \frac{(1 - \tau)^2}{4}\mathbb{E}\|w^k - z^k\|^2$$
$$\leq \mathbb{E}\|z^k - z^*\|^2 + \mathbb{E}\|w^k - z^*\|^2 - (1 - \tau - \gamma^2 C_q^2)\mathbb{E}\|w^k - z^{k+1/2}\|^2$$
$$- \frac{1}{4}\mathbb{E}\|z^{k+1/2} - z^k\|^2 - \frac{\gamma^2}{8}\mathbb{E}\|F(w^k)\|^2$$
$$+ \frac{(1 - \tau)^2}{2}\mathbb{E}\|w^k - z^{k+1/2}\|^2 + \frac{(1 - \tau)^2}{2}\mathbb{E}\|z^{k+1/2} - z^k\|^2$$
$$\leq \mathbb{E}\|z^k - z^*\|^2 + \mathbb{E}\|w^k - z^*\|^2 - (1 - \tau - \gamma^2 C_q^2)\mathbb{E}\|w^k - z^{k+1/2}\|^2$$
$$- \frac{1}{4}\mathbb{E}\|z^{k+1/2} - z^k\|^2 - \frac{\gamma^2}{8}\mathbb{E}\|F(w^k)\|^2$$
$$+ \frac{1 - \tau}{4}\mathbb{E}\|w^k - z^{k+1/2}\|^2 + \frac{1}{8}\mathbb{E}\|z^{k+1/2} - z^k\|^2$$
$$\leq \mathbb{E}\|z^k - z^*\|^2 + \mathbb{E}\|w^k - z^*\|^2 - \left(\frac{1 - \tau}{2} - \gamma^2 C_q^2\right)\mathbb{E}\|w^k - z^{k+1/2}\|^2$$
$$- \frac{\gamma^2}{8}\mathbb{E}\|F(w^k)\|^2.$$

Choice of $\gamma \leq \frac{\sqrt{1-\tau}}{2C_q}$ gives

$$\mathbb{E}\|z^{k+1} - z^*\|^2 + \mathbb{E}\|w^{k+1} - z^*\|^2 \leq \mathbb{E}\|z^k - z^*\|^2 + \mathbb{E}\|w^k - z^*\|^2 - \frac{\gamma^2}{8}\mathbb{E}\|F(w^k)\|^2.$$

Summing over all $k$ from $0$ to $K - 1$ gives

$$\frac{1}{K}\sum_{k=0}^{K-1}\mathbb{E}\|F(w^k)\|^2 \leq \frac{8\mathbb{E}(\|z^0 - z^*\|^2 + \|w^0 - z^*\|^2)}{\gamma^2 K}.$$

$\square$

# E MASHA2: Handling Contractive Compressors

In this section, we provide additional information about Algorithm 2 – MASHA2. We give a full form of MASHA2 – see Algorithm 4.

Similarly with MASHA 1, this Algorithm, locally each device stores three vectors: a current point $z^k$, a reference point $w^k$ and a values $F(w^k)$ at this point. At each iteration, performs compressed communications from devices to the server (line 8) and from the server to devices (line 12). There is also one bit $b_k$ forwarding from the server (line 14). Additionally, communication can occur when $b_k$ is equal to 1 (with a small probability of $1 - \tau$) – in this case, each device $m$ updates point $w^{k+1} = z^k$, computes $F_m$ at this point, sends $F_m(w^{k+1})$ to the server without compression, the server calculates $F(w^{k+1})$ and sends it to devices also without compression. MASHA 2, similarly with MASHA 1, uses communications without compression, but very rarely (about once every $\frac{1}{1-\tau}$ iterations). Because, when $b_k = 0$, $w^{k+1} = w^k$ and all devices have locally value $F(w^{k+1}) = F(w^k)$ obtained sometime in previous communications (when $b = 1$).

---

**Algorithm 4** MASHA2

1: **Parameters:** Stepsize $\gamma > 0$, parameter $\tau$, number of iterations $K$.
2: **Initialization:** Choose $z^0 = w^0 \in \mathcal{Z}$, $e_m^0 = 0$, $e^0 = 0$.
3: Server sends to devices $z^0 = w^0$ and devices compute $F_m(w^0)$ and send to server and get $F(w^0)$
4: **for** $k = 0, 1, 2, \ldots, K - 1$ **do**
5:     **for** each device $m$ in parallel **do**
6:         $\bar{z}^k = \tau z^k + (1 - \tau) w^k$
7:         $z^{k+1/2} = \bar{z}_k - \gamma F(w^k)$
8:         Compute $F_m(z^{k+1/2})$ and send to server $C_m^{\text{dev}}(\gamma F_m(z^{k+1/2}) - \gamma F_m(w^k) + e_m^k)$
9:         $e_m^{k+1} = e_m^k + \gamma F_m(z^{k+1/2}) - \gamma F_m(w^k) - C_m^{\text{dev}}(\gamma F_m(z^{k+1/2}) - \gamma F_m(w^k) + e_m^k)$
10:     **end for**
11:     **for** server **do**
12:         Compute $g^k = C^{\text{serv}} \left[ \frac{1}{M} \sum_{m=1}^{M} C_m^{\text{dev}}(\gamma F_m(z^{k+1/2}) - \gamma F_m(w^k) + e_m^k) + e^k \right]$ & send to devices
13:         $e^{k+1} = e^k + \frac{1}{M} \sum_{m=1}^{M} C_m^{\text{dev}}(\gamma F_m(z^{k+1/2}) - \gamma F_m(w^k) + e_m^k) - g^k$
14:         Sends to devices one bit $b_k$: 1 with probability $1 - \tau$, 0 with with probability $\tau$
15:     **end for**
16:     **for** each device $m$ in parallel **do**
17:         $z^{k+1} = z^{k+1/2} - C^{\text{serv}} \left[ \frac{1}{M} \sum_{m=1}^{M} C_m^{\text{dev}}(\gamma F_m(z^{k+1/2}) - \gamma F_m(w^k) + e_m^k) + e^k \right]$
18:         **if** $b_k = 1$ **then**
19:             $w^{k+1} = z^k$
20:             Compute $F_m(w^{k+1})$ and it send to server; and get $F(w^{k+1})$
21:         **else**
22:             $w^{k+1} = w^k$
23:         **end if**
24:     **end for**
25: **end for**

---

Let us introduce the useful notation:

$$\hat{z}^k = z^k - e^k - \frac{1}{M} \sum_{m=1}^{M} e_m^k, \quad \hat{z}^{k+1/2} = z^{k+1/2} - e^k - \frac{1}{M} \sum_{m=1}^{M} e_m^k, \quad \hat{w}^k = w^k - e^k - \frac{1}{M} \sum_{m=1}^{M} e_m^k.$$

It is easy to verify that such sequences have a very useful property:

$$\hat{z}^{k+1} = z^{k+1} - e^{k+1} - \frac{1}{M} \sum_{m=1}^{M} e_m^{k+1}$$

$$= z^{k+1/2} - C^{\mathrm{serv}} \left[ \frac{1}{M} \sum_{m=1}^{M} C_m^{\mathrm{dev}}(\gamma F_m(z^{k+1/2}) - \gamma F_m(w^k) + e_m^k) + e^k \right]$$

$$- e^k - \frac{1}{M} \sum_{m=1}^{M} C_m^{\mathrm{dev}}(\gamma F_m(z^{k+1/2}) - \gamma F_m(w^k) + e_m^k)$$

$$+ C^{\mathrm{serv}} \left[ \frac{1}{M} \sum_{m=1}^{M} C_m^{\mathrm{dev}}(\gamma F_m(z^{k+1/2}) - \gamma F_m(w^k) + e_m^k) + e^k \right]$$

$$- \frac{1}{M} \sum_{m=1}^{M} \left[ e_m^k + \gamma \cdot F_m(z^{k+1/2}) - \gamma \cdot F_m(w^k) - C_m^{\mathrm{dev}}(\gamma \cdot F_m(z^{k+1/2}) - \gamma \cdot F_m(w^k) + e_m^k) \right]$$

$$= z^{k+1/2} - e^k - \frac{1}{M} \sum_{m=1}^{M} e_m^k - \gamma \cdot (F(z^{k+1/2}) - F(w^k))$$

$$= \hat{z}^{k+1/2} - \gamma \cdot (F(z^{k+1/2}) - F(w^k)). \tag{26}$$

The following theorem gives the convergence of MASHA2.

**Theorem E.1** *Let distributed variational inequality* (3) + (4) *is solved by Algorithm 4 with* $\tau \geq \frac{3}{4}$ *and biased compressor operators* (2): *on server with* $\delta^{serv}$ *parameter, on devices with* $\delta^{dev}$. *Let Assumption 3.4 and one case of Assumption 3.5 are satisfied. Then the following estimates holds*

- *in strongly-monotone case with* $\gamma \leq \min \left[ \frac{1-\tau}{8\mu}; \frac{\sqrt{1-\tau}}{2L + 165 \delta^{serv} \delta^{dev} \tilde{L}} \right]$:

$$\mathbb{E} \left( \|\hat{z}^K - z^*\|^2 + \|w^K - z^*\|^2 \right) \leq \left( 1 - \frac{\mu\gamma}{2} \right)^K \cdot 2\|z^0 - z^*\|^2;$$

- *in monotone case with* $\gamma \leq \frac{\sqrt{1-\tau}}{2L + 165 \delta^{serv} \delta^{dev} \tilde{L}}$:

$$\mathbb{E} \left[ \max_{z \in \mathcal{C}} \langle F(z), \left( \frac{1}{K} \sum_{k=0}^{K-1} z^{k+1/2} \right) - z \rangle \right] \leq \frac{2 \max_{z \in \mathcal{C}} \|z^0 - z\|^2 + 4\|z^0 - z^*\|^2}{\gamma K};$$

- *in non-monotone case with* $\gamma \leq \frac{\sqrt{1-\tau}}{2L + 165 \delta^{serv} \delta^{dev} \tilde{L}}$:

$$\mathbb{E} \left( \frac{1}{K} \sum_{k=0}^{K-1} \|F(w^k)\|^2 \right) \leq \frac{32 \mathbb{E}\|z^0 - z^*\|^2}{\gamma^2 K}.$$

Let us start with the only devices compression. For simplicity, we put $\tilde{L} = L$. We use the same reasoning as in Section D. At each iteration, the device sends to the server $\mathcal{O} \left( \frac{1}{\beta} + 1 - \tau \right)$ bits. Then the optimal choice $\tau$ is $1 - \frac{1}{\beta}$.

**Corollary E.2** *Let distributed variational inequality* (3) + (4) *is solved by Algorithm 4 without compression on server* ($\delta^{serv} = 1$) *and with biased compressor operators* (2) *on devices with* $\delta^{dev} = \delta$. *Let Assumption 3.4 and one case of Assumption 3.5 are satisfied. Then the following estimates holds*

- *in strongly-monotone case with* $\gamma \leq \min \left[ \frac{1}{8\mu\beta}; \frac{1}{167\delta\sqrt{\beta}L} \right]$:

$$\mathbb{E} \left( \|\hat{z}^K - z^*\|^2 + \|w^K - z^*\|^2 \right) \leq \left( 1 - \frac{\mu\gamma}{2} \right)^K \cdot 2\|z^0 - z^*\|^2;$$

- *in monotone case with* $\gamma \leq \frac{1}{167\delta\sqrt{\beta}L}$:

$$\mathbb{E} \left[ \max_{z \in \mathcal{C}} \langle F(z), \left( \frac{1}{K} \sum_{k=0}^{K-1} z^{k+1/2} \right) - z \rangle \right] \leq \frac{2 \max_{z \in \mathcal{C}} \|z^0 - z\|^2 + 4\|z^0 - z^*\|^2}{\gamma K};$$

• *in non-monotone case with $\gamma \leq \frac{1}{167\delta\sqrt{\beta}L}$:*

$$\mathbb{E}\left(\frac{1}{K}\sum_{k=0}^{K-1}\|F(w^k)\|^2\right) \leq \frac{32\mathbb{E}\|z^0 - z^*\|^2}{\gamma^2 K}.$$

In the line 2 of Table 1 we put complexities to achieve $\varepsilon$-solution.

Next, we add server compression. Now the transfer from the server is important. For simplicity, we put $Q^{\text{serv}} = Q_m^{\text{dev}} = Q$ with $q_m^{\text{dev}} = q$ and $\beta_m^{\text{dev}} = \beta$, also $L_m = \tilde{L} = L$. At each iteration, the device is sent to the server and the server to devices $\mathcal{O}\left(\frac{1}{\beta} + 1 - \tau\right)$ bits. Then the optimal choice $\tau$ is still $1 - \frac{1}{\beta}$.

**Corollary E.3** *Let distributed variational inequality (3) + (4) is solved by Algorithm 4 with $\tau \geq \frac{3}{4}$ and biased compressor operators (2): on server with $\delta^{serv} = \delta$ parameter, on devices with $\delta^{dev} = \delta$. Let Assumption 3.4 and one case of Assumption 3.5 are satisfied. Then the following estimates holds*

• *in strongly-monotone case with $\gamma \leq \min\left[\frac{1}{8\mu\beta}; \frac{1}{167\delta^2\sqrt{\beta}L}\right]$:*

$$\mathbb{E}\left(\|\hat{z}^K - z^*\|^2 + \|w^K - z^*\|^2\right) \leq \left(1 - \frac{\mu\gamma}{2}\right)^K \cdot 2\|z^0 - z^*\|^2;$$

• *in monotone case with $\gamma \leq \frac{1}{167\delta^2\sqrt{\beta}L}$:*

$$\mathbb{E}\left[\max_{z\in\mathcal{C}}\langle F(z), \left(\frac{1}{K}\sum_{k=0}^{K-1}z^{k+1/2}\right) - z\rangle\right] \leq \frac{2\max_{z\in\mathcal{C}}\|z^0 - z\|^2 + 4\|z^0 - z^*\|^2}{\gamma K};$$

• *in non-monotone case with $\gamma \leq \frac{1}{167\delta^2\sqrt{\beta}L}$:*

$$\mathbb{E}\left(\frac{1}{K}\sum_{k=0}^{K-1}\|F(w^k)\|^2\right) \leq \frac{32\mathbb{E}\|z^0 - z^*\|^2}{\gamma^2 K}.$$

In the line 4 of Table 1 we put complexities to achieve $\varepsilon$-solution.

### E.1 Proof of the convergence of MASHA2

**Proof of Theorem E.1:** We start from the following equalities for any $z$:

$$\|\hat{z}^{k+1} - z\|^2 = \|z^{k+1/2} - z\|^2 + 2\langle\hat{z}^{k+1} - z^{k+1/2}, z^{k+1/2} - z\rangle + \|\hat{z}^{k+1} - z^{k+1/2}\|^2,$$

$$\|z^{k+1/2} - z\|^2 = \|\hat{z}^k - z\|^2 + 2\langle z^{k+1/2} - \hat{z}^k, z^{k+1/2} - z\rangle - \|z^{k+1/2} - \hat{z}^k\|^2.$$

Summing up, we obtain

$$\begin{aligned}\|\hat{z}^{k+1} - z\|^2 = \|\hat{z}^k - z\|^2 &+ 2\langle\hat{z}^{k+1} - \hat{z}^k, z^{k+1/2} - z\rangle \\ &+ \|\hat{z}^{k+1} - z^{k+1/2}\|^2 - \|z^{k+1/2} - \hat{z}^k\|^2.\end{aligned} \tag{27}$$

Using that (10) and (26), we get

$$\begin{aligned}\|\hat{z}^{k+1} - z^{k+1/2}\|^2 &\leq 2\|\hat{z}^{k+1} - \hat{z}^{k+1/2}\|^2 + 2\|\hat{z}^{k+1/2} - z^{k+1/2}\|^2 \\ &= 2\gamma^2 \cdot \|F(z^{k+1/2}) - F(w^k)\|^2 + 2\left\|e^k - \frac{1}{M}\sum_{m=1}^{M}e_m^k\right\|^2 \\ &\leq 2\gamma^2 L^2 \cdot \|z^{k+1/2} - w^k\|^2 + 4\|e^k\|^2 + \frac{4}{M}\sum_{m=1}^{M}\|e_m^k\|^2\end{aligned}$$

$$\leq 2\gamma^2 L^2 \cdot \|z^{k+1/2} - w^k\|^2 + 4\|e^k\|^2 + \frac{4}{M}\sum_{m=1}^{M}\left\|e_m^k\right\|^2. \qquad (28)$$

Additionally, here we use that $F$ is $L$-Lipschitz (Assumption 3.4). Next, (27) with (28) gives

$$\|\hat{z}^{k+1} - z\|^2 \leq \|\hat{z}^k - z\|^2 + 2\langle \hat{z}^{k+1} - \hat{z}^k, z^{k+1/2} - z\rangle$$

$$+ 2\gamma^2 L^2 \cdot \|z^{k+1/2} - w^k\|^2 + 4\|e^k\|^2 + \frac{4}{M}\sum_{m=1}^{M}\left\|e_m^k\right\|^2$$

$$- \|z^{k+1/2} - \hat{z}^k\|^2. \qquad (29)$$

Now we consider the inner product $\langle \hat{z}^{k+1} - \hat{z}^k, z^{k+1/2} - z\rangle$. Using that

$$\hat{z}^{k+1} - \hat{z}^k = \hat{z}^{k+1} - \hat{z}^{k+1/2} + \hat{z}^{k+1/2} - \hat{z}^k = -\gamma \cdot (F(z^{k+1/2}) - F(w^k)) + z^{k+1/2} - z^k$$

$$= -\gamma \cdot F(z^{k+1/2}) + \bar{z}^k - z^k, \qquad (30)$$

and using the definition of $\bar{z}^k$ (line 6), we get

$$2\langle \hat{z}^{k+1} - \hat{z}^k, z^{k+1/2} - z\rangle = 2\langle -\gamma \cdot F(z^{k+1/2}) + \bar{z}^k - z^k, z^{k+1/2} - z\rangle$$

$$= -2\gamma\langle F(z^{k+1/2}), z^{k+1/2} - z\rangle + 2\langle \bar{z}^k - z^k, z^{k+1/2} - z\rangle$$

$$= -2\gamma\langle F(z^{k+1/2}), z^{k+1/2} - z\rangle + 2(1-\tau)\langle w^k - z^k, z^{k+1/2} - z\rangle.$$

Substituting in (29), we obtain

$$\|\hat{z}^{k+1} - z\|^2 \leq \|\hat{z}^k - z\|^2 - 2\gamma\langle F(z^{k+1/2}), z^{k+1/2} - z\rangle + 2(1-\tau)\langle w^k - z^k, z^{k+1/2} - z\rangle$$

$$+ 2\gamma^2 L^2 \cdot \|z^{k+1/2} - w^k\|^2 + 4\|e^k\|^2 + \frac{4}{M}\sum_{m=1}^{M}\left\|e_m^k\right\|^2$$

$$- \|z^{k+1/2} - \hat{z}^k\|^2.$$

The equality $2\langle a, b\rangle = \|a+b\|^2 - \|a\|^2 - \|b\|^2$ gives

$$\|\hat{z}^{k+1} - z\|^2 \leq \|\hat{z}^k - z\|^2 - 2\gamma\langle F(z^{k+1/2}), z^{k+1/2} - z\rangle$$
$$+ 2(1-\tau)\langle w^k - z^{k+1/2}, z^{k+1/2} - z\rangle$$
$$+ 2(1-\tau)\langle z^{k+1/2} - z^k, z^{k+1/2} - z\rangle$$
$$+ 2\gamma^2 L^2 \cdot \|z^{k+1/2} - w^k\|^2 + 4\|e^k\|^2 + \frac{4}{M}\sum_{m=1}^{M}\left\|e_m^k\right\|^2 - \|z^{k+1/2} - \hat{z}^k\|^2$$

$$= \|\hat{z}^k - z\|^2 - 2\gamma\langle F(z^{k+1/2}), z^{k+1/2} - z\rangle$$
$$+ (1-\tau)\|w^k - z\|^2 - (1-\tau)\|w^k - z^{k+1/2}\|^2 - (1-\tau)\|z^{k+1/2} - z\|^2$$
$$+ (1-\tau)\|z^{k+1/2} - z^k\|^2 + (1-\tau)\|z^{k+1/2} - z\|^2 - (1-\tau)\|z^k - z\|^2$$
$$+ 2\gamma^2 L^2 \cdot \|z^{k+1/2} - w^k\|^2 + 4\|e^k\|^2 + \frac{4}{M}\sum_{m=1}^{M}\left\|e_m^k\right\|^2 - \|z^{k+1/2} - \hat{z}^k\|^2$$

$$= \|\hat{z}^k - z\|^2 - (1-\tau)\|z^k - z^*\|^2 + (1-\tau)\|w^k - z\|^2$$
$$- 2\gamma\langle F(z^{k+1/2}), z^{k+1/2} - z\rangle - (1-\tau)\|w^k - z^{k+1/2}\|^2$$
$$+ 2\gamma^2 L^2 \|w^k - z^{k+1/2}\|^2 + 4\|e^k\|^2 + \frac{4}{M}\sum_{m=1}^{M}\left\|e_m^k\right\|^2$$
$$- \|z^{k+1/2} - \hat{z}^k\|^2 + (1-\tau)\|z^{k+1/2} - z^k\|^2$$

$$\leq \|\hat{z}^k - z\|^2 - (1-\tau)\|z^k - z\|^2 + (1-\tau)\|w^k - z\|^2$$
$$- 2\gamma\langle F(z^{k+1/2}), z^{k+1/2} - z^*\rangle - (1-\tau)\|w^k - z^{k+1/2}\|^2$$

$$+ 2\gamma^2 L^2 \|w^k - z^{k+1/2}\|^2 + 4\|e^k\|^2 + \frac{4}{M}\sum_{m=1}^{M}\left\|e_m^k\right\|^2$$

$$- \frac{1}{2}\|z^{k+1/2} - z^k\|^2 + \|z^k - \hat{z}^k\|^2 + (1-\tau)\|z^{k+1/2} - z^k\|^2.$$

With definition of $\hat{z}^k$ we get

$$\|\hat{z}^{k+1} - z\|^2 \le \|\hat{z}^k - z\|^2 - (1-\tau)\|z^k - z\|^2 + (1-\tau)\|w^k - z\|^2$$
$$- 2\gamma\langle F(z^{k+1/2}), z^{k+1/2} - z\rangle - (1 - \tau - 2\gamma^2 L^2)\|w^k - z^{k+1/2}\|^2$$
$$+ 6\|e^k\|^2 + \frac{6}{M}\sum_{m=1}^{M}\left\|e_m^k\right\|^2 - \left(\tau - \frac{1}{2}\right)\|z^{k+1/2} - z^k\|^2. \tag{31}$$

Next we will consider three cases of monotonicity separately.

### E.1.1  Strongly-monotone

We continue with (31) by putting $z = z^*$ and using optimality condition: $\langle F(z^*), z^{k+1/2} - z^*\rangle \le 0$.

$$\|\hat{z}^{k+1} - z^*\|^2 \le \|\hat{z}^k - z^*\|^2 - (1-\tau)\|z^k - z^*\|^2 + (1-\tau)\|w^k - z^*\|^2$$
$$- 2\gamma\langle F(z^{k+1/2}) - F(z^*), z^{k+1/2} - z^*\rangle - (1 - \tau - 2\gamma^2 L^2)\|w^k - z^{k+1/2}\|^2$$
$$+ 6\|e^k\|^2 + \frac{6}{M}\sum_{m=1}^{M}\left\|e_m^k\right\|^2 - \left(\tau - \frac{1}{2}\right)\|z^{k+1/2} - z^k\|^2.$$

Taking a full mathematical expectation, we obtain

$$\mathbb{E}\|\hat{z}^{k+1} - z^*\|^2 \le \mathbb{E}\|\hat{z}^k - z^*\|^2 - (1-\tau)\mathbb{E}\|z^k - z^*\|^2 + (1-\tau)\mathbb{E}\|w^k - z^*\|^2$$
$$- 2\gamma\mathbb{E}\left[\langle F(z^{k+1/2}) - F(z^*), z^{k+1/2} - z^*\rangle\right] - (1 - \tau - 2\gamma^2 L^2)\mathbb{E}\|w^k - z^{k+1/2}\|^2$$
$$+ 6\mathbb{E}\|e^k\|^2 + \frac{6}{M}\sum_{m=1}^{M}\mathbb{E}\left\|e_m^k\right\|^2 - \left(\tau - \frac{1}{2}\right)\mathbb{E}\|z^{k+1/2} - z^k\|^2. \tag{32}$$

Next, we take into account strong-monotonicity (Assumption 3.5 (SM)):

$$\mathbb{E}\|\hat{z}^{k+1} - z^*\|^2 \le \mathbb{E}\|\hat{z}^k - z^*\|^2 - (1-\tau)\mathbb{E}\|z^k - z^*\|^2 + (1-\tau)\mathbb{E}\|w^k - z^*\|^2$$
$$- 2\gamma\mu\mathbb{E}\|z^{k+1/2} - z^*\|^2 - (1 - \tau - 2\gamma^2 L^2)\mathbb{E}\|w^k - z^{k+1/2}\|^2$$
$$+ 6\mathbb{E}\|e^k\|^2 + \frac{6}{M}\sum_{m=1}^{M}\mathbb{E}\left\|e_m^k\right\|^2 - \left(\tau - \frac{1}{2}\right)\mathbb{E}\|z^{k+1/2} - z^k\|^2.$$

Taking a full mathematical expectation, we obtain

$$\mathbb{E}\|\hat{z}^{k+1} - z^*\|^2 \le \mathbb{E}\|\hat{z}^k - z^*\|^2 - (1-\tau)\mathbb{E}\|z^k - z^*\|^2 + (1-\tau)\mathbb{E}\|w^k - z^*\|^2$$
$$- 2\gamma\mu\mathbb{E}\|z^{k+1/2} - z^*\|^2 - (1 - \tau - 2\gamma^2 L^2)\mathbb{E}\|w^k - z^{k+1/2}\|^2$$
$$+ 6\mathbb{E}\|e^k\|^2 + \frac{6}{M}\sum_{m=1}^{M}\mathbb{E}\left\|e_m^k\right\|^2 - \left(\tau - \frac{1}{2}\right)\mathbb{E}\|z^{k+1/2} - z^k\|^2.$$

Then we use choice of $w^{k+1}$ (lines 14, 19, 22) and get

$$\mathbb{E}\|w^{k+1} - z^*\|^2 = \mathbb{E}\left[\mathbb{E}_{w^{k+1}}\|w^{k+1} - z^*\|^2\right] = \tau\mathbb{E}\left\|w^k - z^*\right\|^2 + (1-\tau)\mathbb{E}\|z^k - z^*\|^2, \tag{33}$$

Summing up the two previous expressions gives

$$\mathbb{E}\|\hat{z}^{k+1} - z^*\|^2 + \mathbb{E}\|w^{k+1} - z^*\|^2$$
$$\le \mathbb{E}\|\hat{z}^k - z^*\|^2 + \mathbb{E}\|w^k - z^*\|^2 - 2\gamma\mu\mathbb{E}\|z^{k+1/2} - z^*\|^2$$
$$- (1 - \tau - 2\gamma^2 L^2)\mathbb{E}\|w^k - z^{k+1/2}\|^2 - \left(\tau - \frac{1}{2}\right)\mathbb{E}\|z^{k+1/2} - z^k\|^2$$

$$+ 6\mathbb{E}\|e^k\|^2 + \frac{6}{M} \sum_{m=1}^{M} \mathbb{E}\left\|e_m^k\right\|^2.$$

Then we can weight previous expression by $p^k$ and get

$$\sum_{k=0}^{K-1} p^k \mathbb{E}\|\hat{z}^{k+1} - z^*\|^2 + \sum_{k=0}^{K-1} p^k \mathbb{E}\|w^{k+1} - z^*\|^2$$

$$\leq \sum_{k=0}^{K-1} p^k \mathbb{E}\|\hat{z}^k - z^*\|^2 + \sum_{k=0}^{K-1} p^k \mathbb{E}\|w^k - z^*\|^2 - 2\gamma\mu \sum_{k=0}^{K-1} p^k \mathbb{E}\|z^{k+1/2} - z^*\|^2$$

$$- (1 - \tau - 2\gamma^2 L^2) \cdot \sum_{k=0}^{K-1} p^k \mathbb{E}\|w^k - z^{k+1/2}\|^2 - \left(\tau - \frac{1}{2}\right) \cdot \sum_{k=0}^{K-1} p^k \mathbb{E}\|z^{k+1/2} - z^k\|^2$$

$$+ 6 \cdot \sum_{k=0}^{K-1} p^k \mathbb{E}\|e^k\|^2 + 6 \cdot \sum_{k=0}^{K-1} p^k \frac{1}{M} \sum_{m=1}^{M} \mathbb{E}\left\|e_m^k\right\|^2. \tag{34}$$

Next we will estimate "error" term:

$$\mathbb{E}\|e^{k+1}\|^2 = \mathbb{E}\left\| e^k + \frac{1}{M} \sum_{m=1}^{M} C_m^{\text{dev}}(\gamma F_m(z^{k+1/2}) - \gamma F_m(w^k) + e_m^k) \right.$$

$$\left. - C^{\text{serv}}\left[\frac{1}{M} \sum_{m=1}^{M} C_m^{\text{dev}}(\gamma F_m(z^{k+1/2}) - \gamma F_m(w^k) + e_m^k) + e^k\right] \right\|^2$$

$$\leq \left(1 - \frac{1}{\delta^{\text{serv}}}\right) \mathbb{E}\left\| e^k + \frac{1}{M} \sum_{m=1}^{M} C_m^{\text{dev}}(\gamma F_m(z^{k+1/2}) - \gamma F_m(w^k) + e_m^k) \right\|^2$$

$$\leq (1+c)\left(1 - \frac{1}{\delta^{\text{serv}}}\right) \mathbb{E}\left\|e^k\right\|^2$$

$$+ \left(1 + \frac{1}{c}\right)\left(1 - \frac{1}{\delta^{\text{serv}}}\right) \frac{1}{M} \sum_{m=1}^{M} \mathbb{E}\left\| C_m^{\text{dev}}(\gamma F_m(z^{k+1/2}) - \gamma F_m(w^k) + e_m^k) \right\|^2.$$

Here we use definition of biased compression (2), (10) and inequality $\|a + b\|^2 \leq (1 + c)\|a\|^2 + (1 + 1/c)\|b\|^2$ (for $c > 0$). Is is easy to prove that for baised compressor $C_m^{\text{dev}}$ from (2) it holds that $\|C_m^{\text{dev}}(x)\|^2 \leq 4\|x\|^2$ (see [10]). Then

$$\mathbb{E}\|e^{k+1}\|^2 \leq (1+c)\left(1 - \frac{1}{\delta^{\text{serv}}}\right) \mathbb{E}\left\|e^k\right\|^2$$

$$+ \left(1 + \frac{1}{c}\right)\left(1 - \frac{1}{\delta^{\text{serv}}}\right) \frac{4}{M} \sum_{m=1}^{M} \mathbb{E}\left\|\gamma F_m(z^{k+1/2}) - \gamma F_m(w^k) + e_m^k\right\|^2$$

$$\leq (1+c)\left(1 - \frac{1}{\delta^{\text{serv}}}\right) \mathbb{E}\left\|e^k\right\|^2$$

$$+ \gamma^2\left(1 + \frac{1}{c}\right)\left(1 - \frac{1}{\delta^{\text{serv}}}\right) \frac{8}{M} \sum_{m=1}^{M} \mathbb{E}\left\|F_m(z^{k+1/2}) - F_m(w^k)\right\|^2$$

$$+ \left(1 + \frac{1}{c}\right)\left(1 - \frac{1}{\delta^{\text{serv}}}\right) \frac{8}{M} \sum_{m=1}^{M} \mathbb{E}\left\|e_m^k\right\|^2$$

$$\leq (1+c)\left(1 - \frac{1}{\delta^{\text{serv}}}\right) \mathbb{E}\left\|e^k\right\|^2 + 8\gamma^2\tilde{L}^2\left(1 + \frac{1}{c}\right)\left(1 - \frac{1}{\delta^{\text{serv}}}\right) \mathbb{E}\left\|z^{k+1/2} - w^k\right\|^2$$

$$+ \left(1 + \frac{1}{c}\right)\left(1 - \frac{1}{\delta^{\text{serv}}}\right) \frac{8}{M} \sum_{m=1}^{M} \mathbb{E}\left\|e_m^k\right\|^2.$$

In the last we use Assumption 3.4 and definition of $\tilde{L}$ from this Assumption. With $c = \frac{1}{2(\delta-1)}$ we get

$$\mathbb{E}\|e^{k+1}\|^2 \le \left(1 - \frac{1}{2\delta^{\text{serv}}}\right)\mathbb{E}\left\|e^k\right\|^2 + 16\delta^{\text{serv}}\gamma^2\tilde{L}^2 \cdot \mathbb{E}\|z^{k+1/2} - w^k\|^2 + 16\delta^{\text{serv}} \cdot \frac{1}{M}\sum_{m=1}^{M}\mathbb{E}\left\|e_m^k\right\|^2$$

$$\le 16\delta^{\text{serv}}\gamma^2\tilde{L}^2\sum_{j=0}^{k}\left(1 - \frac{1}{2\delta^{\text{serv}}}\right)^{k-j} \cdot \left\|z^{j+1/2} - w^j\right\|^2$$

$$+ 16\delta^{\text{serv}}\sum_{j=0}^{k}\left(1 - \frac{1}{2\delta^{\text{serv}}}\right)^{k-j} \cdot \frac{1}{M}\sum_{m=1}^{M}\left\|e_m^j\right\|^2.$$

We weigh the sequence as follows $\sum_{k=0}^{K}p^k\mathbb{E}\left\|e^k\right\|^2$. Here we also assume $p$ such that $p^k \le p^j(1 + 1/4\delta^{\text{serv}})^{k-j}$. Then

$$\sum_{k=0}^{K-1}p^k\mathbb{E}\left\|e^k\right\|^2 \le 16\delta^{\text{serv}}\gamma^2\tilde{L}^2\sum_{k=0}^{K-1}p^k\sum_{j=0}^{k-1}\left(1 - \frac{1}{2\delta^{\text{serv}}}\right)^{k-j-1} \cdot \mathbb{E}\left\|z^{j+1/2} - w^j\right\|^2$$

$$+ 16\delta^{\text{serv}}\sum_{k=0}^{K-1}p^k\sum_{j=0}^{k-1}\left(1 - \frac{1}{2\delta^{\text{serv}}}\right)^{k-j-1} \cdot \frac{1}{M}\sum_{m=1}^{M}\mathbb{E}\left\|e_m^j\right\|^2$$

$$\le \frac{16\delta^{\text{serv}}\gamma^2\tilde{L}^2}{(1 - 1/2\delta^{\text{serv}})}\sum_{k=0}^{K-1}\sum_{j=0}^{k-1}p^j\left(1 + \frac{1}{4\delta^{\text{serv}}}\right)^{k-j}\left(1 - \frac{1}{2\delta^{\text{serv}}}\right)^{k-j} \cdot \mathbb{E}\left\|z^{j+1/2} - w^j\right\|^2$$

$$+ \frac{16\delta^{\text{serv}}}{(1 - 1/2\delta^{\text{serv}})}\sum_{k=0}^{K-1}\sum_{j=0}^{k-1}p^j\left(1 + \frac{1}{4\delta^{\text{serv}}}\right)^{k-j}\left(1 - \frac{1}{2\delta^{\text{serv}}}\right)^{k-j} \cdot \frac{1}{M}\sum_{m=1}^{M}\mathbb{E}\left\|e_m^j\right\|^2$$

$$\le \frac{16\delta^{\text{serv}}\gamma^2\tilde{L}^2}{(1 - 1/2\delta^{\text{serv}})}\sum_{k=0}^{K-1}\sum_{j=0}^{k-1}p^j\left(1 - \frac{1}{4\delta^{\text{serv}}}\right)^{k-j} \cdot \mathbb{E}\left\|z^{j+1/2} - w^j\right\|^2$$

$$+ \frac{16\delta^{\text{serv}}}{(1 - 1/2\delta^{\text{serv}})}\sum_{k=0}^{K-1}\sum_{j=0}^{k-1}p^j\left(1 - \frac{1}{4\delta^{\text{serv}}}\right)^{k-j} \cdot \frac{1}{M}\sum_{m=1}^{M}\mathbb{E}\left\|e_m^j\right\|^2$$

$$\le \frac{16\delta^{\text{serv}}\gamma^2\tilde{L}^2}{(1 - 1/2\delta^{\text{serv}})}\sum_{k=0}^{K-1}p^k\mathbb{E}\left\|z^{k+1/2} - w^k\right\|^2 \cdot \sum_{j=0}^{\infty}\left(1 - \frac{1}{4\delta^{\text{serv}}}\right)^j$$

$$+ \frac{16\delta^{\text{serv}}}{(1 - 1/2\delta^{\text{serv}})}\sum_{k=0}^{K-1}p^k\frac{1}{M}\sum_{m=1}^{M}\mathbb{E}\left\|e_m^k\right\|^2 \cdot \sum_{j=0}^{\infty}\left(1 - \frac{1}{4\delta^{\text{serv}}}\right)^j$$

$$\le 128(\delta^{\text{serv}})^2\gamma^2\tilde{L}^2\sum_{k=0}^{K-1}p^k\mathbb{E}\left\|z^{k+1/2} - w^k\right\|^2$$

$$+ 128(\delta^{\text{serv}})^2\sum_{k=0}^{K-1}p^k\frac{1}{M}\sum_{m=1}^{M}\mathbb{E}\left\|e_m^k\right\|^2. \tag{35}$$

Combining (34) with (35), we obtain

$$\sum_{k=0}^{K-1}p^k\left(\mathbb{E}\|\hat{z}^{k+1} - z^*\|^2 + \mathbb{E}\|w^{k+1} - z^*\|^2\right)$$

$$\le \sum_{k=0}^{K-1}p^k\mathbb{E}\|\hat{z}^k - z^*\|^2 + \sum_{k=0}^{K-1}p^k\mathbb{E}\|w^k - z^*\|^2 - 2\gamma\mu\sum_{k=0}^{K-1}p^k\mathbb{E}\|z^{k+1/2} - z^*\|^2$$

$$- (1 - \tau - 2\gamma^2L^2) \cdot \sum_{k=0}^{K-1}p^k\mathbb{E}\|w^k - z^{k+1/2}\|^2 - \left(\tau - \frac{1}{2}\right) \cdot \sum_{k=0}^{K-1}p^k\mathbb{E}\|z^{k+1/2} - z^k\|^2$$

$$+ 768(\delta^{\text{serv}})^2\gamma^2\tilde{L}^2 \cdot \sum_{k=0}^{K-1} p^k \mathbb{E}\left\|z^{k+1/2} - w^k\right\|^2$$

$$+ 768(\delta^{\text{serv}})^2 \cdot \sum_{k=0}^{K-1} p^k \frac{1}{M} \sum_{m=1}^{M} \mathbb{E}\left\|e_m^k\right\|^2 + 6 \cdot \sum_{k=0}^{K-1} p^k \frac{1}{M} \sum_{m=1}^{M} \mathbb{E}\left\|e_m^k\right\|^2$$

$$\leq \sum_{k=0}^{K-1} p^k \mathbb{E}\|\hat{z}^k - z^*\|^2 + \sum_{k=0}^{K-1} p^k \mathbb{E}\|w^k - z^*\|^2 - 2\gamma\mu \sum_{k=0}^{K-1} p^k \mathbb{E}\|z^{k+1/2} - z^*\|^2$$

$$- (1 - \tau - 2\gamma^2 L^2 - 768(\delta^{\text{serv}})^2\gamma^2\tilde{L}^2) \cdot \sum_{k=0}^{K-1} p^k \mathbb{E}\|w^k - z^{k+1/2}\|^2$$

$$- \left(\tau - \frac{1}{2}\right) \cdot \sum_{k=0}^{K-1} p^k \mathbb{E}\|z^{k+1/2} - z^k\|^2 + 775(\delta^{\text{serv}})^2 \cdot \sum_{k=0}^{K-1} p^k \frac{1}{M} \sum_{m=1}^{M} \mathbb{E}\left\|e_m^k\right\|^2.$$

Using $-\|a\|^2 \leq -\frac{1}{2}\|a + b\|^2 + \|b\|^2$, we get

$$\sum_{k=0}^{K-1} p^k \left(\mathbb{E}\|\hat{z}^{k+1} - z^*\|^2 + \mathbb{E}\|w^{k+1} - z^*\|^2\right)$$

$$\leq \sum_{k=0}^{K-1} p^k \left(1 - \frac{\mu\gamma}{2}\right) \left(\mathbb{E}\|\hat{z}^k - z^*\|^2 + \mathbb{E}\|w^k - z^*\|^2\right)$$

$$- (1 - \tau - \gamma\mu - 2\gamma^2 L^2 - 768(\delta^{\text{serv}})^2\gamma^2\tilde{L}^2) \cdot \sum_{k=0}^{K-1} p^k \mathbb{E}\|w^k - z^{k+1/2}\|^2$$

$$- \left(\tau - \frac{1}{2}\right) \cdot \sum_{k=0}^{K-1} p^k \mathbb{E}\|z^{k+1/2} - z^k\|^2 + 775(\delta^{\text{serv}})^2 \cdot \sum_{k=0}^{K-1} p^k \frac{1}{M} \sum_{m=1}^{M} \mathbb{E}\left\|e_m^k\right\|^2$$

$$+ \gamma\mu \cdot \sum_{k=0}^{K-1} p^k \mathbb{E}\|z^{k+1/2} - \hat{z}^k\|^2$$

$$\leq \sum_{k=0}^{K-1} p^k \left(1 - \frac{\mu\gamma}{2}\right) \left(\mathbb{E}\|\hat{z}^k - z^*\|^2 + \mathbb{E}\|w^k - z^*\|^2\right)$$

$$- (1 - \tau - \gamma\mu - 2\gamma^2 L^2 - 768(\delta^{\text{serv}})^2\gamma^2\tilde{L}^2) \cdot \sum_{k=0}^{K-1} p^k \mathbb{E}\|w^k - z^{k+1/2}\|^2$$

$$- \left(\tau - 2\gamma\mu - \frac{1}{2}\right) \cdot \sum_{k=0}^{K-1} p^k \mathbb{E}\|z^{k+1/2} - z^k\|^2$$

$$+ (775(\delta^{\text{serv}})^2 + 2\mu\gamma) \cdot \sum_{k=0}^{K-1} p^k \frac{1}{M} \sum_{m=1}^{M} \mathbb{E}\left\|e_m^k\right\|^2.$$

With $\gamma \leq \frac{1}{2\mu}$ we get

$$\sum_{k=0}^{K-1} p^k \left(\mathbb{E}\|\hat{z}^{k+1} - z^*\|^2 + \mathbb{E}\|w^{k+1} - z^*\|^2\right)$$

$$\leq \sum_{k=0}^{K-1} p^k \left(1 - \frac{\mu\gamma}{2}\right) \left(\mathbb{E}\|\hat{z}^k - z^*\|^2 + \mathbb{E}\|w^k - z^*\|^2\right)$$

$$- (1 - \tau - \gamma\mu - 2\gamma^2 L^2 - 768(\delta^{\text{serv}})^2\gamma^2\tilde{L}^2) \cdot \sum_{k=0}^{K-1} p^k \mathbb{E}\|w^k - z^{k+1/2}\|^2$$

$$- \left( \tau - 2\gamma\mu - \frac{1}{2} \right) \cdot \sum_{k=0}^{K-1} p^k \mathbb{E} \| z^{k+1/2} - z^k \|^2$$

$$+ 776 (\delta^{\mathrm{serv}})^2 \cdot \sum_{k=0}^{K-1} p^k \frac{1}{M} \sum_{m=1}^{M} \mathbb{E} \left\| e_m^k \right\|^2. \tag{36}$$

Next, we work with the other "error" term. The same way as for (35) we get

$$\frac{1}{M} \sum_{m=1}^{M} \mathbb{E} \left\| e_m^{k+1} \right\|^2 = \frac{1}{M} \sum_{m=1}^{M} \left\| e_m^k + \gamma \cdot F_m(z^{k+1/2}) - \gamma \cdot F_m(w^k) - C_m^{\mathrm{dev}}(\gamma \cdot F_m(z^{k+1/2}) - \gamma \cdot F_m(w^k) + e_m^k) \right\|^2$$

$$\leq \frac{1}{M} \sum_{m=1}^{M} \left( 1 - \frac{1}{\delta^{\mathrm{dev}}} \right) \left\| e_m^k + \gamma \cdot F_m(z^{k+1/2}) - \gamma \cdot F_m(w^k) \right\|^2$$

$$\leq \frac{1}{M} \sum_{m=1}^{M} (1+c) \left( 1 - \frac{1}{\delta^{\mathrm{dev}}} \right) \left\| e_m^k \right\|^2 + \left( 1 + \frac{1}{c} \right) \left( 1 - \frac{1}{\delta^{\mathrm{dev}}} \right) \gamma^2 \cdot \left\| F_m(z^{k+1/2}) - F_m(w^k) \right\|^2.$$

With $c = \frac{1}{2(\delta^{\mathrm{dev}} - 1)}$

$$\frac{1}{M} \sum_{m=1}^{M} \left\| e_m^{k+1} \right\|^2 \leq \frac{1}{M} \sum_{m=1}^{M} \left( 1 - \frac{1}{2\delta^{\mathrm{dev}}} \right) \left\| e_m^k \right\|^2 + 2\delta^{\mathrm{dev}} \gamma^2 \cdot \left\| F_m(z^{k+1/2}) - F_m(w^k) \right\|^2$$

$$\leq \left( 1 - \frac{1}{2\delta^{\mathrm{dev}}} \right) \cdot \frac{1}{M} \sum_{m=1}^{M} \left\| e_m^k \right\|^2 + 2\delta^{\mathrm{dev}} \gamma^2 \tilde{L}^2 \cdot \left\| z^{k+1/2} - w^k \right\|^2$$

$$\leq 2\delta^{\mathrm{dev}} \gamma^2 \tilde{L}^2 \sum_{j=0}^{k} \left( 1 - \frac{1}{2\delta^{\mathrm{dev}}} \right)^{k-j} \cdot \left\| z^{j+1/2} - w^j \right\|^2.$$

We weigh the sequence as follows $\sum_{k=0}^{K} p^k \frac{1}{M} \sum_{m=1}^{M} \left\| e_m^k \right\|^2$. Here we assume that $p$ such that $p^k \leq p^j (1 + 1/4\delta^{\mathrm{dev}})^{k-j}$. Then

$$\sum_{k=0}^{K-1} p^k \frac{1}{M} \sum_{m=1}^{M} \left\| e_m^k \right\|^2 \leq 2\delta^{\mathrm{dev}} \gamma^2 \tilde{L}^2 \sum_{k=0}^{K-1} p^k \sum_{j=0}^{k-1} \left( 1 - \frac{1}{2\delta^{\mathrm{dev}}} \right)^{k-j-1} \cdot \left\| z^{j+1/2} - w^j \right\|^2$$

$$\leq \frac{2\delta^{\mathrm{dev}} \gamma^2 \tilde{L}^2}{(1 - 1/2\delta^{\mathrm{dev}})} \sum_{k=0}^{K-1} \sum_{j=0}^{k-1} p^j \left( 1 + \frac{1}{4\delta^{\mathrm{dev}}} \right)^{k-j} \left( 1 - \frac{1}{2\delta^{\mathrm{dev}}} \right)^{k-j} \cdot \left\| z^{j+1/2} - w^j \right\|^2$$

$$\leq \frac{2\delta^{\mathrm{dev}} \gamma^2 \tilde{L}^2}{(1 - 1/2\delta^{\mathrm{dev}})} \sum_{k=0}^{K-1} \sum_{j=0}^{k-1} p^j \left( 1 - \frac{1}{4\delta^{\mathrm{dev}}} \right)^{k-j} \cdot \left\| z^{j+1/2} - w^j \right\|^2$$

$$\leq \frac{2\delta^{\mathrm{dev}} \gamma^2 \tilde{L}^2}{(1 - 1/2\delta^{\mathrm{dev}})} \sum_{k=0}^{K-1} p^k \left\| z^{k+1/2} - w^k \right\|^2 \cdot \sum_{j=0}^{\infty} \left( 1 - \frac{1}{4\delta^{\mathrm{dev}}} \right)^j$$

$$\leq 16 (\delta^{\mathrm{dev}})^2 \gamma^2 \tilde{L}^2 \sum_{k=0}^{K-1} p^k \left\| z^{k+1/2} - w^k \right\|^2. \tag{37}$$

(36) together with (37) gives

$$\sum_{k=0}^{K-1} p^k \left( \mathbb{E} \| \hat{z}^{k+1} - z^* \|^2 + \mathbb{E} \| w^{k+1} - z^* \|^2 \right)$$

$$\leq \sum_{k=0}^{K-1} p^k \left( 1 - \frac{\mu\gamma}{2} \right) \left( \mathbb{E} \| \hat{z}^k - z^* \|^2 + \mathbb{E} \| w^k - z^* \|^2 \right)$$

$$- (1 - \tau - \gamma\mu - 2\gamma^2 L^2 - 13200(\delta^{\mathrm{serv}})^2(\delta^{\mathrm{dev}})^2\gamma^2\tilde{L}^2) \cdot \sum_{k=0}^{K-1} p^k \mathbb{E}\|w^k - z^{k+1/2}\|^2$$

$$- \left(\tau - 2\gamma\mu - \frac{1}{2}\right) \cdot \sum_{k=0}^{K-1} p^k \mathbb{E}\|z^{k+1/2} - z^k\|^2. \tag{38}$$

With $\tau \geq \frac{3}{4}$, $\gamma \leq \min\left[\frac{1-\tau}{8\mu}; \frac{\sqrt{1-\tau}}{2L+165\delta^{\mathrm{serv}}\delta^{\mathrm{dev}}\tilde{L}}\right]$ we obtain

$$\sum_{k=0}^{K-1} p^k \left(\mathbb{E}\|\hat{z}^{k+1} - z^*\|^2 + \mathbb{E}\|w^{k+1} - z^*\|^2\right) \leq \sum_{k=0}^{K-1} p^k \left(1 - \frac{\mu\gamma}{2}\right)\left(\mathbb{E}\|\hat{z}^{k+1} - z^*\|^2 + \mathbb{E}\|w^{k+1} - z^*\|^2\right).$$

Then we just need to take $p = 1/(1 - \mu\gamma/2)$ (easy to check that $p^k \leq p^j(1 + 1/8\delta^{\mathrm{serv}})^{k-j}$ and $p^k \leq p^j(1 + 1/8\delta^{\mathrm{dev}})^{k-j}$ work with our $\gamma \leq \frac{\sqrt{1-\tau}}{2L+150\delta^{\mathrm{serv}}\delta^{\mathrm{dev}}\tilde{L}}$) and get

$$\mathbb{E}\left(\|\hat{z}^K - z^*\|^2 + \|w^K - z^*\|^2\right) \leq \left(1 - \frac{\mu\gamma}{2}\right)^K \left(\|\hat{z}^0 - z^*\|^2 + \|w^0 - z^*\|^2\right).$$

This ends the proof for strongly-monotone case.

$\square$

### E.1.2    Monotone

Let us comeback and start from (31):

$$2\gamma\langle F(z^{k+1/2}), z^{k+1/2} - z\rangle \leq \|\hat{z}^k - z\|^2 - \|\hat{z}^{k+1} - z\|^2 - (1-\tau)\|z^k - z\|^2 + (1-\tau)\|w^k - z\|^2$$
$$- (1 - \tau - 2\gamma^2 L^2)\|w^k - z^{k+1/2}\|^2$$
$$+ 6\|e^k\|^2 + \frac{6}{M}\sum_{m=1}^{M}\|e_m^k\|^2 - \left(\tau - \frac{1}{2}\right)\|z^{k+1/2} - z^k\|^2.$$

Then we use monotonicity (Assumption 3.5 (M)) and get

$$2\gamma\langle F(z), z^{k+1/2} - z\rangle \leq \|\hat{z}^k - z\|^2 - \|\hat{z}^{k+1} - z\|^2 - (1-\tau)\|z^k - z\|^2 + (1-\tau)\|w^k - z\|^2$$
$$- (1 - \tau - 2\gamma^2 L^2)\|w^k - z^{k+1/2}\|^2$$
$$+ 6\|e^k\|^2 + \frac{6}{M}\sum_{m=1}^{M}\|e_m^k\|^2 - \left(\tau - \frac{1}{2}\right)\|z^{k+1/2} - z^k\|^2$$
$$= \|\hat{z}^k - z\|^2 - \|\hat{z}^{k+1} - z\|^2 + \|w^k - z\|^2 - \|w^{k+1} - z\|^2$$
$$\|w^{k+1} - z\|^2 - (1-\tau)\|z^k - z\|^2 - \tau\|w^k - z\|^2$$
$$- (1 - \tau - 2\gamma^2 L^2)\|w^k - z^{k+1/2}\|^2$$
$$+ 6\|e^k\|^2 + \frac{6}{M}\sum_{m=1}^{M}\|e_m^k\|^2 - \left(\tau - \frac{1}{2}\right)\|z^{k+1/2} - z^k\|^2.$$

Next, we sum from 0 to $K - 1$:

$$2\gamma\sum_{k=0}^{K-1}\langle F(z), z^{k+1/2} - z\rangle$$
$$\leq \|\hat{z}^0 - z\|^2 + \|w^0 - z\|^2 + \sum_{k=0}^{K-1}\left(\|w^{k+1} - z\|^2 - (1-\tau)\|z^k - z\|^2 - \tau\|w^k - z\|^2\right)$$
$$- (1 - \tau - 2\gamma^2 L^2) \cdot \sum_{k=0}^{K-1}\|w^k - z^{k+1/2}\|^2 - \left(\tau - \frac{1}{2}\right) \cdot \sum_{k=0}^{K-1}\|z^{k+1/2} - z^k\|^2$$

$$+6 \cdot \sum_{k=0}^{K-1} \|e^k\|^2 + 6 \cdot \sum_{k=0}^{K} \frac{1}{M} \sum_{m=1}^{M} \|e_m^k\|^2.$$

Then we take maximum of both sides over $z \in \mathcal{C}$, after take expectation and get

$$2\gamma \mathbb{E} \left[ \max_{z \in \mathcal{C}} \sum_{k=0}^{K-1} \langle F(z), z^{k+1/2} - z \rangle \right]$$

$$\leq \mathbb{E} \left[ \max_{z \in \mathcal{C}} \|\hat{z}^0 - z\|^2 \right] + \mathbb{E} \left[ \max_{z \in \mathcal{C}} \|w^0 - z\|^2 \right]$$

$$+ \mathbb{E} \left[ \max_{z \in \mathcal{C}} \sum_{k=0}^{K-1} \left( \|w^{k+1} - z\|^2 - (1 - \tau)\|z^k - z\|^2 - \tau\|w^k - z\|^2 \right) \right]$$

$$- (1 - \tau - 2\gamma^2 L^2) \cdot \sum_{k=0}^{K-1} \mathbb{E}\|w^k - z^{k+1/2}\|^2 - \left( \tau - \frac{1}{2} \right) \cdot \sum_{k=0}^{K-1} \mathbb{E}\|z^{k+1/2} - z^k\|^2$$

$$+ 6 \cdot \sum_{k=0}^{K-1} \mathbb{E}\|e^k\|^2 + 6 \cdot \sum_{k=0}^{K-1} \frac{1}{M} \sum_{m=1}^{M} \mathbb{E}\|e_m^k\|^2.$$

We star with using (35) with $p = 1$ and get

$$2\gamma \mathbb{E} \left[ \max_{z \in \mathcal{C}} \sum_{k=0}^{K-1} \langle F(z), z^{k+1/2} - z \rangle \right]$$

$$\leq \mathbb{E} \left[ \max_{z \in \mathcal{C}} \|\hat{z}^0 - z\|^2 \right] + \mathbb{E} \left[ \max_{z \in \mathcal{C}} \|w^0 - z\|^2 \right]$$

$$+ \mathbb{E} \left[ \max_{z \in \mathcal{C}} \sum_{k=0}^{K-1} \left( \|w^{k+1} - z\|^2 - (1 - \tau)\|z^k - z\|^2 - \tau\|w^k - z\|^2 \right) \right]$$

$$- (1 - \tau - 2\gamma^2 L^2 - 768(\delta^{\mathrm{serv}})^2 \gamma^2 \tilde{L}^2) \cdot \sum_{k=0}^{K-1} \mathbb{E}\|w^k - z^{k+1/2}\|^2$$

$$- \left( \tau - \frac{1}{2} \right) \cdot \sum_{k=0}^{K-1} \mathbb{E}\|z^{k+1/2} - z^k\|^2 + 775(\delta^{\mathrm{serv}})^2 \cdot \sum_{k=0}^{K} p^k \frac{1}{M} \sum_{m=1}^{M} \mathbb{E}\|e_m^k\|^2.$$

And then (37) (also with $p = 1$):

$$2\gamma \mathbb{E} \left[ \max_{z \in \mathcal{C}} \sum_{k=0}^{K-1} \langle F(z), z^{k+1/2} - z \rangle \right]$$

$$\leq \mathbb{E} \left[ \max_{z \in \mathcal{C}} \|\hat{z}^0 - z\|^2 \right] + \mathbb{E} \left[ \max_{z \in \mathcal{C}} \|w^0 - z\|^2 \right]$$

$$+ \mathbb{E} \left[ \max_{z \in \mathcal{C}} \sum_{k=0}^{K-1} \left( \|w^{k+1} - z\|^2 - (1 - \tau)\|z^k - z\|^2 - \tau\|w^k - z\|^2 \right) \right]$$

$$- (1 - \tau - 2\gamma^2 L^2 - 13200(\delta^{\mathrm{serv}})^2 (\delta^{\mathrm{dev}})^2 \gamma^2 \tilde{L}^2) \cdot \sum_{k=0}^{K-1} \mathbb{E}\|w^k - z^{k+1/2}\|^2$$

$$- \left( \tau - \frac{1}{2} \right) \cdot \sum_{k=0}^{K-1} \mathbb{E}\|z^{k+1/2} - z^k\|^2.$$

With $t \geq \frac{3}{4}$ and $\gamma \leq \frac{\sqrt{1-\tau}}{2L + 165\delta^{\mathrm{serv}}\delta^{\mathrm{dev}}\tilde{L}}$ we get

$$2\gamma \mathbb{E} \left[ \max_{z \in \mathcal{C}} \sum_{k=0}^{K-1} \langle F(z), z^{k+1/2} - z \rangle \right]$$

$$\leq \mathbb{E}\left[\max_{z\in\mathcal{C}}\|\hat{z}^0 - z\|^2\right] + \mathbb{E}\left[\max_{z\in\mathcal{C}}\|w^0 - z\|^2\right]$$

$$+ \mathbb{E}\left[\max_{z\in\mathcal{C}}\sum_{k=0}^{K-1}\left(\|w^{k+1} - z\|^2 - (1-\tau)\|z^k - z\|^2 - \tau\|w^k - z\|^2\right)\right]. \tag{39}$$

Let us estimate $\mathbb{E}\left[\max_{z\in\mathcal{C}}\sum_{k=0}^{K-1}\left(\|w^{k+1} - z\|^2 - (1-\tau)\|z^k - z\|^2 - \tau\|w^k - z\|^2\right)\right]$. For this we note that

$$\mathbb{E}\left[\max_{z\in\mathcal{C}}\sum_{k=0}^{K-1}\left[-\tau\|w^k - z\|^2 - (1-\tau)\|z^k - z\|^2 + \|w^{k+1} - z\|^2\right]\right]$$

$$= \mathbb{E}\left[\max_{z\in\mathcal{C}}\sum_{k=0}^{K-1}\left[-2\langle(1-\tau)z^k + \tau w^k - w^{k+1}, z\rangle - (1-\tau)\|z^k\|^2 - \tau\|w^k\|^2 + \|w^{k+1}\|^2\right]\right]$$

$$= \mathbb{E}\left[\max_{z\in\mathcal{C}}\sum_{k=0}^{K-1}\left[-2\langle(1-\tau)z^k + \tau w^k - w^{k+1}, z\rangle\right]\right]$$

$$+ \mathbb{E}\left[\sum_{k=0}^{K-1}-(1-\tau)\|z^k\|^2 - \tau\|w^k\|^2 + \|w^{k+1}\|^2\right].$$

One can note that by definition $w^{k+1}$: $\mathbb{E}\left[(1-\tau)\|z^k\|^2 + \tau\|w^k\|^2 - \|w^{k+1}\|^2\right] = 0$, then

$$\mathbb{E}\left[\max_{z\in\mathcal{C}}\sum_{k=0}^{K-1}\left[-\tau\|w^k - z\|^2 - (1-\tau)\|z^k - z\|^2 + \|w^{k+1} - z\|^2\right]\right]$$

$$= 2\mathbb{E}\left[\max_{z\in\mathcal{C}}\sum_{k=0}^{K-1}\langle(1-\tau)z^k + \tau w^k - w^{k+1}, -z\rangle\right]$$

$$= 2\mathbb{E}\left[\max_{z\in\mathcal{C}}\sum_{k=0}^{K-1}\langle(1-\tau)z^k + \tau w^k - w^{k+1}, z\rangle\right].$$

Let define sequence $v$: $v^0 = z^0$, $v^{k+1} = v^k - \delta_k$ with $\delta^k = (1-\tau)z^k + \tau w^k - w^{k+1}$. Then we have

$$\sum_{k=0}^{K-1}\langle\delta^k, z^{k+1/2} - z\rangle = \sum_{k=0}^{K-1}\langle\delta^k, z^{k+1/2} - v^k\rangle + \sum_{k=0}^{K-1}\langle\delta^k, v^k - z\rangle. \tag{40}$$

By the definition of $v^{k+1}$, we have for all $z$

$$\langle v^{k+1} - v^k + \delta^k, z - v^{k+1}\rangle = 0.$$

Rewriting this inequality, we get

$$\langle\delta^k, v^k - z\rangle = \langle\delta^k, v^k - v^{k+1}\rangle + \langle v^{k+1} - v^k, z - v^{k+1}\rangle$$

$$= \langle\delta^k, v^k - v^{k+1}\rangle + \frac{1}{2}\|v^k - z\|^2 - \frac{1}{2}\|v^{k+1} - z\|^2 - \frac{1}{2}\|v^k - v^{k+1}\|^2$$

$$= \frac{1}{2}\|\delta^k\|^2 + \frac{1}{2}\|v^k - v^{k+1}\|^2 + \frac{1}{2}\|v^k - z\|^2 - \frac{1}{2}\|v^{k+1} - z\|^2 - \frac{1}{2}\|v^k - v^{k+1}\|^2$$

$$= \frac{1}{2}\|\delta^k\|^2 + \frac{1}{2}\|v^k - z\|^2 - \frac{1}{2}\|v^{k+1} - z\|^2.$$

With (40) it gives

$$\sum_{k=0}^{K-1}\langle\delta^k, z^{k+1/2} - z\rangle \leq \sum_{k=0}^{K-1}\langle\delta^k, z^{k+1/2} - v^k\rangle + \sum_{k=0}^{K-1}\left(\frac{1}{2}\|\delta^k\|^2 + \frac{1}{2}\|v^k - z\|^2 - \frac{1}{2}\|v^{k+1} - z\|^2\right)$$

$$\leq \sum_{k=0}^{K-1} \langle \delta^k, z^{k+1/2} - v^k \rangle + \frac{1}{2} \sum_{k=0}^{K-1} \|\delta^k\|^2 + \frac{1}{2} \|z^0 - z\|^2.$$

We take the maximum on $z$ and get

$$\max_{z \in \mathcal{C}} \sum_{k=0}^{K-1} \langle \delta^k, z^{k+1/2} - z \rangle \leq \sum_{k=0}^{K-1} \langle \delta^k, z^{k+1/2} - v^k \rangle$$

$$+ \frac{1}{2} \sum_{k=0}^{K-1} \|(1-\tau)z^k + \tau w^k - w^{k+1}\|^2 + \frac{1}{2} \max_{z \in \mathcal{C}} \|z^0 - z\|^2.$$

Taking the full expectation, we get

$$\mathbb{E}\left[\max_{z \in \mathcal{C}} \sum_{k=0}^{K-1} \langle \delta^k, z^{k+1/2} - z \rangle\right] \leq \mathbb{E}\left[\sum_{k=0}^{K-1} \langle \delta^k, z^{k+1/2} - v^k \rangle\right]$$

$$+ \frac{1}{2} \sum_{k=0}^{K-1} \mathbb{E}\left[\|(1-\tau)z^k + \tau w^k - w^{k+1}\|^2\right] + \frac{1}{2}\mathbb{E}\left[\max_{z \in \mathcal{C}} \|z^0 - z\|^2\right]$$

$$= \mathbb{E}\left[\sum_{k=0}^{K-1} \langle \mathbb{E}_{w^{k+1}}\left[(1-\tau)z^k + \tau w^k - w^{k+1}\right], z^{k+1/2} - v^k \rangle\right]$$

$$+ \frac{1}{2} \sum_{k=0}^{K-1} \mathbb{E}\left[\|(1-\tau)z^k + \tau w^k - w^{k+1}\|^2\right] + \frac{1}{2}\mathbb{E}\left[\max_{z \in \mathcal{C}} \|z^0 - z\|^2\right]$$

$$= \frac{1}{2} \sum_{k=0}^{K-1} \mathbb{E}\left[\|(1-\tau)z^k + \tau w^k - w^{k+1}\|^2\right] + \frac{1}{2}\mathbb{E}\left[\max_{z \in \mathcal{C}} \|z^0 - z\|^2\right]$$

$$\leq \frac{1}{2} \sum_{k=0}^{K-1} \mathbb{E}\left[\|(1-\tau)z^k + \tau w^k - w^{k+1}\|^2\right] + \frac{1}{2}\mathbb{E}\left[\max_{z \in \mathcal{C}} \|z^0 - z\|^2\right]$$

$$= \frac{1}{2} \sum_{k=0}^{K-1} \mathbb{E}\left[\|\mathbb{E}_{w^{k+1}}[w^{k+1}] - w^{k+1}\|^2\right] + \frac{1}{2}\mathbb{E}\left[\max_{z \in \mathcal{C}} \|z^0 - z\|^2\right]$$

$$= \frac{1}{2} \sum_{k=0}^{K-1} \mathbb{E}\left[-\|\mathbb{E}_{w^{k+1}}[w^{k+1}]\|^2 + \mathbb{E}_{w^{k+1}}\|w^{k+1}\|^2\right] + \frac{1}{2}\mathbb{E}\left[\max_{z \in \mathcal{C}} \|z^0 - z\|^2\right]$$

$$= \frac{1}{2} \sum_{k=0}^{K-1} \mathbb{E}\left[-\|(1-\tau)z^k + \tau w^k\|^2 + (1-\tau)\|z^k\|^2 + \tau\|w^k\|^2\right] + \frac{1}{2}\mathbb{E}\left[\max_{z \in \mathcal{C}} \|z^0 - z\|^2\right]$$

$$= \frac{1}{2} \sum_{k=0}^{K-1} \tau(1-\tau)\mathbb{E}\left[\|z^k - w^k\|^2\right] + \frac{1}{2}\mathbb{E}\left[\max_{z \in \mathcal{C}} \|z^0 - z\|^2\right].$$

Finally, we have

$$\mathbb{E}\left[\max_{z \in \mathcal{C}} \sum_{k=0}^{K-1} \left[-\tau\|w^k - z\|^2 - (1-\tau)\|z^k - z\|^2 + \|w^{k+1} - z\|^2\right]\right]$$

$$\leq \sum_{k=0}^{K-1} \tau(1-\tau)\mathbb{E}\left[\|z^k - w^k\|^2\right] + \max_{z \in \mathcal{C}} \|z^0 - z\|^2. \tag{41}$$

Together with (39) we obtain

$$2\gamma\mathbb{E}\left[\max_{z \in \mathcal{C}} \sum_{k=0}^{K-1} \langle F(z), z^{k+1/2} - z \rangle\right]$$

$$\leq \mathbb{E}\left[\max_{z \in \mathcal{C}} \|\hat{z}^0 - z\|^2\right] + \mathbb{E}\left[\max_{z \in \mathcal{C}} \|w^0 - z\|^2\right]$$

$$+ \mathbb{E}\left[\max_{z \in \mathcal{C}} \|z^0 - z\|^2\right] + (1-\tau) \cdot \sum_{k=0}^{K-1} \mathbb{E}\left[\|z^k - w^k\|^2\right]$$

$$\leq \mathbb{E}\left[\max_{z \in \mathcal{C}} \|\hat{z}^0 - z\|^2\right] + \mathbb{E}\left[\max_{z \in \mathcal{C}} \|w^0 - z\|^2\right] + \mathbb{E}\left[\max_{z \in \mathcal{C}} \|z^0 - z\|^2\right]$$

$$+ 2(1-\tau) \cdot \sum_{k=0}^{K-1} \left( \mathbb{E}\left[\|z^{k+1/2} - w^k\|^2\right] + \mathbb{E}\left[\|z^{k+1/2} - z^k\|^2\right] \right). \tag{42}$$

Let us use (38) with $p = 1$ and $\mu = 0$ (monotone case):

$$\sum_{k=0}^{K-1} \left( \mathbb{E}\|\hat{z}^{k+1} - z^*\|^2 + \mathbb{E}\|w^{k+1} - z^*\|^2 \right)$$

$$\leq \sum_{k=0}^{K-1} \left( \mathbb{E}\|\hat{z}^k - z^*\|^2 + \mathbb{E}\|w^k - z^*\|^2 \right)$$

$$- (1 - \tau - 2\gamma^2 L^2 - 13200(\delta^{\mathrm{serv}})^2(\delta^{\mathrm{dev}})^2 \gamma^2 \tilde{L}^2) \cdot \sum_{k=0}^{K-1} \mathbb{E}\|w^k - z^{k+1/2}\|^2$$

$$- \left( \tau - \frac{1}{2} \right) \cdot \sum_{k=0}^{K-1} \mathbb{E}\|z^{k+1/2} - z^k\|^2.$$

Taking into account that $\gamma \leq \frac{\sqrt{1-\tau}}{2L + 165\delta^{\mathrm{serv}}\delta^{\mathrm{dev}}\tilde{L}}$ we get

$$\sum_{k=0}^{K-1} \left( \mathbb{E}\|\hat{z}^{k+1} - z^*\|^2 + \mathbb{E}\|w^{k+1} - z^*\|^2 \right) \leq \sum_{k=0}^{K-1} \left( \mathbb{E}\|\hat{z}^k - z^*\|^2 + \mathbb{E}\|w^k - z^*\|^2 \right)$$

$$- \frac{1-\tau}{2} \cdot \sum_{k=0}^{K-1} \left( \mathbb{E}\|w^k - z^{k+1/2}\|^2 + \mathbb{E}\|z^{k+1/2} - z^k\|^2 \right).$$

Small rearrangement gives

$$2(1-\tau) \cdot \sum_{k=0}^{K-1} \left( \mathbb{E}\|w^k - z^{k+1/2}\|^2 + \mathbb{E}\|z^{k+1/2} - z^k\|^2 \right) \leq 4 \left( \mathbb{E}\|\hat{z}^0 - z^*\|^2 + \mathbb{E}\|w^0 - z^*\|^2 \right).$$

Substituting this expression to (42), we get:

$$2\gamma \mathbb{E}\left[ \max_{z \in \mathcal{C}} \sum_{k=0}^{K-1} \langle F(z), z^{k+1/2} - z \rangle \right] \leq \mathbb{E}\left[ \max_{z \in \mathcal{C}} \|\hat{z}^0 - z\|^2 \right] + 2\mathbb{E}\left[ \max_{z \in \mathcal{C}} \|z^0 - z\|^2 \right]$$

$$+ 4 \left( \mathbb{E}\|\hat{z}^0 - z^*\|^2 + \mathbb{E}\|w^0 - z^*\|^2 \right).$$

Then we can obtain

$$\mathbb{E}\left[ \max_{z \in \mathcal{C}} \langle F(z), \left( \frac{1}{K} \sum_{k=0}^{K-1} z^{k+1/2} \right) - z \rangle \right] \leq \frac{2\max_{z \in \mathcal{C}} \|z^0 - z\|^2 + 4\|z^0 - z^*\|^2}{\gamma K},$$

and finish the proof.

$\square$

### E.1.3  Non-monotone

Again we start from (31):

$$\mathbb{E}\|\hat{z}^{k+1} - z\|^2 \leq \mathbb{E}\|\hat{z}^k - z\|^2 - (1-\tau)\mathbb{E}\|z^k - z\|^2 + (1-\tau)\mathbb{E}\|w^k - z\|^2$$

$$- 2\gamma \mathbb{E}\left[ \langle F(z^{k+1/2}), z^{k+1/2} - z \rangle \right] - (1 - \tau - 2\gamma^2 L^2)\mathbb{E}\|w^k - z^{k+1/2}\|^2$$

$$+ 6\mathbb{E}\|e^k\|^2 + \frac{6}{M}\sum_{m=1}^{M}\mathbb{E}\left\|e_m^k\right\|^2 - \left(\tau - \frac{1}{2}\right)\mathbb{E}\|z^{k+1/2} - z^k\|^2.$$

Putting $z = z^*$ and using non-monotonicity (Assumption 3.5 (NM)), we get

$$\mathbb{E}\|\hat{z}^{k+1} - z^*\|^2 \leq \mathbb{E}\|\hat{z}^k - z^*\|^2 - (1-\tau)\mathbb{E}\|z^k - z^*\|^2 + (1-\tau)\mathbb{E}\|w^k - z^*\|^2$$
$$- (1 - \tau - 2\gamma^2 L^2)\mathbb{E}\|w^k - z^{k+1/2}\|^2$$
$$+ 6\mathbb{E}\|e^k\|^2 + \frac{6}{M}\sum_{m=1}^{M}\mathbb{E}\left\|e_m^k\right\|^2 - \left(\tau - \frac{1}{2}\right)\mathbb{E}\|z^{k+1/2} - z^k\|^2. \qquad (43)$$

With $\tau \geq \frac{3}{4}$ and rule for $z^{k+1/2}$ (line 7) we obtain

$$\mathbb{E}\|\hat{z}^{k+1} - z^*\|^2 \leq \mathbb{E}\|\hat{z}^k - z^*\|^2 - (1-\tau)\mathbb{E}\|z^k - z^*\|^2 + (1-\tau)\|w^k - z^*\|^2$$
$$- (1 - \tau - 2\gamma^2 L^2)\mathbb{E}\|w^k - z^{k+1/2}\|^2$$
$$+ 6\mathbb{E}\|e^k\|^2 + \frac{6}{M}\sum_{m=1}^{M}\mathbb{E}\left\|e_m^k\right\|^2 - \frac{1}{4}\|z^{k+1/2} - z^k\|^2$$
$$= \mathbb{E}\|\hat{z}^k - z^*\|^2 - (1-\tau)\mathbb{E}\|z^k - z^*\|^2 + (1-\tau)\mathbb{E}\|w^k - z^*\|^2$$
$$- (1 - \tau - 2\gamma^2 L^2)\mathbb{E}\|w^k - z^{k+1/2}\|^2 - \frac{1}{8}\mathbb{E}\|z^{k+1/2} - z^k\|^2$$
$$+ 6\mathbb{E}\|e^k\|^2 + \frac{6}{M}\sum_{m=1}^{M}\mathbb{E}\left\|e_m^k\right\|^2 - \frac{1}{8}\mathbb{E}\|\tau z^k + (1-\tau)w^k - \gamma F(w^k) - z^k\|^2$$
$$= \mathbb{E}\|\hat{z}^k - z^*\|^2 - (1-\tau)\mathbb{E}\|z^k - z^*\|^2 + (1-\tau)\mathbb{E}\|w^k - z^*\|^2$$
$$- (1 - \tau - 2\gamma^2 L^2)\mathbb{E}\|w^k - z^{k+1/2}\|^2 - \frac{1}{8}\mathbb{E}\|z^{k+1/2} - z^k\|^2$$
$$+ 6\mathbb{E}\|e^k\|^2 + \frac{6}{M}\sum_{m=1}^{M}\mathbb{E}\left\|e_m^k\right\|^2 - \frac{1}{8}\mathbb{E}\|(1-\tau)(w^k - z^k) - \gamma F(w^k)\|^2.$$

Using $-\|a\|^2 \leq -\frac{1}{2}\|a+b\|^2 + \|b\|^2$ gives

$$\mathbb{E}\|\hat{z}^{k+1} - z^*\|^2 \leq \mathbb{E}\|\hat{z}^k - z^*\|^2 - (1-\tau)\mathbb{E}\|z^k - z^*\|^2 + (1-\tau)\mathbb{E}\|w^k - z^*\|^2$$
$$- (1 - \tau - 2\gamma^2 L^2)\mathbb{E}\|w^k - z^{k+1/2}\|^2 - \frac{1}{8}\mathbb{E}\|z^{k+1/2} - z^k\|^2$$
$$+ 6\mathbb{E}\|e^k\|^2 + \frac{6}{M}\sum_{m=1}^{M}\mathbb{E}\left\|e_m^k\right\|^2 - \frac{\gamma^2}{16}\mathbb{E}\|F(w^k)\|^2 + \frac{(1-\tau)^2}{8}\mathbb{E}\|w^k - z^k\|^2.$$

And then

$$\mathbb{E}\|\hat{z}^{k+1} - z^*\|^2 \leq \mathbb{E}\|\hat{z}^k - z^*\|^2 - (1-\tau)\mathbb{E}\|z^k - z^*\|^2 + (1-\tau)\mathbb{E}\|w^k - z^*\|^2$$
$$- (1 - \tau - 2\gamma^2 L^2)\mathbb{E}\|w^k - z^{k+1/2}\|^2 - \frac{1}{8}\mathbb{E}\|z^{k+1/2} - z^k\|^2$$
$$+ 6\mathbb{E}\|e^k\|^2 + \frac{6}{M}\sum_{m=1}^{M}\mathbb{E}\left\|e_m^k\right\|^2 - \frac{\gamma^2}{16}\mathbb{E}\|F(w^k)\|^2 + \frac{(1-\tau)^2}{4}\mathbb{E}\|z^{k+1/2} - z^k\|^2$$
$$+ \frac{(1-\tau)^2}{4}\mathbb{E}\|z^{k+1/2} - w^k\|^2$$
$$\leq \mathbb{E}\|\hat{z}^k - z^*\|^2 - (1-\tau)\mathbb{E}\|z^k - z^*\|^2 + (1-\tau)\mathbb{E}\|w^k - z^*\|^2$$
$$- (1 - \tau - 2\gamma^2 L^2)\mathbb{E}\|w^k - z^{k+1/2}\|^2 - \frac{1}{8}\mathbb{E}\|z^{k+1/2} - z^k\|^2$$
$$+ 6\mathbb{E}\|e^k\|^2 + \frac{6}{M}\sum_{m=1}^{M}\mathbb{E}\left\|e_m^k\right\|^2 - \frac{\gamma^2}{16}\mathbb{E}\|F(w^k)\|^2 + \frac{1}{64}\mathbb{E}\|z^{k+1/2} - z^k\|^2$$

$$+ \frac{1-\tau}{16} \mathbb{E}\|z^{k+1/2} - w^k\|^2$$

$$\leq \mathbb{E}\|\hat{z}^k - z^*\|^2 - (1-\tau)\mathbb{E}\|z^k - z^*\|^2 + (1-\tau)\mathbb{E}\|w^k - z^*\|^2$$

$$- \left(\frac{1-\tau}{2} - 2\gamma^2 L^2\right) \mathbb{E}\|w^k - z^{k+1/2}\|^2$$

$$+ 6\mathbb{E}\|e^k\|^2 + \frac{6}{M} \sum_{m=1}^{M} \mathbb{E}\left\|e_m^k\right\|^2 - \frac{\gamma^2}{16}\mathbb{E}\|F(w^k)\|^2.$$

Here we additionally use that $\tau \geq \frac{3}{4}$. Then we add (33)

$$\mathbb{E}\|\hat{z}^{k+1} - z^*\|^2 + \mathbb{E}\|w^{k+1} - z^*\|^2$$

$$\leq \mathbb{E}\|\hat{z}^k - z^*\|^2 + \mathbb{E}\|w^k - z^*\|^2 - \left(\frac{1-\tau}{2} - 2\gamma^2 L^2\right) \mathbb{E}\|w^k - z^{k+1/2}\|^2$$

$$+ 6\mathbb{E}\|e^k\|^2 + \frac{6}{M} \sum_{m=1}^{M} \mathbb{E}\left\|e_m^k\right\|^2 - \frac{\gamma^2}{16}\mathbb{E}\|F(w^k)\|^2.$$

Next, we sum over all $k$ from 0 to $K-1$ and get

$$\frac{\gamma^2}{16} \sum_{k=0}^{K-1} \mathbb{E}\|F(w^k)\|^2 \leq \mathbb{E}\|\hat{z}^0 - z^*\|^2 + \mathbb{E}\|w^0 - z^*\|^2 - \mathbb{E}\|\hat{z}^K - z^*\|^2 - \mathbb{E}\|w^K - z^*\|^2$$

$$- \left(\frac{1-\tau}{2} - 2\gamma^2 L^2\right) \sum_{k=0}^{K-1} \mathbb{E}\|w^k - z^{k+1/2}\|^2$$

$$+ 6 \sum_{k=0}^{K-1} \mathbb{E}\|e^k\|^2 + 6 \sum_{k=0}^{K-1} \frac{1}{M} \sum_{m=1}^{M} \mathbb{E}\left\|e_m^k\right\|^2.$$

It remains to use (35) and (37) with $p=1$:

$$\frac{\gamma^2}{16} \sum_{k=0}^{K-1} \mathbb{E}\|F(w^k)\|^2 \leq \mathbb{E}\|\hat{z}^0 - z^*\|^2 + \mathbb{E}\|w^0 - z^*\|^2 - \mathbb{E}\|\hat{z}^K - z^*\|^2 - \mathbb{E}\|w^K - z^*\|^2$$

$$- \left(\frac{1-\tau}{2} - 2\gamma^2 L^2\right) \sum_{k=0}^{K-1} \mathbb{E}\|w^k - z^{k+1/2}\|^2$$

$$+ 768(\delta^{\mathrm{serv}})^2 \gamma^2 \tilde{L}^2 \sum_{k=0}^{K-1} \mathbb{E}\left\|z^{k+1/2} - w^k\right\|^2$$

$$+ 768(\delta^{\mathrm{serv}})^2 \sum_{k=0}^{K-1} \frac{1}{M} \sum_{m=1}^{M} \mathbb{E}\left\|e_m^k\right\|^2 + 6 \sum_{k=0}^{K-1} \frac{1}{M} \sum_{m=1}^{M} \mathbb{E}\left\|e_m^k\right\|^2$$

$$\leq \mathbb{E}\|\hat{z}^0 - z^*\|^2 + \mathbb{E}\|w^0 - z^*\|^2$$

$$- \left(\frac{1-\tau}{2} - 2\gamma^2 L^2 - 768(\delta^{\mathrm{serv}})^2 \gamma^2 \tilde{L}^2\right) \sum_{k=0}^{K-1} \mathbb{E}\|w^k - z^{k+1/2}\|^2$$

$$+ 775(\delta^{\mathrm{serv}})^2 \sum_{k=0}^{K-1} \frac{1}{M} \sum_{m=1}^{M} \mathbb{E}\left\|e_m^k\right\|^2$$

$$\leq \mathbb{E}\|\hat{z}^0 - z^*\|^2 + \mathbb{E}\|w^0 - z^*\|^2$$

$$- \left(\frac{1-\tau}{2} - 2\gamma^2 L^2 - 768(\delta^{\mathrm{serv}})^2 \gamma^2 \tilde{L}^2\right) \sum_{k=0}^{K-1} \mathbb{E}\|w^k - z^{k+1/2}\|^2$$

$$+ 12400(\delta^{\mathrm{serv}})^2 (\delta^{\mathrm{dev}})^2 \gamma^2 \tilde{L}^2 \sum_{k=0}^{K-1} \mathbb{E}\left\|z^{k+1/2} - w^k\right\|^2$$

$$\leq \mathbb{E}\|\hat{z}^0 - z^*\|^2 + \mathbb{E}\|w^0 - z^*\|^2$$

$$- \frac{1}{2}\left(1 - \tau - 4\gamma^2 L^2 - 26400(\delta^{\text{serv}})^2(\delta^{\text{dev}})^2\gamma^2 L^2\right)\sum_{k=0}^{K-1}\mathbb{E}\|w^k - z^{k+1/2}\|^2.$$

Then we choose $\gamma \leq \frac{\sqrt{1-\tau}}{2L + 165\delta^{\text{serv}}\delta^{\text{dev}}\tilde{L}}$ and get

$$\frac{1}{K}\sum_{k=0}^{K-1}\mathbb{E}\|F(w^k)\|^2 \leq \frac{16(\mathbb{E}\|\hat{z}^0 - z^*\|^2 + \mathbb{E}\|w^0 - z^*\|^2)}{\gamma^2 K}.$$

$\square$

# F  Stochastic case and variance reduction

In this subsection, we assume that the local operators on each node has either a finite-sum form:

$$F_m(z) := \frac{1}{r} \sum_{i=1}^{r} F_{m,i}(z). \tag{44}$$

This case corresponds to the stochastic setting, when it is expensive to calculate the full operator $F_m$, and it is cheaper to calculate the value $F_{m,i}$ one of the terms (batches). For this setup we additionally assume that

**Assumption F.1** *Each operator $F_{m,i}$ is $L_{m,i}$-Lipschitz continuous, i.e. for all $z_1, z_2 \in \mathbb{R}^d$ it holds*

$$\|F_{m,i}(z_1) - F_{m,i}(z_2)\|^2 \le L_{m,i}\|z_1 - z_2\|. \tag{45}$$

*Let $\tilde{L}_m^2 = \frac{1}{r} \sum_{i=1}^{r} L_{m,i}^2$ and $\hat{L}^2 = \frac{1}{M} \sum_{m=1}^{M} \frac{1}{r} \sum_{i=1}^{r} L_{m,i}^2$.*

Next, we modify MASHA1 and MASHA2 for this setup. Modifications of the other steps (computing $g^k$, $z^{k+1}$, $e_m^k$, $e^k$ etc.) in VR-MASHA1 and VR-MASHA2 occur according to the new $g_m^k$.

## F.1  VR-MASHA1: stochastic and batch version

In this section, we provide information about VR-MASHA1. This is a modification of MASHA1 for the stochastic case of a finite sum. Changes compared to MASHA1 are highlighted in blue – see Algorithm 5. Note that without compression VR-MASHA1 is an analogue of methods from [1].

---

**Algorithm 5** VR-MASHA1

1: **Parameters:** Stepsize $\gamma > 0$, parameter $\tau$, number of iterations $K$.
2: **Initialization:** Choose $z^0 = w^0 \in \mathcal{Z}$.
3: Server sends to devices $z^0 = w^0$ and devices compute $F_m(w^0)$ and send to server and get $F(w^0)$
4: **for** $k = 0, 1, 2, \ldots, K-1$ **do**
5:      **for** each device $m$ in parallel **do**
6:          $\bar{z}^k = \tau z^k + (1-\tau)w^k$
7:          $z^{k+1/2} = \bar{z}_k - \gamma F(w^k)$,
8:          Generate $\pi_m^k$ from $\{1, \ldots, r\}$ independently
9:          Compute $F_{m,\pi_m^k}(z^{k+1/2})$ & send $Q_m^{\text{dev}}(F_{m,\pi_m^k}(z^{k+1/2}) - F_{m,\pi_m^k}(w^k))$ to server
10:      **end for**
11:      **for** server **do**
12:          Compute $Q^{\text{serv}}\left[\frac{1}{M}\sum_{m=1}^{M} Q_m^{\text{dev}}(F_{m,\pi_m^k}(z^{k+1/2}) - F_{m,\pi_m^k}(w^k))\right]$ & send to devices
13:          Sends to devices one bit $b_k$: 1 with probability $1-\tau$, 0 with with probability $\tau$
14:      **end for**
15:      **for** each device $m$ in parallel **do**
16:          $z^{k+1} = z^{k+1/2} - \gamma Q^{\text{serv}}\left[\frac{1}{M}\sum_{m=1}^{M} Q_m^{\text{dev}}(F_{m,\pi_m^k}(z^{k+1/2}) - F_{m,\pi_m^k}(w^k))\right]$
17:          **if** $b_k = 1$ **then**
18:              $w^{k+1} = z^{k+1}$
19:              Compute $F_m(w^{k+1})$ & send it to server; and get $F(w^{k+1})$ as a response from server
20:          **else**
21:              $w^{k+1} = w^k$
22:          **end if**
23:      **end for**
24: **end for**

---

The following theorem gives the convergence of VR-MASHA1.

**Theorem F.2** *Let distributed variational inequality* (3) + (4) + (44) *is solved by Algorithm* 5 *with unbiased compressor operators* (1): *on server with* $q^{serv}$ *parameter, on devices with* $\{q_m^{dev}\}$. *Let Assumption* 3.4 *and one case of Assumption* 3.5 *are satisfied. Then the following estimates holds*

- *in strongly-monotone case with* $\gamma \leq \min\left[\frac{\sqrt{1-\tau}}{2\tilde{C}_q}; \frac{1-\tau}{2\mu}\right]$
*(where* $\tilde{C}_q = \sqrt{\frac{q^{serv}}{M^2} \cdot \sum_{m=1}^M (q_m^{dev}\tilde{L}_m^2 + (M-1)\tilde{L}^2)}$*):*

$$\mathbb{E}\left(\|z^K - z^*\|^2 + \|w^K - z^*\|^2\right) \leq \left(1 - \frac{\mu\gamma}{2}\right)^K \cdot 2\|z^0 - z^*\|^2;$$

- *in monotone case with* $\gamma \leq \frac{\sqrt{1-\tau}}{2\tilde{C}_q + 4\tilde{L}}$:

$$\mathbb{E}\left[\max_{z\in\mathcal{C}}\left[\left\langle F(u), \left(\frac{1}{K}\sum_{k=0}^{K-1} z^{k+1/2}\right) - u\right\rangle\right]\right] \leq \frac{2\max_{z\in\mathcal{C}}\left[\|z^0 - z\|^2\right] + 6\|z^0 - z^*\|^2}{\gamma K};$$

- *in non-monotone case with* $\gamma \leq \frac{\sqrt{1-\tau}}{2\tilde{C}_q}$:

$$\mathbb{E}\left(\frac{1}{K}\sum_{k=0}^{K-1}\|F(w^k)\|^2\right) \leq \frac{16\|z^0 - z^*\|^2}{\gamma^2 K}.$$

For VR-MASHA1 we consider the case of only devices compression. For simplicity, we put $Q_m^{dev} = Q$ with $q_m^{dev} = q$ and $\beta_m^{dev} = \beta$, also $\tilde{L}_m = \tilde{L} = L$. Let us discuss the difference (with MASHA1) in choosing $\tau$. When $b_k = 1$, we need not only to send uncompressed information to the server, but also to compute the full $F_m$, which in the stochastic case is $r$ times more expensive than computing one batch $F_{m,i}$. Then, at each iteration, we send $\mathcal{O}\left(\frac{1}{\beta} + 1 - \tau\right)$ bits of information, and also count $\mathcal{O}\left(1 + r(1-\tau)\right)$ batches. Therefore, the optimal choice of $\tau$ depends on two factors and $1 - \tau = \frac{1}{\max\{\beta,r\}}$.

**Corollary F.3** *Let distributed variational inequality* (3) + (4) + (44) *is solved by Algorithm* 5 *without compression on server* ($q^{serv} = 1$) *and with unbiased compressor operators* (1) *on devices with* $\{q_m^{dev} = q\}$. *Let Assumption* 3.4 *and one case of Assumption* 3.5 *are satisfied. Then the following estimates holds*

- *in strongly-monotone case with* $\gamma \leq \min\left[\frac{1}{2L} \cdot \left(\sqrt{\frac{q}{M}} + 1\right)^{-1} \cdot \left(\sqrt{\max\{\beta,r\}}\right)^{-1}; \frac{1}{2\mu}\cdot\left(\max\{\beta,r\}\right)^{-1}\right]$:

$$\mathbb{E}\left(\|z^K - z^*\|^2 + \|w^K - z^*\|^2\right) \leq \left(1 - \frac{\mu\gamma}{2}\right)^K \cdot 2\|z^0 - z^*\|^2;$$

- *in monotone case with* $\gamma \leq \frac{1}{6L} \cdot \left(\sqrt{\frac{q}{M}} + 1\right)^{-1} \cdot \left(\sqrt{\max\{\beta,r\}}\right)^{-1}$:

$$\mathbb{E}\left[\max_{z\in\mathcal{C}}\left[\left\langle F(u), \left(\frac{1}{K}\sum_{k=0}^{K-1} z^{k+1/2}\right) - u\right\rangle\right]\right] \leq \frac{2\max_{z\in\mathcal{C}}\left[\|z^0 - z\|^2\right] + 6\|z^0 - z^*\|^2}{\gamma K};$$

- *in non-monotone case with* $\gamma \leq \frac{1}{2L} \cdot \left(\sqrt{\frac{q}{M}} + 1\right)^{-1} \cdot \left(\sqrt{\max\{\beta,r\}}\right)^{-1}$:

$$\mathbb{E}\left(\frac{1}{K}\sum_{k=0}^{K-1}\|F(w^k)\|^2\right) \leq \frac{16\|z^0 - z^*\|^2}{\gamma^2 K}.$$

In the line 5 of Table 1 we put complexities to achieve $\varepsilon$-solution.

### F.1.1 Proof of the convergence of VR-MASHA1

**Proof of Theorem F.2:**

The proof is very close to the proof of Theorem D.1. Only two estimates need to be modified. First is (15)

$$
\mathbb{E}\left[\|z^{k+1} - z^{k+1/2}\|^2\right] = \gamma^2 \cdot \mathbb{E}\left[\left\|Q^{\mathrm{serv}}\left[\frac{1}{M}\sum_{m=1}^{M} Q_m^{\mathrm{dev}}(F_{m,\pi_m^k}(z^{k+1/2}) - F_{m,\pi_m^k}(w^k))\right]\right\|^2\right]
$$

$$
\leq \gamma^2 \cdot \frac{q^{\mathrm{serv}}}{M^2}\mathbb{E}\left[\left\|\sum_{m=1}^{M} Q_m^{\mathrm{dev}}(F_{m,\pi_m^k}(z^{k+1/2}) - F_{m,\pi_m^k}(w^k))\right\|^2\right]
$$

$$
= \gamma^2 \cdot \frac{q^{\mathrm{serv}}}{M^2}\sum_{m=1}^{M}\mathbb{E}\left[\left\|Q_m^{\mathrm{dev}}(F_{m,\pi_m^k}(z^{k+1/2}) - F_{m,\pi_m^k}(w^k))\right\|^2\right]
$$

$$
+ \gamma^2 \cdot \frac{q^{\mathrm{serv}}}{M^2}\sum_{m\neq l}\mathbb{E}\left[\langle Q_m^{\mathrm{dev}}(F_{m,\pi_m^k}(z^{k+1/2}) - F_{m,\pi_m^k}(w^k)); Q_m^{\mathrm{dev}}(F_l(z^{k+1/2}) - F_{l,\pi_l^k}(w^k))\rangle\right]
$$

Next we apply (1) and Assumption 3.4 for the first term and independence and unbiasedness of $Q$ and uniformess of $\xi$ for the second term:

$$
\mathbb{E}\left[\|z^{k+1} - z^{k+1/2}\|^2\right] \leq \gamma^2 \cdot \frac{q^{\mathrm{serv}}}{M^2}\sum_{m=1}^{M} q_m^{\mathrm{dev}}\frac{1}{r}\sum_{i=1}^{r}L_{m,i}^2\mathbb{E}\left[\left\|z^{k+1/2} - w^k\right\|^2\right]
$$

$$
+ \gamma^2 \cdot \frac{q^{\mathrm{serv}}}{M^2}\sum_{m\neq l}\mathbb{E}\left[\langle F_m(z^{k+1/2}) - F_m(w^k); F_l(z^{k+1/2}) - F_l(w^k)\rangle\right]
$$

$$
\leq \gamma^2 \cdot \frac{q^{\mathrm{serv}}}{M^2}\sum_{m=1}^{M} q_m^{\mathrm{dev}}\tilde{L}_m^2\mathbb{E}\left[\left\|z^{k+1/2} - w^k\right\|^2\right]
$$

$$
+ \gamma^2 \cdot \frac{q^{\mathrm{serv}}}{2M^2}\sum_{m\neq l}\mathbb{E}\left[\|F_m(z^{k+1/2}) - F_m(w^k)\|^2 + \|F_l(z^{k+1/2}) - F_l(w^k)\|^2\right]
$$

$$
\leq \gamma^2 \cdot \frac{q^{\mathrm{serv}}}{M^2}\sum_{m=1}^{M} q_m^{\mathrm{dev}}\tilde{L}_m^2\mathbb{E}\left[\left\|z^{k+1/2} - w^k\right\|^2\right]
$$

$$
+ \gamma^2 \cdot \frac{q^{\mathrm{serv}}}{2M^2}\sum_{m\neq l}\mathbb{E}\left[L_m^2\|z^{k+1/2} - w^k\|^2 + L_l^2\|z^{k+1/2} - w^k\|^2\right]
$$

$$
= \gamma^2 \cdot \frac{q^{\mathrm{serv}}}{M^2}\sum_{m=1}^{M} q_m^{\mathrm{dev}}\tilde{L}_m^2\mathbb{E}\left[\left\|z^{k+1/2} - w^k\right\|^2\right]
$$

$$
+ \gamma^2 \cdot \frac{q^{\mathrm{serv}}(M-1)}{M}\tilde{L}^2\mathbb{E}\left[\|z^{k+1/2} - w^k\|^2\right]
$$

$$
= \gamma^2 \cdot \frac{q^{\mathrm{serv}}}{M^2}\mathbb{E}\left[\|z^{k+1/2} - w^k\|^2\right] \cdot \sum_{m=1}^{M} q_m^{\mathrm{dev}}\tilde{L}_m^2 + (M-1)\tilde{L}^2
$$

Here we can use new $\tilde{C}_q = \sqrt{\frac{q^{\mathrm{serv}}}{M^2} \cdot \sum_{m=1}^{M}(q_m^{\mathrm{dev}}\tilde{L}_m^2 + (M-1)\tilde{L}^2)}$.

The second modified estimate is (25):

$$
\mathbb{E}\left[\left\|Q^{\mathrm{serv}}\left[\frac{1}{M}\sum_{m=1}^{M} Q_m^{\mathrm{dev}}(F_m(z^{k+1/2}) - F_m(w^k))\right] + F(w^k) - F(z^{k+1/2})\right\|^2\right]
$$

$$
\leq \tilde{C}_q^2\mathbb{E}\left[\left\|z^{k+1/2} - w^k\right\|^2\right] + 2\tilde{L}^2\mathbb{E}\left[\left\|z^{k+1/2} - w^k\right\|^2\right].
$$

$\square$

### F.2  VR-MASHA2: stochastic and batch version

In this section, we provide information about VR-MASHA2. This is a modification of MASHA2 for the stochastic case of a finite sum. Changes compared to MASHA2 are highlighted in blue – see Algorithm 6.

---

**Algorithm 6** VR-MASHA2

1: **Parameters:** Stepsize $\gamma > 0$, parameter $\tau$, number of iterations $K$.
2: **Initialization:** Choose $z^0 = w^0 \in \mathcal{Z}$, $e_m^0 = 0$, $e^0 = 0$.
3: Server sends to devices $z^0 = w^0$ and devices compute $F_m(w^0)$ and send to server and get $F(w^0)$
4: **for** $k = 0, 1, 2, \ldots, K-1$ **do**
5:     **for** each device $m$ in parallel **do**
6:         $\bar{z}^k = \tau z^k + (1-\tau)w^k$
7:         $z^{k+1/2} = \bar{z}_k - \gamma F(w^k)$
8:         Generate $\pi_m^k$ from $\{1, \ldots, r\}$ independently
9:         Compute $F_{m,\pi_m^k}(z^{k+1/2})$ and send $C_m^{\text{dev}}(\gamma F_{m,\pi_m^k}(z^{k+1/2}) - \gamma F_{m,\pi_m^k}(w^k) + e_m^k)$
10:         $e_m^{k+1} = e_m^k + \gamma F_{m,\pi_m^k}(z^{k+1/2}) - \gamma F_{m,\pi_m^k}(w^k) - C_m^{\text{dev}}(\gamma F_{m,\pi_m^k}(z^{k+1/2}) - \gamma F_{m,\pi_m^k}(w^k) + e_m^k)$
11:     **end for**
12:     **for** server **do**
13:         Compute $g^k = C^{\text{serv}}\left[\frac{1}{M}\sum_{m=1}^{M} C_m^{\text{dev}}(\gamma F_{m,\pi_m^k}(z^{k+1/2}) - \gamma F_{m,\pi_m^k}(w^k) + e_m^k) + e^k\right]$ &
    send to devices
14:         $e^{k+1} = e^k + \frac{1}{M}\sum_{m=1}^{M} C_m^{\text{dev}}(\gamma F_{m,\pi_m^k}(z^{k+1/2}) - \gamma F_{m,\pi_m^k}(w^k) + e_m^k) - g^k$
15:         Sends to devices one bit $b_k$: 1 with probability $1-\tau$, 0 with with probability $\tau$
16:     **end for**
17:     **for** each device $m$ in parallel **do**
18:         $z^{k+1} = z^{k+1/2} - C^{\text{serv}}\left[\frac{1}{M}\sum_{m=1}^{M} C_m^{\text{dev}}(\gamma F_{m,\pi_m^k}(z^{k+1/2}) - \gamma F_{m,\pi_m^k}(w^k) + e_m^k) + e^k\right]$
19:         **if** $b_k = 1$ **then**
20:             $w^{k+1} = z^k$
21:             Compute $F_m(w^{k+1})$ and it send to server; and get $F(w^{k+1})$
22:         **else**
23:             $w^{k+1} = w^k$
24:         **end if**
25:     **end for**
26: **end for**

---

The following theorem gives the convergence of VR-MASHA2.

**Theorem F.4** *Let distributed variational inequality* (3) + (4) + (44) *is solved by Algorithm 6 with* $\tau \geq \frac{3}{4}$ *and biased compressor operators* (2): *on server with* $\delta^{serv}$ *parameter, on devices with* $\delta^{dev}$. *Let Assumption 3.4 and one case of Assumption 3.5 are satisfied. Then the following estimates holds*

- *in strongly-monotone case with* $\gamma \leq \min\left[\frac{1-\tau}{8\mu}; \frac{\sqrt{1-\tau}}{2L + 165\delta^{serv}\delta^{dev}\hat{L}}\right]$:

$$\mathbb{E}\left(\|\hat{z}^K - z^*\|^2 + \|w^K - z^*\|^2\right) \leq \left(1 - \frac{\mu\gamma}{2}\right)^K \cdot 2\|z^0 - z^*\|^2;$$

- *in monotone case with* $\gamma \leq \frac{\sqrt{1-\tau}}{2L + 165\delta^{serv}\delta^{dev}\hat{L}}$:

$$\mathbb{E}\left[\max_{z\in\mathcal{C}}\langle F(z), \left(\frac{1}{K}\sum_{k=0}^{K-1} z^{k+1/2}\right) - z\rangle\right] \leq \frac{2\max_{z\in\mathcal{C}}\|z^0 - z\|^2 + 6\|z^0 - z^*\|^2}{\gamma K};$$

• *in non-monotone case with $\gamma \leq \frac{\sqrt{1-\tau}}{2L+165\delta^{serv}\delta^{dev}\hat{L}}$:*

$$\frac{1}{K}\sum_{k=0}^{K-1}\mathbb{E}\|F(w^k)\|^2 \leq \frac{32\mathbb{E}\|z^0-z^*\|^2}{\gamma^2 K}.$$

We consider the only devices compression. For simplicity, we put $\tilde{L} = \hat{L} = L$. We use the same reasoning as in Section F.1. The optimal choice is $1 - \tau = \frac{1}{\max\{\beta,r\}}$.

**Corollary F.5** *Let distributed variational inequality (3) + (4)+(44) is solved by Algorithm 6 without compression on server ($\delta^{serv} = 1$) and with biased compressor operators (2) on devices with $\delta^{dev} = \delta$. Let Assumption 3.4 and one case of Assumption 3.5 are satisfied. Then the following estimates holds*

• *in strongly-monotone case with $\gamma \leq \min\left[\frac{1}{8\mu}\cdot(\max\{\beta,r\})^{-1}; \frac{1}{167\delta L}\cdot\left(\sqrt{\max\{\beta,r\}}\right)^{-1}\right]$:*

$$\mathbb{E}\left(\|\hat{z}^K - z^*\|^2 + \|w^K - z^*\|^2\right) \leq \left(1-\frac{\mu\gamma}{2}\right)^K \cdot 2\|z^0 - z^*\|^2;$$

• *in monotone case with $\gamma \leq \frac{1}{167\delta L}\cdot\left(\sqrt{\max\{\beta,r\}}\right)^{-1}$:*

$$\mathbb{E}\left[\max_{z\in\mathcal{C}}\left\langle F(z),\left(\frac{1}{K}\sum_{k=0}^{K-1}z^{k+1/2}\right)-z\right\rangle\right] \leq \frac{2\max_{z\in\mathcal{C}}\|z^0-z\|^2 + 4\|z^0-z^*\|^2}{\gamma K};$$

• *in non-monotone case with $\gamma \leq \frac{1}{167\delta L}\cdot\left(\sqrt{\max\{\beta,r\}}\right)^{-1}$:*

$$\mathbb{E}\left(\frac{1}{K}\sum_{k=0}^{K-1}\|F(w^k)\|^2\right) \leq \frac{32\mathbb{E}\|z^0-z^*\|^2}{\gamma^2 K}.$$

In the line 6 of Table 1 we put complexities to achieve $\varepsilon$-solution.

### F.2.1 Proof of the convergence of VR-MASHA2

**Proof of Theorem F.4:** The proofs of Theorem F.4 partially repeat the proofs of Theorem E.1. We note the main changes in comparison with Theorem E.1.

The first difference is an update of "hat" sequence (26):

$$\hat{z}^{k+1} = z^{k+1} - e^{k+1} - \frac{1}{M}\sum_{m=1}^{M}e_m^{k+1}$$

$$= z^{k+1/2} - C^{serv}\left[\frac{1}{M}\sum_{m=1}^{M}C_m^{dev}(\gamma F_{m,\pi_m^k}(z^{k+1/2}) - \gamma F_{m,\pi_m^k}(w^k) + e_m^k) + e^k\right]$$

$$- e^k - \frac{1}{M}\sum_{m=1}^{M}C_m^{dev}(\gamma F_{m,\pi_m^k}(z^{k+1/2}) - \gamma F_{m,\pi_m^k}(w^k) + e_m^k)$$

$$+ C^{serv}\left[\frac{1}{M}\sum_{m=1}^{M}C_m^{dev}(\gamma F_{m,\pi_m^k}(z^{k+1/2}) - \gamma F_{m,\pi_m^k}(w^k) + e_m^k) + e^k\right]$$

$$- \frac{1}{M}\sum_{m=1}^{M}\left[e_m^k + \gamma F_{m,\pi_m^k}(z^{k+1/2}) - \gamma F_{m,\pi_m^k}(w^k) - C_m^{dev}(\gamma \cdot F_{m,\pi_m^k}(z^{k+1/2}) - \gamma F_{m,\pi_m^k}(w^k) + e_m^k)\right]$$

$$= z^{k+1/2} - e^k - \frac{1}{M}\sum_{m=1}^{M}e_m^k - \gamma \cdot \frac{1}{M}\sum_{m=1}^{M}(F_{m,\pi_m^k}(z^{k+1/2}) - \gamma F_{m,\pi_m^k}(w^k))$$

$$= \hat{z}^{k+1/2} - \gamma \cdot \frac{1}{M} \sum_{m=1}^{M} (F_{m,\pi_m^k}(z^{k+1/2}) - \gamma F_{m,\pi_m^k}(w^k)).$$

Hence, we need to modify (29)

$$\|\hat{z}^{k+1} - z\|^2 \le \|\hat{z}^k - z\|^2 + 2\langle \hat{z}^{k+1} - \hat{z}^k, z^{k+1/2} - z \rangle$$

$$+ 2\gamma^2 \cdot \left\| \frac{1}{M} \sum_{m=1}^{M} (F_{m,\pi_m^k}(z^{k+1/2}) - F_{m,\pi_m^k}(w^k)) \right\|^2 + 4\|e^k\|^2 + \frac{4}{M} \sum_{m=1}^{M} \left\| e_m^k \right\|^2$$

$$- \|z^{k+1/2} - \hat{z}^k\|^2;$$

and (30):

$$\hat{z}^{k+1} - \hat{z}^k = \hat{z}^{k+1} - \hat{z}^{k+1/2} + \hat{z}^{k+1/2} - \hat{z}^k$$

$$= -\gamma \cdot \left( \frac{1}{M} \sum_{m=1}^{M} (F_{m,\pi_m^k}(z^{k+1/2}) - F_{m,\pi_m^k}(w^k)) \right) + z^{k+1/2} - z^k$$

$$= -\gamma \cdot \left( \frac{1}{M} \sum_{m=1}^{M} (F_{m,\pi_m^k}(z^{k+1/2}) - F_{m,\pi_m^k}(w^k)) \right) - \gamma \cdot F(w^k) + \bar{z}^k - z^k,$$

Then (31) is also modified:

$$\|\hat{z}^{k+1} - z\|^2 \le \|\hat{z}^k - z\|^2 - (1-\tau)\|z^k - z\|^2 + (1-\tau)\|w^k - z\|^2$$

$$- 2\gamma \left\langle \left( \frac{1}{M} \sum_{m=1}^{M} (F_{m,\pi_m^k}(z^{k+1/2}) - F_{m,\pi_m^k}(w^k)) \right) + F(w^k), z^{k+1/2} - z \right\rangle$$

$$- (1-\tau)\|w^k - z^{k+1/2}\|^2 + 2\gamma^2 \cdot \left\| \frac{1}{M} \sum_{m=1}^{M} (F_{m,\pi_m^k}(z^{k+1/2}) - F_{m,\pi_m^k}(w^k)) \right\|^2$$

$$+ 6\|e^k\|^2 + \frac{6}{M} \sum_{m=1}^{M} \left\| e_m^k \right\|^2 - \left( \tau - \frac{1}{2} \right) \|z^{k+1/2} - z^k\|^2. \tag{46}$$

Next, we move to different cases of monotonicity.

**Strongly-monotone**

The same way as in Theorem E.1 we put $z = z^*$, use property of the solution and then take full expectation:

$$\mathbb{E}\|\hat{z}^{k+1} - z^*\|^2 \le \mathbb{E}\|\hat{z}^k - z^*\|^2 - (1-\tau)\mathbb{E}\|z^k - z^*\|^2 + (1-\tau)\mathbb{E}\|w^k - z^*\|^2$$

$$- 2\gamma\mathbb{E}\left[ \left\langle \left( \frac{1}{M} \sum_{m=1}^{M} (F_{m,\pi_m^k}(z^{k+1/2}) - F_{m,\pi_m^k}(w^k)) \right) + F(w^k) - F(z^*), z^{k+1/2} - z^* \right\rangle \right]$$

$$- (1-\tau)\mathbb{E}\|w^k - z^{k+1/2}\|^2 + 2\gamma^2 \cdot \mathbb{E}\left\| \frac{1}{M} \sum_{m=1}^{M} (F_{m,\pi_m^k}(z^{k+1/2}) - F_{m,\pi_m^k}(w^k)) \right\|^2$$

$$+ 6\mathbb{E}\|e^k\|^2 + \frac{6}{M} \sum_{m=1}^{M} \mathbb{E}\left\| e_m^k \right\|^2 - \left( \tau - \frac{1}{2} \right) \mathbb{E}\|z^{k+1/2} - z^k\|^2$$

$$\le \mathbb{E}\|\hat{z}^k - z^*\|^2 - (1-\tau)\mathbb{E}\|z^k - z^*\|^2 + (1-\tau)\mathbb{E}\|w^k - z^*\|^2$$

$$- 2\gamma\mathbb{E}\left[ \left\langle \mathbb{E}_{\pi^k}\left[ \frac{1}{M} \sum_{m=1}^{M} (F_{m,\pi_m^k}(z^{k+1/2}) - F_{m,\pi_m^k}(w^k)) + F(w^k) - F(z^*) \right], z^{k+1/2} - z^* \right\rangle \right]$$

$$- (1-\tau)\mathbb{E}\|w^k - z^{k+1/2}\|^2 + 2\gamma^2 \cdot \frac{1}{M} \sum_{m=1}^{M} \mathbb{E}\left[ \mathbb{E}_{\pi^k}\left\| F_{m,\pi_m^k}(z^{k+1/2}) - F_{m,\pi_m^k}(w^k) \right\|^2 \right]$$

$$+ 6\mathbb{E}\|e^k\|^2 + \frac{6}{M} \sum_{m=1}^{M} \mathbb{E} \left\|e_m^k\right\|^2 - \left(\tau - \frac{1}{2}\right) \mathbb{E}\|z^{k+1/2} - z^k\|^2$$

$$= \mathbb{E}\|\hat{z}^k - z^*\|^2 - (1-\tau)\mathbb{E}\|z^k - z^*\|^2 + (1-\tau)\mathbb{E}\|w^k - z^*\|^2$$
$$- 2\gamma\mathbb{E}\left[\langle F(z^{k+1/2} - F(z^*), z^{k+1/2} - z^*\rangle\right]$$
$$- (1-\tau)\mathbb{E}\|w^k - z^{k+1/2}\|^2 + 2\gamma^2 \cdot \frac{1}{M} \sum_{m=1}^{M} \mathbb{E}\left[\frac{1}{r} \sum_{i=1}^{r} \left\|F_{m,i}(z^{k+1/2}) - F_{m,i}(w^k)\right\|^2\right]$$
$$+ 6\mathbb{E}\|e^k\|^2 + \frac{6}{M} \sum_{m=1}^{M} \mathbb{E} \left\|e_m^k\right\|^2 - \left(\tau - \frac{1}{2}\right) \mathbb{E}\|z^{k+1/2} - z^k\|^2$$

$$= \mathbb{E}\|\hat{z}^k - z^*\|^2 - (1-\tau)\mathbb{E}\|z^k - z^*\|^2 + (1-\tau)\mathbb{E}\|w^k - z^*\|^2$$
$$- 2\gamma\mathbb{E}\left[\langle F(z^{k+1/2} - F(z^*), z^{k+1/2} - z^*\rangle\right]$$
$$- (1-\tau)\mathbb{E}\|w^k - z^{k+1/2}\|^2 + 2\gamma^2\hat{L}^2\mathbb{E}\|w^k - z^{k+1/2}\|^2$$
$$+ 6\mathbb{E}\|e^k\|^2 + \frac{6}{M} \sum_{m=1}^{M} \mathbb{E} \left\|e_m^k\right\|^2 - \left(\tau - \frac{1}{2}\right) \mathbb{E}\|z^{k+1/2} - z^k\|^2. \tag{47}$$

In the last we use Assumption 3.4 and definition of $\hat{L}$ from this Assumption. The new inequality (47) is absolutely similar to inequality (32) (only $L$ is changed to $\hat{L}$). Therefore, we can safely reach the analogue of expression (34):

$$\sum_{k=0}^{K-1} p^k \mathbb{E}\|\hat{z}^{k+1} - z^*\|^2 + \sum_{k=0}^{K-1} p^k \mathbb{E}\|w^{k+1} - z^*\|^2$$

$$\leq \sum_{k=0}^{K-1} p^k \mathbb{E}\|\hat{z}^k - z^*\|^2 + \sum_{k=0}^{K-1} p^k \mathbb{E}\|w^k - z^*\|^2 - 2\gamma\mu \sum_{k=0}^{K-1} p^k \mathbb{E}\|z^{k+1/2} - z^*\|^2$$
$$- (1 - \tau - 2\gamma^2\hat{L}^2) \cdot \sum_{k=0}^{K-1} p^k \mathbb{E}\|w^k - z^{k+1/2}\|^2 - \left(\tau - \frac{1}{2}\right) \cdot \sum_{k=0}^{K-1} p^k \mathbb{E}\|z^{k+1/2} - z^k\|^2$$
$$+ 6 \cdot \sum_{k=0}^{K-1} p^k \mathbb{E}\|e^k\|^2 + 6 \cdot \sum_{k=0}^{K-1} p^k \frac{1}{M} \sum_{m=1}^{M} \mathbb{E} \left\|e_m^k\right\|^2. \tag{48}$$

The only difference in the estimates on "errors" $e^k$ and $e_m^k$ is in the constant $\tilde{L}$. It needs to be changed to $\hat{L}$. And we have analogue of (38):

$$\sum_{k=0}^{K-1} p^k \left(\mathbb{E}\|\hat{z}^{k+1} - z^*\|^2 + \mathbb{E}\|w^{k+1} - z^*\|^2\right)$$

$$\leq \sum_{k=0}^{K-1} p^k \left(1 - \frac{\mu\gamma}{2}\right) \left(\mathbb{E}\|\hat{z}^k - z^*\|^2 + \mathbb{E}\|w^k - z^*\|^2\right)$$
$$- (1 - \tau - \gamma\mu - 13200(\delta^{\text{serv}})^2(\delta^{\text{dev}})^2\gamma^2\hat{L}^2) \cdot \sum_{k=0}^{K-1} p^k \mathbb{E}\|w^k - z^{k+1/2}\|^2$$
$$- \left(\tau - 2\gamma\mu - \frac{1}{2}\right) \cdot \sum_{k=0}^{K-1} p^k \mathbb{E}\|z^{k+1/2} - z^k\|^2. \tag{49}$$

Choice $\tau \geq \frac{3}{4}$, $\gamma \leq \min\left[\frac{1-\tau}{8\mu}; \frac{\sqrt{1-\tau}}{165\delta^{\text{serv}}\delta^{\text{dev}}\hat{L}}\right]$ finishes the proof.

$\square$

**Monotone case**

We start from (46) with small rearrangements:

$$
\begin{aligned}
2\gamma\langle F(z^{k+1/2}), & z^{k+1/2} - z\rangle \\
\leq\; & \|\hat{z}^k - z\|^2 - \|\hat{z}^{k+1} - z\|^2 + \|w^k - z\|^2 - \|w^{k+1} - z\|^2 \\
& + \|w^{k+1} - z\|^2 - (1-\tau)\|z^k - z\|^2 - \tau\|w^k - z\|^2 \\
& - 2\gamma\langle\left(\frac{1}{M}\sum_{m=1}^{M}(F_{m,\pi_m^k}(z^{k+1/2}) - F_{m,\pi_m^k}(w^k))\right) + F(w^k) - F(z^{k+1/2}), z^{k+1/2} - z\rangle \\
& - (1-\tau)\|w^k - z^{k+1/2}\|^2 + 2\gamma^2\cdot\left\|\frac{1}{M}\sum_{m=1}^{M}(F_{m,\pi_m^k}(z^{k+1/2}) - F_{m,\pi_m^k}(w^k))\right\|^2 \\
& + 6\|e^k\|^2 + \frac{6}{M}\sum_{m=1}^{M}\|e_m^k\|^2 - \left(\tau - \frac{1}{2}\right)\|z^{k+1/2} - z^k\|^2.
\end{aligned}
$$

The same way as in Theorem E.1 we use monotonicity (Assumption 3.5 (M)) and then sum from 0 to $K-1$:

$$
\begin{aligned}
2\gamma\sum_{k=0}^{K-1}\langle F(z), & z^{k+1/2} - z\rangle \\
\leq\; & \|\hat{z}^0 - z\|^2 + \|w^0 - z\|^2 \\
& + \sum_{k=0}^{K-1}\left(\|w^{k+1} - z\|^2 - (1-\tau)\|z^k - z\|^2 - \tau\|w^k - z\|^2\right) \\
& - 2\gamma\cdot\sum_{k=0}^{K-1}\langle\left(\frac{1}{M}\sum_{m=1}^{M}(F_{m,\pi_m^k}(z^{k+1/2}) - F_{m,\pi_m^k}(w^k))\right) + F(w^k) - F(z^{k+1/2}), z^{k+1/2} - z\rangle \\
& - (1-\tau)\cdot\sum_{k=0}^{K-1}\|w^k - z^{k+1/2}\|^2 + 2\gamma^2\cdot\sum_{k=0}^{K-1}\left\|\frac{1}{M}\sum_{m=1}^{M}(F_{m,\pi_m^k}(z^{k+1/2}) - F_{m,\pi_m^k}(w^k))\right\|^2 \\
& + 6\cdot\sum_{k=0}^{K-1}\|e^k\|^2 + 6\cdot\sum_{k=0}^{K}\frac{1}{M}\sum_{m=1}^{M}\|e_m^k\|^2 - \left(\tau - \frac{1}{2}\right)\cdot\sum_{k=0}^{K-1}\|z^{k+1/2} - z^k\|^2.
\end{aligned}
$$

Then we take maximum of both sides over $z \in \mathcal{C}$, after take expectation and get

$$
\begin{aligned}
2\gamma\mathbb{E}\left[\max_{z\in\mathcal{C}}\sum_{k=0}^{K-1}\langle F(z), z^{k+1/2} - z\rangle\right] \\
\leq\; & \mathbb{E}\left[\max_{z\in\mathcal{C}}\|\hat{z}^0 - z\|^2\right] + \mathbb{E}\left[\max_{z\in\mathcal{C}}\|w^0 - z\|^2\right] \\
& + \mathbb{E}\left[\max_{z\in\mathcal{C}}\sum_{k=0}^{K-1}\left(\|w^{k+1} - z\|^2 - (1-\tau)\|z^k - z\|^2 - \tau\|w^k - z\|^2\right)\right] \\
& + 2\gamma\mathbb{E}\left[\max_{z\in\mathcal{C}}\sum_{k=0}^{K-1}-\langle\left(\frac{1}{M}\sum_{m=1}^{M}(F_{m,\pi_m^k}(z^{k+1/2}) - F_{m,\pi_m^k}(w^k))\right) + F(w^k) - F(z^{k+1/2}), z^{k+1/2} - z\rangle\right] \\
& - (1-\tau)\cdot\sum_{k=0}^{K-1}\mathbb{E}\|w^k - z^{k+1/2}\|^2 + 2\gamma^2\cdot\sum_{k=0}^{K-1}\mathbb{E}\left\|\frac{1}{M}\sum_{m=1}^{M}(F_{m,\pi_m^k}(z^{k+1/2}) - F_{m,\pi_m^k}(w^k))\right\|^2 \\
& + 6\cdot\sum_{k=0}^{K-1}\mathbb{E}\|e^k\|^2 + 6\cdot\sum_{k=0}^{K}\frac{1}{M}\sum_{m=1}^{M}\mathbb{E}\|e_m^k\|^2 - \left(\tau - \frac{1}{2}\right)\cdot\sum_{k=0}^{K-1}\mathbb{E}\|z^{k+1/2} - z^k\|^2.
\end{aligned}
$$

The same way as in strongly-monotone case of this Theorem (Theorem F.4) we estimate

$$2\gamma^2 \sum_{k=0}^{K-1} \mathbb{E}\left\|\frac{1}{M}\sum_{m=1}^{M}(F_{m,\pi_m^k}(z^{k+1/2}) - F_{m,\pi_m^k}(w^k))\right\|^2 + 6\sum_{k=0}^{K-1}\mathbb{E}\|e^k\|^2 + 6\sum_{k=0}^{K}\frac{1}{M}\sum_{m=1}^{M}\mathbb{E}\left\|e_m^k\right\|^2:$$

$$2\gamma\mathbb{E}\left[\max_{z\in\mathcal{C}}\sum_{k=0}^{K-1}\langle F(z), z^{k+1/2} - z\rangle\right]$$

$$\leq \mathbb{E}\left[\max_{z\in\mathcal{C}}\|\hat{z}^0 - z\|^2\right] + \mathbb{E}\left[\max_{z\in\mathcal{C}}\|w^0 - z\|^2\right]$$

$$+ \mathbb{E}\left[\max_{z\in\mathcal{C}}\sum_{k=0}^{K-1}\left(\|w^{k+1} - z\|^2 - (1-\tau)\|z^k - z\|^2 - \tau\|w^k - z\|^2\right)\right]$$

$$+ 2\gamma\mathbb{E}\left[\max_{z\in\mathcal{C}}\sum_{k=0}^{K-1} -\left\langle\left(\frac{1}{M}\sum_{m=1}^{M}(F_{m,\pi_m^k}(z^{k+1/2}) - F_{m,\pi_m^k}(w^k))\right) + F(w^k) - F(z^{k+1/2}), z^{k+1/2} - z\right\rangle\right]$$

$$- (1 - \tau - 13200(\delta^{\text{serv}})^2(\delta^{\text{dev}})^2\gamma^2\hat{L}^2) \cdot \sum_{k=0}^{K-1}\mathbb{E}\|w^k - z^{k+1/2}\|^2$$

$$- \left(\tau - \frac{1}{2}\right) \cdot \sum_{k=0}^{K-1}\mathbb{E}\|z^{k+1/2} - z^k\|^2.$$

With $t \geq \frac{3}{4}$ and $\gamma \leq \frac{\sqrt{1-\tau}}{165\delta^{\text{serv}}\delta^{\text{dev}}\hat{L}}$ we get

$$2\gamma\mathbb{E}\left[\max_{z\in\mathcal{C}}\sum_{k=0}^{K-1}\langle F(z), z^{k+1/2} - z\rangle\right]$$

$$\leq \mathbb{E}\left[\max_{z\in\mathcal{C}}\|\hat{z}^0 - z\|^2\right] + \mathbb{E}\left[\max_{z\in\mathcal{C}}\|w^0 - z\|^2\right]$$

$$+ \mathbb{E}\left[\max_{z\in\mathcal{C}}\sum_{k=0}^{K-1}\left(\|w^{k+1} - z\|^2 - (1-\tau)\|z^k - z\|^2 - \tau\|w^k - z\|^2\right)\right]$$

$$+ 2\gamma\mathbb{E}\left[\max_{z\in\mathcal{C}}\sum_{k=0}^{K-1} -\left\langle\left(\frac{1}{M}\sum_{m=1}^{M}(F_{m,\pi_m^k}(z^{k+1/2}) - F_{m,\pi_m^k}(w^k))\right) + F(w^k) - F(z^{k+1/2}), z^{k+1/2} - z\right\rangle\right].$$

Using (41), we obtain

$$2\gamma\mathbb{E}\left[\max_{z\in\mathcal{C}}\sum_{k=0}^{K-1}\langle F(z), z^{k+1/2} - z\rangle\right]$$

$$\leq \mathbb{E}\left[\max_{z\in\mathcal{C}}\|\hat{z}^0 - z\|^2\right] + \mathbb{E}\left[\max_{z\in\mathcal{C}}\|w^0 - z\|^2\right] + \mathbb{E}\left[\max_{z\in\mathcal{C}}\|z^0 - z\|^2\right]$$

$$+ (1 - \tau) \cdot \sum_{k=0}^{K-1}\mathbb{E}\left[\|z^k - w^k\|^2\right] \tag{50}$$

$$+ 2\gamma\mathbb{E}\left[\max_{z\in\mathcal{C}}\sum_{k=0}^{K-1} -\left\langle\left(\frac{1}{M}\sum_{m=1}^{M}(F_{m,\pi_m^k}(z^{k+1/2}) - F_{m,\pi_m^k}(w^k))\right) + F(w^k) - F(z^{k+1/2}), z^{k+1/2} - z\right\rangle\right].$$

Let us work with the last line. For this define sequence $v$: $v^0 = z^0$, $v^{k+1} = v^k - \gamma\delta_k$ with $\delta^k = F(z^{k+1/2}) - \left(\frac{1}{M}\sum_{m=1}^{M}(F_{m,\pi_m^k}(z^{k+1/2}) - F_{m,\pi_m^k}(w^k))\right) + F(w^k)$. Then we have

$$\sum_{k=0}^{K-1}\langle\delta^k, z^{k+1/2} - z\rangle = \sum_{k=0}^{K-1}\langle\delta^k, z^{k+1/2} - v^k\rangle + \sum_{k=0}^{K-1}\langle\delta^k, v^k - z\rangle. \tag{51}$$

By the definition of $v^{k+1}$ we get

$$\langle \gamma \delta^k, v^k - z \rangle = \langle \gamma \delta^k, v^k - v^{k+1} \rangle + \langle v^{k+1} - v^k, z - v^{k+1} \rangle$$

$$= \langle \gamma \delta^k, v^k - v^{k+1} \rangle + \frac{1}{2}\|v^k - z\|^2 - \frac{1}{2}\|v^{k+1} - z\|^2 - \frac{1}{2}\|v^k - v^{k+1}\|^2$$

$$= \frac{\gamma^2}{2}\|\delta^k\|^2 + \frac{1}{2}\|v^k - v^{k+1}\|^2 + \frac{1}{2}\|v^k - z\|^2 - \frac{1}{2}\|v^{k+1} - z\|^2 - \frac{1}{2}\|v^k - v^{k+1}\|^2$$

$$= \frac{\gamma^2}{2}\|\delta^k\|^2 + \frac{1}{2}\|v^k - z\|^2 - \frac{1}{2}\|v^{k+1} - z\|^2.$$

With (51) it gives

$$\sum_{k=0}^{K-1} \langle \delta^k, z^{k+1/2} - z \rangle \le \sum_{k=0}^{K-1} \langle \delta^k, z^{k+1/2} - v^k \rangle + \frac{1}{\gamma} \sum_{k=0}^{K-1} \left( \frac{\gamma^2}{2}\|\delta^k\|^2 + \frac{1}{2}\|v^k - z\|^2 - \frac{1}{2}\|v^{k+1} - z\|^2 \right)$$

$$\le \sum_{k=0}^{K-1} \langle \delta^k, z^{k+1/2} - v^k \rangle + \frac{\gamma}{2} \sum_{k=0}^{K-1} \|\delta^k\|^2 + \frac{1}{2\gamma}\|v^0 - z\|^2.$$

We take the maximum on $z$ and get

$$\max_{z \in \mathcal{C}} \sum_{k=0}^{K-1} \langle \delta^k, z^{k+1/2} - z \rangle \le \sum_{k=0}^{K-1} \langle \delta^k, z^{k+1/2} - v^k \rangle + \frac{1}{2\gamma} \max_{z \in \mathcal{C}} \|z^0 - z\|^2$$

$$+ \frac{\gamma}{2} \sum_{k=0}^{K-1} \left\| F(z^{k+1/2}) - \left( \frac{1}{M} \sum_{m=1}^{M} (F_{m,\pi_m^k}(z^{k+1/2}) - F_{m,\pi_m^k}(w^k)) \right) - F(w^k) \right\|^2.$$

Taking the full expectation, we get

$$\mathbb{E}\left[ \max_{z \in \mathcal{C}} \sum_{k=0}^{K-1} \langle \delta^k, z^{k+1/2} - z \rangle \right] \le \mathbb{E}\left[ \sum_{k=0}^{K-1} \langle \delta^k, z^{k+1/2} - v^k \rangle \right]$$

$$+ \frac{\gamma}{2} \sum_{k=0}^{K-1} \mathbb{E}\left[ \left\| F(z^{k+1/2}) - \left( \frac{1}{M} \sum_{m=1}^{M} (F_{m,\pi_m^k}(z^{k+1/2}) - F_{m,\pi_m^k}(w^k)) \right) - F(w^k) \right\|^2 \right]$$

$$+ \frac{1}{2\gamma} \max_{z \in \mathcal{C}} \|v^0 - z\|^2$$

$$= \mathbb{E}\left[ \sum_{k=0}^{K-1} \langle \mathbb{E}_{\pi^k}\left[ F(z^{k+1/2}) - \left( \frac{1}{M} \sum_{m=1}^{M} (F_{m,\pi_m^k}(z^{k+1/2}) - F_{m,\pi_m^k}(w^k)) \right) - F(w^k) \right], z^{k+1/2} - v^k \rangle \right]$$

$$+ \frac{\gamma}{2} \sum_{k=0}^{K-1} \mathbb{E}\left[ \left\| F(z^{k+1/2}) - \left( \frac{1}{M} \sum_{m=1}^{M} (F_{m,\pi_m^k}(z^{k+1/2}) - F_{m,\pi_m^k}(w^k)) \right) - F(w^k) \right\|^2 \right]$$

$$+ \frac{1}{2\gamma} \max_{z \in \mathcal{C}} \|z^0 - z\|^2$$

$$= \frac{\gamma}{2} \sum_{k=0}^{K-1} \mathbb{E}\left[ \left\| F(z^{k+1/2}) - \left( \frac{1}{M} \sum_{m=1}^{M} (F_{m,\pi_m^k}(z^{k+1/2}) - F_{m,\pi_m^k}(w^k)) \right) - F(w^k) \right\|^2 \right]$$

$$+ \frac{1}{2\gamma} \max_{z \in \mathcal{C}} \|z^0 - z\|^2$$

$$\le \gamma \sum_{k=0}^{K-1} \mathbb{E}\left[ \frac{1}{M} \sum_{m=1}^{M} \|F_{m,\pi_m^k}(z^{k+1/2}) - F_{m,\pi_m^k}(w^k)\|^2 \right] + \gamma \sum_{k=0}^{K-1} \mathbb{E}\left[ \|F(z^{k+1/2}) - F(w^k)\|^2 \right]$$

$$+ \frac{1}{2\gamma} \max_{z \in \mathcal{C}} \|z^0 - z\|^2$$

$$\le \gamma(\hat{L}^2 + L^2) \sum_{k=0}^{K-1} \mathbb{E}\|z^{k+1/2} - w^k\|^2 + \frac{1}{2\gamma} \max_{z \in \mathcal{C}} \|z^0 - z\|^2. \tag{52}$$

Substituting (52) to (50), we get

$$2\gamma\mathbb{E}\left[\max_{z\in\mathcal{C}}\sum_{k=0}^{K-1}\langle F(z), z^{k+1/2} - z\rangle\right]$$

$$\leq \mathbb{E}\left[\max_{z\in\mathcal{C}}\|\hat{z}^0 - z\|^2\right] + \mathbb{E}\left[\max_{z\in\mathcal{C}}\|w^0 - z\|^2\right] + \mathbb{E}\left[\max_{z\in\mathcal{C}}\|z^0 - z\|^2\right]$$

$$+ (1-\tau)\cdot\sum_{k=0}^{K-1}\mathbb{E}\left[\|z^k - w^k\|^2\right]$$

$$+ 2\gamma^2(\hat{L}^2 + L^2)\sum_{k=0}^{K-1}\mathbb{E}\|z^{k+1/2} - w^k\|^2 + \max_{z\in\mathcal{C}}\|z^0 - z\|^2.$$

With $\gamma \leq \frac{\sqrt{1-\tau}}{2L+2\hat{L}}$ we have

$$2\gamma\mathbb{E}\left[\max_{z\in\mathcal{C}}\sum_{k=0}^{K-1}\langle F(z), z^{k+1/2} - z\rangle\right]$$

$$\leq \mathbb{E}\left[\max_{z\in\mathcal{C}}\|\hat{z}^0 - z\|^2\right] + \mathbb{E}\left[\max_{z\in\mathcal{C}}\|w^0 - z\|^2\right] + 2\mathbb{E}\left[\max_{z\in\mathcal{C}}\|z^0 - z\|^2\right]$$

$$+ 3(1-\tau)\cdot\sum_{k=0}^{K-1}\mathbb{E}\left[\|z^k - z^{k+1/2}\|^2\right] + \mathbb{E}\left[\|z^{k+1/2} - w^k\|^2\right]. \tag{53}$$

Taking into account (49) with $p = 1$ and $\mu = 0$ (monotone case), we get

$$\sum_{k=0}^{K-1}\left(\mathbb{E}\|\hat{z}^{k+1} - z^*\|^2 + \mathbb{E}\|w^{k+1} - z^*\|^2\right)$$

$$\leq \sum_{k=0}^{K-1}\left(\mathbb{E}\|\hat{z}^k - z^*\|^2 + \mathbb{E}\|w^k - z^*\|^2\right)$$

$$- (1 - \tau - 13200(\delta^{\mathrm{serv}})^2(\delta^{\mathrm{dev}})^2\gamma^2\hat{L}^2)\cdot\sum_{k=0}^{K-1}\mathbb{E}\|w^k - z^{k+1/2}\|^2$$

$$- \left(\tau - \frac{1}{2}\right)\cdot\sum_{k=0}^{K-1}\mathbb{E}\|z^{k+1/2} - z^k\|^2.$$

With $\gamma \leq \frac{\sqrt{1-\tau}}{2L+165\delta^{\mathrm{serv}}\delta^{\mathrm{dev}}\tilde{L}}$ we get

$$\frac{1-\tau}{2}\cdot\sum_{k=0}^{K-1}\left(\mathbb{E}\|w^k - z^{k+1/2}\|^2 + \mathbb{E}\|z^{k+1/2} - z^k\|^2\right) \leq \left(\mathbb{E}\|\hat{z}^0 - z^*\|^2 + \mathbb{E}\|w^0 - z^*\|^2\right).$$

Combing this expression with (53), we obtain

$$2\gamma\mathbb{E}\left[\max_{z\in\mathcal{C}}\sum_{k=0}^{K-1}\langle F(z), z^{k+1/2} - z\rangle\right]$$

$$\leq \mathbb{E}\left[\max_{z\in\mathcal{C}}\|\hat{z}^0 - z\|^2\right] + \mathbb{E}\left[\max_{z\in\mathcal{C}}\|w^0 - z\|^2\right] + 2\mathbb{E}\left[\max_{z\in\mathcal{C}}\|z^0 - z\|^2\right]$$

$$+ 6\left(\mathbb{E}\|\hat{z}^0 - z^*\|^2 + \mathbb{E}\|w^0 - z^*\|^2\right),$$

and finish the proof in the monotone case.

$$\square$$

## Non-monotone case

We start from (46), put $z = z^*$, use non-monotonicity assumption and then take a full mathematical expectation:

$$
\begin{aligned}
\mathbb{E}\|\hat{z}^{k+1} - z^*\|^2 \leq{} & \mathbb{E}\|\hat{z}^k - z^*\|^2 - (1-\tau)\mathbb{E}\|z^k - z^*\|^2 + (1-\tau)\mathbb{E}\|w^k - z^*\|^2 \\
& - 2\gamma\mathbb{E}\left[\left\langle \left(\frac{1}{M}\sum_{m=1}^{M}(F_{m,\pi_m^k}(z^{k+1/2}) - F_{m,\pi_m^k}(w^k))\right) + F(w^k), z^{k+1/2} - z^*\right\rangle\right] \\
& - (1-\tau)\mathbb{E}\|w^k - z^{k+1/2}\|^2 + 2\gamma^2 \cdot \mathbb{E}\left\|\frac{1}{M}\sum_{m=1}^{M}(F_{m,\pi_m^k}(z^{k+1/2}) - F_{m,\pi_m^k}(w^k))\right\|^2 \\
& + 6\mathbb{E}\|e^k\|^2 + \frac{6}{M}\sum_{m=1}^{M}\mathbb{E}\left\|e_m^k\right\|^2 - \left(\tau - \frac{1}{2}\right)\mathbb{E}\|z^{k+1/2} - z^k\|^2 \\
\leq{} & \mathbb{E}\|\hat{z}^k - z^*\|^2 - (1-\tau)\mathbb{E}\|z^k - z^*\|^2 + (1-\tau)\mathbb{E}\|w^k - z^*\|^2 \\
& - 2\gamma\mathbb{E}\left[\left\langle \mathbb{E}_{\pi^k}\left[\frac{1}{M}\sum_{m=1}^{M}(F_{m,\pi_m^k}(z^{k+1/2}) - F_{m,\pi_m^k}(w^k)) + F(w^k)\right], z^{k+1/2} - z^*\right\rangle\right] \\
& - (1-\tau)\mathbb{E}\|w^k - z^{k+1/2}\|^2 + 2\gamma^2 \cdot \frac{1}{M}\sum_{m=1}^{M}\mathbb{E}\left[\mathbb{E}_{\pi^k}\left\|F_{m,\pi_m^k}(z^{k+1/2}) - F_{m,\pi_m^k}(w^k)\right\|^2\right] \\
& + 6\mathbb{E}\|e^k\|^2 + \frac{6}{M}\sum_{m=1}^{M}\mathbb{E}\left\|e_m^k\right\|^2 - \left(\tau - \frac{1}{2}\right)\mathbb{E}\|z^{k+1/2} - z^k\|^2 \\
={} & \mathbb{E}\|\hat{z}^k - z^*\|^2 - (1-\tau)\mathbb{E}\|z^k - z^*\|^2 + (1-\tau)\mathbb{E}\|w^k - z^*\|^2 \\
& - 2\gamma\mathbb{E}\left[\langle F(z^{k+1/2}, z^{k+1/2} - z^*\rangle\right] \\
& - (1-\tau)\mathbb{E}\|w^k - z^{k+1/2}\|^2 + 2\gamma^2 \cdot \frac{1}{M}\sum_{m=1}^{M}\mathbb{E}\left[\frac{1}{r}\sum_{i=1}^{r}\left\|F_{m,i}(z^{k+1/2}) - F_{m,i}(w^k)\right\|^2\right] \\
& + 6\mathbb{E}\|e^k\|^2 + \frac{6}{M}\sum_{m=1}^{M}\mathbb{E}\left\|e_m^k\right\|^2 - \left(\tau - \frac{1}{2}\right)\mathbb{E}\|z^{k+1/2} - z^k\|^2 \\
={} & \mathbb{E}\|\hat{z}^k - z^*\|^2 - (1-\tau)\mathbb{E}\|z^k - z^*\|^2 + (1-\tau)\mathbb{E}\|w^k - z^*\|^2 \\
& - (1-\tau)\mathbb{E}\|w^k - z^{k+1/2}\|^2 + 2\gamma^2\hat{L}^2\mathbb{E}\|w^k - z^{k+1/2}\|^2 \\
& + 6\mathbb{E}\|e^k\|^2 + \frac{6}{M}\sum_{m=1}^{M}\mathbb{E}\left\|e_m^k\right\|^2 - \left(\tau - \frac{1}{2}\right)\mathbb{E}\|z^{k+1/2} - z^k\|^2.
\end{aligned}
$$

This expression is the same with (43). Then we repeat all steps from Theorem E.1. And then with $\gamma \leq \frac{\sqrt{1-\tau}}{2L + 165\delta^{\text{serv}}\delta^{\text{dev}}\hat{L}}$ we get

$$
\frac{1}{K}\sum_{k=0}^{K-1}\mathbb{E}\|F(w^k)\|^2 \leq \frac{16(\mathbb{E}\|\hat{z}^0 - z^*\|^2 + \mathbb{E}\|w^0 - z^*\|^2)}{\gamma^2 K}.
$$

# G Federated learning and partial participation

Here we consider a popular federated learning feature - partial participation. We model it as follows. At each iteration, only $b$ random devices send information to the server. The rest do not compute and do not communicate. More formally, at each iteration we

$$\text{generate subset } \{\xi_i^k\}_{i=1}^b \text{ of } \{1, \dots, M\} \tag{54}$$

devices, which takes part in the current iteration. Next, we show how to modify MASHA1 and MASHA2 for partial participation.

## G.1 PP-MASHA1: federated learning version

In this section, we provide information about PP-MASHA1. This is a modification of MASHA1 for the federated learning case. Changes compared to MASHA1 are highlighted in blue – see Algorithm 7.

---
**Algorithm 7** PP-MASHA1
---
1: **Parameters:** Stepsize $\gamma > 0$, parameters $\tau$ and $b$, number of iterations $K$.
2: **Initialization:** Choose $z^0 = w^0 \in \mathcal{Z}$.
3: Server sends to devices $z^0 = w^0$ and devices compute $F_m(w^0)$ and send to server and get $F(w^0)$
4: **for** $k = 0, 1, 2, \dots, K-1$ **do**
5:     Generate subset $\{\xi_i^k\}_{i=1}^b$ of $\{1, \dots, M\}$ independently
6:     **for** each device $m$ from $\{\xi_i^k\}_{i=1}^b$ in parallel **do**
7:         $\bar{z}^k = \tau z^k + (1 - \tau) w^k$
8:         $z^{k+1/2} = \bar{z}_k - \gamma F(w^k)$,
9:         Compute $F_m(z^{k+1/2})$ & send $Q_m^{\text{dev}}(F_m(z^{k+1/2}) - F_m(w^k))$ to server
10:     **end for**
11:     **for** server **do**
12:         Compute $Q^{\text{serv}}\left[\frac{1}{b}\sum_{i=1}^b Q_{\xi_i^k}^{\text{dev}}(F_{\xi_i^k}(z^{k+1/2}) - F_{\xi_i^k}(w^k))\right]$ & send to devices
13:         Sends to devices one bit $b_k$: 1 with probability $1 - \tau$, 0 with with probability $\tau$
14:     **end for**
15:     **for** each device $m$ in parallel **do**
16:         $z^{k+1} = z^{k+1/2} - \gamma Q^{\text{serv}}\left[\frac{1}{b}\sum_{i=1}^b Q_{\xi_i^k}^{\text{dev}}(F_{\xi_i^k}(z^{k+1/2}) - F_{\xi_i^k}(w^k))\right]$
17:         **if** $b_k = 1$ **then**
18:             $w^{k+1} = z^{k+1}$
19:             Compute $F_m(w^{k+1})$ & send it to server; and get $F(w^{k+1})$ as a response from server
20:         **else**
21:             $w^{k+1} = w^k$
22:         **end if**
23:     **end for**
24: **end for**
---

The following theorem gives the convergence of PP-MASHA1.

**Theorem G.1** *Let distributed variational inequality (3) + (4) + (54) is solved by Algorithm 7 with unbiased compressor operators (1): on server with $q^{\text{serv}}$ parameter, on devices with $\{q_m^{\text{dev}}\}$. Let Assumption 3.4 and one case of Assumption 3.5 are satisfied. Then the following estimates holds*

- *in strongly-monotone case with $\gamma \leq \min\left[\frac{\sqrt{1-\tau}}{2C_q^b}; \frac{1-\tau}{2\mu}\right]$ (where $C_q^b = \sqrt{\frac{q^{\text{serv}}}{bM} \cdot \sum_{m=1}^M (q_m^{\text{dev}} \tilde{L}_m^2 + (b-1)\tilde{L}^2)}$):*

$$\mathbb{E}\left(\|z^K - z^*\|^2 + \|w^K - z^*\|^2\right) \leq \left(1 - \frac{\mu\gamma}{2}\right)^K \cdot 2\|z^0 - z^*\|^2;$$

- *in monotone case with $\gamma \leq \frac{\sqrt{1-\tau}}{2C_q^b + 4\tilde{L}}$:*

$$\mathbb{E}\left[\max_{z \in \mathcal{C}}\left[\left\langle F(u), \left(\frac{1}{K}\sum_{k=0}^{K-1}z^{k+1/2}\right) - u\right\rangle\right]\right] \leq \frac{2\max_{z \in \mathcal{C}}\left[\|z^0 - z\|^2\right] + 6\|z^0 - z^*\|^2}{\gamma K};$$

- *in non-monotone case with $\gamma \leq \frac{\sqrt{1-\tau}}{2C_q^b}$:*

$$\mathbb{E}\left(\frac{1}{K}\sum_{k=0}^{K-1}\|F(w^k)\|^2\right) \leq \frac{16\|z^0 - z^*\|^2}{\gamma^2 K}.$$

For PP-MASHA1 we consider the case of only devices compression. For simplicity, we put $Q_m^{\text{dev}} = Q$ with $q_m^{\text{dev}} = q$ and $\beta_m^{\text{dev}} = \beta$, also $\tilde{L}_m = \tilde{L} = L$. Let us discuss the difference (with MASHA1) in choosing $\tau$. When $b_k = 1$, all devices send uncompressed information to the server, but only $b$ devices send compressed information (line 9 PP-MASHA1). Then, at each iteration, we send $\mathcal{O}\left(\frac{b}{\beta} + M(1-\tau)\right)$ bits of information. Therefore, the optimal choice of $\tau$ is $1 - \tau = \frac{b}{\beta M}$.

**Corollary G.2** *Let distributed variational inequality* (3) + (4) + (54) *is solved by Algorithm 7 without compression on server ($q^{serv} = 1$) and with unbiased compressor operators* (1) *on devices with $\{q_m^{dev} = q\}$. Let Assumption 3.4 and one case of Assumption 3.5 are satisfied. Then the following estimates holds*

- *in strongly-monotone case with $\gamma \leq \min\left[\frac{1}{2L} \cdot \left(\sqrt{\frac{q\beta}{b} + \frac{\beta M}{b}}\right)^{-1}; \frac{b}{2\mu\beta M}\right]$:*

$$\mathbb{E}\left(\|z^K - z^*\|^2 + \|w^K - z^*\|^2\right) \leq \left(1 - \frac{\mu\gamma}{2}\right)^K \cdot 2\|z^0 - z^*\|^2;$$

- *in monotone case with $\gamma \leq \frac{1}{6L} \cdot \left(\sqrt{\frac{q\beta}{b} + \frac{\beta M}{b}}\right)^{-1}$:*

$$\mathbb{E}\left[\max_{z \in \mathcal{C}}\left[\left\langle F(u), \left(\frac{1}{K}\sum_{k=0}^{K-1}z^{k+1/2}\right) - u\right\rangle\right]\right] \leq \frac{2\max_{z \in \mathcal{C}}\left[\|z^0 - z\|^2\right] + 6\|z^0 - z^*\|^2}{\gamma K};$$

- *in non-monotone case with $\gamma \leq \frac{1}{2L} \cdot \left(\sqrt{\frac{q\beta}{b} + \frac{\beta M}{b}}\right)^{-1}$:*

$$\mathbb{E}\left(\frac{1}{K}\sum_{k=0}^{K-1}\|F(w^k)\|^2\right) \leq \frac{16\|z^0 - z^*\|^2}{\gamma^2 K}.$$

In the line 7 of Table 1 we put complexities to achieve $\varepsilon$-solution.

### G.1.1 Proof of the convergence of PP-MASHA1

**Proof of Theorem G.1:**

The proof is very close to the proof of Theorem D.1. Only two estimates need to be modified. First is (15)

$$\mathbb{E}\left[\|z^{k+1} - z^{k+1/2}\|^2\right] = \gamma^2 \cdot \mathbb{E}\left[\left\|Q^{\text{serv}}\left[\frac{1}{b}\sum_{i=1}^{b}Q_{\xi_i^k}^{\text{dev}}(F_{\xi_i^k}(z^{k+1/2}) - F_{\xi_i^k}(w^k))\right]\right\|^2\right]$$

$$\leq \gamma^2 \cdot \frac{q^{\text{serv}}}{b^2}\mathbb{E}\left[\left\|\sum_{m=1}^{M}Q_{\xi_i^k}^{\text{dev}}(F_{\xi_i^k}(z^{k+1/2}) - F_{\xi_i^k}(w^k))\right\|^2\right]$$

$$= \gamma^2 \cdot \frac{q^{\text{serv}}}{b^2} \sum_{i=1}^{b} \mathbb{E}\left[\left\|Q_{\xi_i^k}^{\text{dev}}(F_{\xi_i^k}(z^{k+1/2}) - F_{\xi_i^k}(w^k))\right\|^2\right]$$

$$+ \gamma^2 \cdot \frac{q^{\text{serv}}}{b^2} \sum_{i \neq j} \mathbb{E}\left[\langle Q_{\xi_i^k}^{\text{dev}}(F_{\xi_i^k}(z^{k+1/2}) - F_{\xi_i^k}(w^k)); Q_{\xi_j^k}^{\text{dev}}(F_{\xi_j^k}(z^{k+1/2}) - F_{\xi_j^k}(w^k))\rangle\right]$$

Next we apply (1) and Assumption 3.4 for the first term and independence and unbiasedness of $Q$ and uniformess of $\xi$ for the second term:

$$\mathbb{E}\left[\|z^{k+1} - z^{k+1/2}\|^2\right] \leq \gamma^2 \cdot \frac{q^{\text{serv}}}{b^2} \sum_{i=1}^{b} \mathbb{E}\left[\mathbb{E}_{\xi^k}\left[q_{\xi_i^k}^{\text{dev}} L_{\xi_i^k}^2\right]\left\|z^{k+1/2} - w^k\right\|^2\right]$$

$$+ \gamma^2 \cdot \frac{q^{\text{serv}}}{b^2} \sum_{i \neq j} \mathbb{E}\left[\langle F_{\xi_i^k}(z^{k+1/2}) - F_{\xi_i^k}(w^k); F_{\xi_j^k}(z^{k+1/2}) - F_{\xi_j^k}(w^k)\rangle\right]$$

$$\leq \gamma^2 \cdot \frac{q^{\text{serv}}}{bM} \sum_{m=1}^{M} q_m^{\text{dev}} \tilde{L}_m^2 \mathbb{E}\left[\left\|z^{k+1/2} - w^k\right\|^2\right]$$

$$+ \gamma^2 \cdot \frac{q^{\text{serv}}}{2b^2} \sum_{i \neq j} \mathbb{E}\left[\|F_{\xi_i^k}(z^{k+1/2}) - F_{\xi_i^k}(w^k)\|^2 + \|F_{\xi_j^k}(z^{k+1/2}) - F_{\xi_j^k}(w^k)\|^2\right]$$

$$\leq \gamma^2 \cdot \frac{q^{\text{serv}}}{bM} \sum_{m=1}^{M} q_m^{\text{dev}} \tilde{L}_m^2 \mathbb{E}\left[\left\|z^{k+1/2} - w^k\right\|^2\right]$$

$$+ \gamma^2 \cdot \frac{q^{\text{serv}}}{2b^2} \sum_{i \neq j} \mathbb{E}\left[L_{\xi_i^k}^2 \|z^{k+1/2} - w^k\|^2 + L_{\xi_j^k}^2 \|z^{k+1/2} - w^k\|^2\right]$$

$$= \gamma^2 \cdot \frac{q^{\text{serv}}}{bM} \sum_{m=1}^{M} q_m^{\text{dev}} \tilde{L}_m^2 \mathbb{E}\left[\left\|z^{k+1/2} - w^k\right\|^2\right]$$

$$+ \gamma^2 \cdot \frac{q^{\text{serv}}(b-1)}{b} \tilde{L}^2 \mathbb{E}\left[\|z^{k+1/2} - w^k\|^2\right]$$

$$= \gamma^2 \cdot \frac{q^{\text{serv}}}{M^2} \mathbb{E}\left[\|z^{k+1/2} - w^k\|^2\right] \cdot \sum_{m=1}^{M} q_m^{\text{dev}} \tilde{L}_m^2 + (M-1)\tilde{L}^2$$

Here we can use new $C_q^b = \sqrt{\frac{q^{\text{serv}}}{bM} \cdot \sum_{m=1}^{M}(q_m^{\text{dev}} \tilde{L}_m^2 + (b-1)\tilde{L}^2)}$.

The second modified estimate is (25):

$$\mathbb{E}\left[\left\|Q^{\text{serv}}\left[\frac{1}{M}\sum_{m=1}^{M} Q_m^{\text{dev}}(F_m(z^{k+1/2}) - F_m(w^k))\right] + F(w^k) - F(z^{k+1/2})\right\|^2\right]$$

$$\leq (C_q^b)^2 \mathbb{E}\left[\left\|z^{k+1/2} - w^k\right\|^2\right] + 2\tilde{L}^2 \mathbb{E}\left[\left\|z^{k+1/2} - w^k\right\|^2\right].$$

$\square$

## G.2  PP-MASHA2: federated learning version

In this section, we provide information about PP-MASHA2 from Section G. This is a modification of MASHA2 for the federated learning case. Changes compared to MASHA2 are highlighted in blue – see Algorithm 8.

---

**Algorithm 8** PP-MASHA2

1: **Parameters:** Stepsize $\gamma > 0$, parameters $\tau$ and $b$, number of iterations $K$.
2: **Initialization:** Choose $z^0 = w^0 \in \mathcal{Z}$, $e_m^0 = 0$, $e^0 = 0$.
3: Server sends to devices $z^0 = w^0$ and devices compute $F_m(w^0)$ and send to server and get $F(w^0)$
4: **for** $k = 0, 1, 2, \ldots, K - 1$ **do**
5:    Generate subset $\{\xi_i^k\}_{i=1}^b$ of $\{1, \ldots, M\}$ independently
6:    **for** each device $m$ from $\{\xi_i^k\}_{i=1}^b$ in parallel **do**
7:        $\bar{z}^k = \tau z^k + (1 - \tau) w^k$
8:        $z^{k+1/2} = \bar{z}_k - \gamma F(w^k)$
9:        Compute $F_m(z^{k+1/2})$ and send to server $C_m^{\text{dev}}(\gamma F_m(z^{k+1/2}) - \gamma F_m(w^k) + e_m^k)$
10:        $e_m^{k+1} = e_m^k + \gamma F_m(z^{k+1/2}) - \gamma F_m(w^k) - C_m^{\text{dev}}(\gamma F_m(z^{k+1/2}) - \gamma F_m(w^k) + e_m^k)$
11:    **end for**
12:    **for** devices not from $\{\xi_i^k\}_{i=1}^b$ in parallel **do**
13:        $e_m^{k+1} = e_m^k$
14:    **end for**
15:    **for** server **do**
16:        Compute $g^k = C^{\text{serv}}\left[\frac{1}{b}\sum_{i=1}^b C_{\xi_i^k}^{\text{dev}}(\gamma F_{\xi_i^k}(z^{k+1/2}) - \gamma F_{\xi_i^k}(w^k) + e_{\xi_i^k}^k) + e^k\right]$ & send to

   devices

17:        $e^{k+1} = e^k + \frac{1}{b}\sum_{i=1}^b C_{\xi_i^k}^{\text{dev}}(\gamma F_{\xi_i^k}(z^{k+1/2}) - \gamma F_{\xi_i^k}(w^k) + e_{\xi_i^k}^k) - g^k$.
18:        Sends to devices one bit $b_k$: 1 with probability $1 - \tau$, 0 with with probability $\tau$
19:    **end for**
20:    **for** each device $m$ in parallel **do**
21:        $z^{k+1} = z^{k+1/2} - C^{\text{serv}}\left[\frac{1}{b}\sum_{i=1}^b C_{\xi_i^k}^{\text{dev}}(\gamma F_{\xi_i^k}(z^{k+1/2}) - \gamma F_{\xi_i^k}(w^k) + e_{\xi_i^k}^k) + e^k\right]$
22:        **if** $b_k = 1$ **then**
23:            $w^{k+1} = z^k$
24:            Compute $F_m(w^{k+1})$ and it send to server; and get $F(w^{k+1})$
25:        **else**
26:            $w^{k+1} = w^k$
27:        **end if**
28:    **end for**
29: **end for**

---

The following theorem gives the convergence of PP-MASHA2.

**Theorem G.3** *Let distributed variational inequality (3) + (4) is solved by Algorithm 8 with $\tau \geq \frac{3}{4}$ and biased compressor operators (2): on server with $\delta^{serv}$ parameter, on devices with $\delta^{dev}$. Let Assumption 3.4 and one case of Assumption 3.5 are satisfied. Then the following estimates holds*

• *in strongly-monotone case with $\gamma \leq \min\left[\frac{1-\tau}{8\mu}; \frac{\sqrt{1-\tau}}{(30\delta^{serv}+10\delta^{dev}\frac{M}{b}+165\delta^{dev}\delta^{serv}\sqrt{\frac{M}{b}})\bar{L}}\right]$:*

$$\mathbb{E}\left(\|\hat{z}^K - z^*\|^2 + \|w^K - z^*\|^2\right) \leq \left(1 - \frac{\mu\gamma}{2}\right)^K \cdot 2\|z^0 - z^*\|^2;$$

• *in monotone case with $\gamma \leq \frac{\sqrt{1-\tau}}{(30\delta^{serv}+10\delta^{dev}\frac{M}{b}+165\delta^{dev}\delta^{serv}\sqrt{\frac{M}{b}})\bar{L}}$:*

$$\mathbb{E}\left[\max_{z\in\mathcal{C}}\langle F(z), \left(\frac{1}{K}\sum_{k=0}^{K-1} z^{k+1/2}\right) - z\rangle\right] \leq \frac{2\max_{z\in\mathcal{C}}\|z^0 - z\|^2 + 6\|z^0 - z^*\|^2}{\gamma K};$$

• *in non-monotone case with* $\gamma \le \frac{\sqrt{1-\tau}}{(30\delta^{serv}+10\delta^{dev}\frac{M}{b}+165\delta^{dev}\delta^{serv}\sqrt{\frac{M}{b}})\tilde{L}}$:

$$\frac{1}{K}\sum_{k=0}^{K-1}\mathbb{E}\|F(w^k)\|^2 \le \frac{32\mathbb{E}\|z^0-z^*\|^2}{\gamma^2 K}.$$

We consider the only devices compression. For simplicity, we put $\tilde{L}=\hat{L}=L$. We use the same reasoning as in Section G.1. The optimal choice is $1-\tau = \frac{b}{\beta M}$.

**Corollary G.4** *Let distributed variational inequality* (3) + (4) + (54) *is solved by Algorithm 8 without compression on server* ($\delta^{serv}=1$) *and with biased compressor operators* (2) *on devices with* $\delta^{dev}=\delta$. *Let Assumption 3.4 and one case of Assumption 3.5 are satisfied. Then the following estimates holds*

• *in strongly-monotone case with* $\gamma \le \min\left[\frac{1}{8\mu\beta}; \frac{\sqrt{b^3}}{205\delta\sqrt{\beta M^3 L}}\right]$:

$$\mathbb{E}\left(\|\hat{z}^K-z^*\|^2 + \|w^K-z^*\|^2\right) \le \left(1-\frac{\mu\gamma}{2}\right)^K \cdot 2\|z^0-z^*\|^2;$$

• *in monotone case with* $\gamma \le \frac{\sqrt{b^3}}{205\delta\sqrt{\beta M^3 L}}$:

$$\mathbb{E}\left[\max_{z\in\mathcal{C}}\langle F(z), \left(\frac{1}{K}\sum_{k=0}^{K-1}z^{k+1/2}\right)-z\rangle\right] \le \frac{2\max_{z\in\mathcal{C}}\|z^0-z\|^2+4\|z^0-z^*\|^2}{\gamma K};$$

• *in non-monotone case with* $\gamma \le \frac{\sqrt{b^3}}{205\delta\sqrt{\beta M^3 L}}$:

$$\mathbb{E}\left(\frac{1}{K}\sum_{k=0}^{K-1}\|F(w^k)\|^2\right) \le \frac{32\mathbb{E}\|z^0-z^*\|^2}{\gamma^2 K}.$$

In the line 8 of Table 1 we put complexities to achieve $\varepsilon$-solution.

### G.2.1  Proof of the convergence of PP-MASHA2

**Proof of Theorem G.3:** The proofs of Theorem G.3 partially repeat the proofs of Theorem E.1. We note the main changes in comparison with Theorem E.1.

The first difference is definition of "hat" sequences:

$$\hat{z}^k = z^k - e^k - \frac{1}{b}\sum_{m=1}^{M}e_m^k, \quad \hat{z}^{k+1/2} = z^{k+1/2} - e^k - \frac{1}{b}\sum_{m=1}^{M}e_m^k, \quad \hat{w}^k = w^k - e^k - \frac{1}{b}\sum_{m=1}^{M}e_m^k.$$

Then we modify an update of "hat" sequence (26):

$$\hat{z}^{k+1} = z^{k+1} - e^{k+1} - \frac{1}{b}\sum_{m=1}^{M}e_m^{k+1}$$

$$= z^{k+1/2} - C^{serv}\left[\frac{1}{b}\sum_{i=1}^{b}C_{\xi_i^k}^{dev}(\gamma F_{\xi_i^k}(z^{k+1/2}) - \gamma F_{\xi_i^k}(w^k) + e_{\xi_i^k}^k) + e^k\right]$$

$$- e^k - \frac{1}{b}\sum_{i=1}^{b}C_{\xi_i^k}^{dev}(\gamma F_{\xi_i^k}(z^{k+1/2}) - \gamma F_{\xi_i^k}(w^k) + e_{\xi_i^k}^k)$$

$$+ C^{serv}\left[\frac{1}{b}\sum_{i=1}^{b}C_{\xi_i^k}^{dev}(\gamma F_{\xi_i^k}(z^{k+1/2}) - \gamma F_{\xi_i^k}(w^k) + e_{\xi_i^k}^k) + e^k\right]$$

$$-\frac{1}{b}\sum_{i=1}^{b}\left[e_{\xi_i^k}^k + \gamma F_{\xi_i^k}(z^{k+1/2}) - \gamma F_{\xi_i^k}(w^k) - C_{\xi_i^k}^{\text{dev}}(\gamma F_{\xi_i^k}(z^{k+1/2}) - \gamma F_{\xi_i^k}(w^k) + e_{\xi_i^k}^k)\right]$$

$$-\frac{1}{b}\sum_{j\notin\{\xi_i^k\}_{i=1}^b} e_j^k$$

$$= z^{k+1/2} - e^k - \frac{1}{b}\sum_{m=1}^{M} e_m^k - \gamma\cdot\frac{1}{b}\sum_{i=1}^{b}(F_{\xi_i^k}(z^{k+1/2}) - F_{\xi_i^k}(w^k))$$

$$= \hat{z}^{k+1/2} - \gamma\cdot\frac{1}{b}\sum_{i=1}^{b}(F_{\xi_i^k}(z^{k+1/2}) - F_{\xi_i^k}(w^k)).$$

Hence, we need to modify (29)

$$\|\hat{z}^{k+1} - z\|^2 \le \|\hat{z}^k - z\|^2 + 2\langle\hat{z}^{k+1} - \hat{z}^k, z^{k+1/2} - z\rangle$$

$$+ 2\gamma^2\cdot\left\|\frac{1}{b}\sum_{i=1}^{b}(F_{\xi_i^k}(z^{k+1/2}) - F_{\xi_i^k}(w^k))\right\|^2 + 4\|e^k\|^2 + \frac{4M}{b^2}\sum_{m=1}^{M}\|e_m^k\|^2$$

$$- \|z^{k+1/2} - \hat{z}^k\|^2;$$

and (30):

$$\hat{z}^{k+1} - \hat{z}^k = \hat{z}^{k+1} - \hat{z}^{k+1/2} + \hat{z}^{k+1/2} - \hat{z}^k$$

$$= -\gamma\cdot\left(\frac{1}{b}\sum_{i=1}^{b}(F_{\xi_i^k}(z^{k+1/2}) - F_{\xi_i^k}(w^k))\right) + z^{k+1/2} - z^k$$

$$= -\gamma\cdot\left(\frac{1}{b}\sum_{i=1}^{b}(F_{\xi_i^k}(z^{k+1/2}) - F_{\xi_i^k}(w^k))\right) - \gamma\cdot F(w^k) + \bar{z}^k - z^k,$$

Then (31) is also modified:

$$\|\hat{z}^{k+1} - z\|^2 \le \|\hat{z}^k - z\|^2 - (1-\tau)\|z^k - z\|^2 + (1-\tau)\|w^k - z\|^2$$

$$- 2\gamma\langle\left(\frac{1}{b}\sum_{i=1}^{b}(F_{\xi_i^k}(z^{k+1/2}) - F_{\xi_i^k}(w^k))\right) + F(w^k), z^{k+1/2} - z\rangle$$

$$- (1-\tau)\|w^k - z^{k+1/2}\|^2 + 2\gamma^2\cdot\left\|\frac{1}{b}\sum_{i=1}^{b}(F_{\xi_i^k}(z^{k+1/2}) - F_{\xi_i^k}(w^k))\right\|^2$$

$$+ 6\|e^k\|^2 + \frac{6M}{b^2}\sum_{m=1}^{M}\|e_m^k\|^2 - \left(\tau - \frac{1}{2}\right)\|z^{k+1/2} - z^k\|^2. \tag{55}$$

Next, we move to different cases of monotonicity.

**Strongly-monotone**

The same way as in Theorem E.1 we put $z = z^*$, use property of the solution and then take full expectation:

$$\mathbb{E}\|\hat{z}^{k+1} - z^*\|^2 \le \mathbb{E}\|\hat{z}^k - z^*\|^2 - (1-\tau)\mathbb{E}\|z^k - z^*\|^2 + (1-\tau)\mathbb{E}\|w^k - z^*\|^2$$

$$- 2\gamma\mathbb{E}\left[\langle\left(\frac{1}{b}\sum_{i=1}^{b}(F_{\xi_i^k}(z^{k+1/2}) - F_{\xi_i^k}(w^k))\right) + F(w^k) - F(z^*), z^{k+1/2} - z^*\rangle\right]$$

$$- (1-\tau)\mathbb{E}\|w^k - z^{k+1/2}\|^2 + 2\gamma^2\cdot\mathbb{E}\left\|\frac{1}{b}\sum_{i=1}^{b}(F_{\xi_i^k}(z^{k+1/2}) - F_{\xi_i^k}(w^k))\right\|^2$$

$$+ 6\mathbb{E}\|e^k\|^2 + \frac{6M}{b^2}\sum_{m=1}^{M}\mathbb{E}\|e_m^k\|^2 - \left(\tau - \frac{1}{2}\right)\mathbb{E}\|z^{k+1/2} - z^k\|^2$$

$$\leq \mathbb{E}\|\hat{z}^k - z^*\|^2 - (1-\tau)\mathbb{E}\|z^k - z^*\|^2 + (1-\tau)\mathbb{E}\|w^k - z^*\|^2$$

$$- 2\gamma\mathbb{E}\left[\left\langle\mathbb{E}_{\xi^k}\left[\frac{1}{b}\sum_{i=1}^b(F_{\xi_i^k}(z^{k+1/2}) - F_{\xi_i^k}(w^k)) + F(w^k) - F(z^*)\right], z^{k+1/2} - z^*\right\rangle\right]$$

$$- (1-\tau)\mathbb{E}\|w^k - z^{k+1/2}\|^2 + 2\gamma^2 \cdot \frac{1}{b}\sum_{i=1}^b\mathbb{E}\left[\mathbb{E}_{\xi^k}L_{\xi_i^k}^2\left\|z^{k+1/2} - w^k\right\|^2\right]$$

$$+ 6\mathbb{E}\|e^k\|^2 + \frac{6M}{b^2}\sum_{m=1}^M\mathbb{E}\left\|e_m^k\right\|^2 - \left(\tau - \frac{1}{2}\right)\mathbb{E}\|z^{k+1/2} - z^k\|^2$$

$$= \mathbb{E}\|\hat{z}^k - z^*\|^2 - (1-\tau)\mathbb{E}\|z^k - z^*\|^2 + (1-\tau)\mathbb{E}\|w^k - z^*\|^2$$

$$- 2\gamma\mathbb{E}\left[\langle F(z^{k+1/2} - F(z^*), z^{k+1/2} - z^*\rangle\right]$$

$$- (1-\tau)\mathbb{E}\|w^k - z^{k+1/2}\|^2 + 2\gamma^2 \cdot \frac{1}{M}\sum_{m=1}^M L_m^2\mathbb{E}\left\|z^{k+1/2} - w^k\right\|^2$$

$$+ 6\mathbb{E}\|e^k\|^2 + \frac{6M}{b^2}\sum_{m=1}^M\mathbb{E}\left\|e_m^k\right\|^2 - \left(\tau - \frac{1}{2}\right)\mathbb{E}\|z^{k+1/2} - z^k\|^2$$

$$= \mathbb{E}\|\hat{z}^k - z^*\|^2 - (1-\tau)\mathbb{E}\|z^k - z^*\|^2 + (1-\tau)\mathbb{E}\|w^k - z^*\|^2$$

$$- 2\gamma\mathbb{E}\left[\langle F(z^{k+1/2} - F(z^*), z^{k+1/2} - z^*\rangle\right]$$

$$- (1-\tau)\mathbb{E}\|w^k - z^{k+1/2}\|^2 + 2\gamma^2\tilde{L}^2\mathbb{E}\|w^k - z^{k+1/2}\|^2$$

$$+ 6\mathbb{E}\|e^k\|^2 + \frac{6M}{b^2}\sum_{m=1}^M\mathbb{E}\left\|e_m^k\right\|^2 - \left(\tau - \frac{1}{2}\right)\mathbb{E}\|z^{k+1/2} - z^k\|^2. \tag{56}$$

In the last we use Assumption 3.4 and definition of $\tilde{L}$ from this Assumption. The new inequality (56) is absolutely similar to inequality (32) (only $L$ is changed to $\tilde{L}$ and a coefficient near $\sum_{m=1}^M\mathbb{E}\left\|e_m^k\right\|^2$). Therefore, we can safely reach the analogue of expression (34):

$$\sum_{k=0}^{K-1}p^k\mathbb{E}\|\hat{z}^{k+1} - z^*\|^2 + \sum_{k=0}^{K-1}p^k\mathbb{E}\|w^{k+1} - z^*\|^2$$

$$\leq \sum_{k=0}^{K-1}p^k\mathbb{E}\|\hat{z}^k - z^*\|^2 + \sum_{k=0}^{K-1}p^k\mathbb{E}\|w^k - z^*\|^2 - 2\gamma\mu\sum_{k=0}^{K-1}p^k\mathbb{E}\|z^{k+1/2} - z^*\|^2$$

$$- (1 - \tau - 2\gamma^2\tilde{L}^2) \cdot \sum_{k=0}^{K-1}p^k\mathbb{E}\|w^k - z^{k+1/2}\|^2 - \left(\tau - \frac{1}{2}\right) \cdot \sum_{k=0}^{K-1}p^k\mathbb{E}\|z^{k+1/2} - z^k\|^2$$

$$+ 6 \cdot \sum_{k=0}^{K-1}p^k\mathbb{E}\|e^k\|^2 + 6 \cdot \sum_{k=0}^{K-1}p^k\frac{M}{b^2}\sum_{m=1}^M\mathbb{E}\left\|e_m^k\right\|^2. \tag{57}$$

Next, we need modify estimates on "error" terms:

$$\mathbb{E}\|e^{k+1}\|^2 = \mathbb{E}\left\|e^k + \frac{1}{b}\sum_{i=1}^b C_{\xi_i^k}^{\text{dev}}(\gamma F_{\xi_i^k}(z^{k+1/2}) - \gamma F_{\xi_i^k}(w^k) + e_{\xi_i^k}^k)\right.$$

$$\left. - C^{\text{serv}}\left[\frac{1}{b}\sum_{i=1}^b C_{\xi_i^k}^{\text{dev}}(\gamma F_{\xi_i^k}(z^{k+1/2}) - \gamma F_{\xi_i^k}(w^k) + e_{\xi_i^k}^k) + e^k\right]\right\|^2$$

$$\leq \left(1 - \frac{1}{\delta^{\text{serv}}}\right)\mathbb{E}\left\|e^k + \frac{1}{b}\sum_{i=1}^b C_{\xi_i^k}^{\text{dev}}(\gamma F_{\xi_i^k}(z^{k+1/2}) - \gamma F_{\xi_i^k}(w^k) + e_{\xi_i^k}^k)\right\|^2$$

$$\leq (1 + c)\left(1 - \frac{1}{\delta^{\text{serv}}}\right)\mathbb{E}\left\|e^k\right\|^2$$

$$+ \left(1 + \frac{1}{c}\right)\left(1 - \frac{1}{\delta^{\text{serv}}}\right)\frac{1}{b}\sum_{i=1}^{b}\mathbb{E}\left\|C_{\xi_i^k}^{\text{dev}}(\gamma F_{\xi_i^k}(z^{k+1/2}) - \gamma F_{\xi_i^k}(w^k) + e_{\xi_i^k}^k)\right\|^2.$$

Here we use definition of biased compression (2), (10) and inequality $\|a + b\|^2 \le (1 + c)\|a\|^2 + (1 + 1/c)\|b\|^2$ (for $c > 0$). Is is easy to prove that for baised compressor $C_m^{\text{dev}}$ from (2) it holds that $\|C_m^{\text{dev}}(x)\|^2 \le 4\|x\|^2$ (see [10]). Then

$$\mathbb{E}\|e^{k+1}\|^2 \le (1 + c)\left(1 - \frac{1}{\delta^{\text{serv}}}\right)\mathbb{E}\left\|e^k\right\|^2$$

$$+ \left(1 + \frac{1}{c}\right)\left(1 - \frac{1}{\delta^{\text{serv}}}\right)\frac{4}{b}\sum_{i=1}^{b}\mathbb{E}\left\|\gamma F_{\xi_i^k}(z^{k+1/2}) - \gamma F_{\xi_i^k}(w^k) + e_{\xi_i^k}^k\right\|^2$$

$$\le (1 + c)\left(1 - \frac{1}{\delta^{\text{serv}}}\right)\mathbb{E}\left\|e^k\right\|^2$$

$$+ \gamma^2\left(1 + \frac{1}{c}\right)\left(1 - \frac{1}{\delta^{\text{serv}}}\right)\frac{8}{b}\sum_{i=1}^{b}\mathbb{E}\left\|F_{\xi_i^k}(z^{k+1/2}) - F_{\xi_i^k}(w^k)\right\|^2$$

$$+ \left(1 + \frac{1}{c}\right)\left(1 - \frac{1}{\delta^{\text{serv}}}\right)\frac{8}{b}\sum_{i=1}^{b}\mathbb{E}\left\|e_{\xi_i^k}^k\right\|^2$$

$$\le (1 + c)\left(1 - \frac{1}{\delta^{\text{serv}}}\right)\mathbb{E}\left\|e^k\right\|^2 + 8\gamma^2\tilde{L}^2\left(1 + \frac{1}{c}\right)\left(1 - \frac{1}{\delta^{\text{serv}}}\right)\mathbb{E}\left\|z^{k+1/2} - w^k\right\|^2$$

$$+ \left(1 + \frac{1}{c}\right)\left(1 - \frac{1}{\delta^{\text{serv}}}\right)\frac{8}{b}\sum_{m=1}^{M}\mathbb{E}\left\|e_m^k\right\|^2.$$

In the last we use Assumption 3.4 and definition of $\tilde{L}$ from this Assumption. With $c = \frac{1}{2(\delta - 1)}$ we get

$$\mathbb{E}\|e^{k+1}\|^2 \le \left(1 - \frac{1}{2\delta^{\text{serv}}}\right)\mathbb{E}\left\|e^k\right\|^2 + 16\delta^{\text{serv}}\gamma^2\tilde{L}^2 \cdot \mathbb{E}\|z^{k+1/2} - w^k\|^2 + 16\delta^{\text{serv}} \cdot \frac{1}{b}\sum_{m=1}^{M}\mathbb{E}\left\|e_m^k\right\|^2$$

$$\le 16\delta^{\text{serv}}\gamma^2\tilde{L}^2\sum_{j=0}^{k}\left(1 - \frac{1}{2\delta^{\text{serv}}}\right)^{k-j} \cdot \left\|z^{j+1/2} - w^j\right\|^2$$

$$+ 16\delta^{\text{serv}}\sum_{j=0}^{k}\left(1 - \frac{1}{2\delta^{\text{serv}}}\right)^{k-j} \cdot \frac{1}{b}\sum_{m=1}^{M}\left\|e_m^j\right\|^2.$$

The same way we can get analogue of (35):

$$\sum_{k=0}^{K-1}p^k\mathbb{E}\left\|e^k\right\|^2 \le 128(\delta^{\text{serv}})^2\gamma^2\tilde{L}^2\sum_{k=0}^{K-1}p^k\mathbb{E}\left\|z^{k+1/2} - w^k\right\|^2$$

$$+ 128(\delta^{\text{serv}})^2\sum_{k=0}^{K-1}p^k\frac{1}{b}\sum_{m=1}^{M}\mathbb{E}\left\|e_m^k\right\|^2. \tag{58}$$

Combining (57) with (58), we obtain

$$\sum_{k=0}^{K-1}p^k\mathbb{E}\|\hat{z}^{k+1} - z^*\|^2 + \sum_{k=0}^{K-1}p^k\mathbb{E}\|w^{k+1} - z^*\|^2$$

$$\le \sum_{k=0}^{K-1}p^k\mathbb{E}\|\hat{z}^k - z^*\|^2 + \sum_{k=0}^{K-1}p^k\mathbb{E}\|w^k - z^*\|^2 - 2\gamma\mu\sum_{k=0}^{K-1}p^k\mathbb{E}\|z^{k+1/2} - z^*\|^2$$

$$- (1 - \tau - 2\gamma^2\tilde{L}^2) \cdot \sum_{k=0}^{K-1}p^k\mathbb{E}\|w^k - z^{k+1/2}\|^2 - \left(\tau - \frac{1}{2}\right) \cdot \sum_{k=0}^{K-1}p^k\mathbb{E}\|z^{k+1/2} - z^k\|^2$$

$$+ 768(\delta^{\text{serv}})^2 \gamma^2 \tilde{L}^2 \sum_{k=0}^{K-1} p^k \mathbb{E} \left\| z^{k+1/2} - w^k \right\|^2$$

$$+ \left( 768(\delta^{\text{serv}})^2 + \frac{6M}{b} \right) \sum_{k=0}^{K-1} p^k \frac{1}{b} \sum_{m=1}^{M} \mathbb{E} \left\| e_m^k \right\|^2. \tag{59}$$

For the other "error" term. Let us note that $\|e_m^{k+1}\|^2 = \|e_m^k\|$ with probability $1 - \frac{b}{M}$ and $\|e_m^{k+1}\|^2 = \|e_m^k + \gamma F_m(z^{k+1/2}) - \gamma F_m(w^k) - C_m^{\text{dev}}(\gamma F_m(z^{k+1/2}) - \gamma F_m(w^k) + e_m^k)\|$ with probability $\frac{b}{M}$, then

$$\frac{1}{b} \sum_{m=1}^{M} \mathbb{E} \left\| e_m^{k+1} \right\|^2 = \frac{1}{b} \sum_{m=1}^{M} \frac{b}{M} \mathbb{E} \left\| e_m^k + \gamma F_m(z^{k+1/2}) - \gamma F_m(w^k) - C_m^{\text{dev}}(\gamma F_m(z^{k+1/2}) - \gamma F_m(w^k) + e_m^k) \right\|^2$$

$$+ \frac{1}{b} \sum_{m=1}^{M} \left( 1 - \frac{b}{M} \right) \mathbb{E} \left\| e_m^k \right\|^2$$

$$\leq \frac{1}{b} \sum_{m=1}^{M} \frac{b}{M} \left( 1 - \frac{1}{\delta^{\text{dev}}} \right) \left\| e_m^k + \gamma \cdot F_m(z^{k+1/2}) - \gamma \cdot F_m(w^k) \right\|^2 + \left( 1 - \frac{b}{M} \right) \mathbb{E} \left\| e_m^k \right\|^2$$

$$\leq \frac{1}{b} \sum_{m=1}^{M} \frac{b}{M} (1+c) \left( 1 - \frac{1}{\delta^{\text{dev}}} \right) \left\| e_m^k \right\|^2 + \frac{b}{M} \left( 1 + \frac{1}{c} \right) \left( 1 - \frac{1}{\delta^{\text{dev}}} \right) \gamma^2 \cdot \left\| F_m(z^{k+1/2}) - F_m(w^k) \right\|^2$$

$$+ \frac{1}{b} \sum_{m=1}^{M} \left( 1 - \frac{b}{M} \right) \mathbb{E} \left\| e_m^k \right\|^2.$$

With $c = \frac{1}{2(\delta^{\text{dev}} - 1)}$

$$\frac{1}{b} \sum_{m=1}^{M} \left\| e_m^{k+1} \right\|^2 \leq \frac{1}{b} \sum_{m=1}^{M} \frac{b}{M} \left( 1 - \frac{1}{2\delta^{\text{dev}}} \right) \left\| e_m^k \right\|^2 + \frac{2b\delta^{\text{dev}}\gamma^2}{M} \cdot \left\| F_m(z^{k+1/2}) - F_m(w^k) \right\|^2$$

$$+ \frac{1}{b} \sum_{m=1}^{M} \left( 1 - \frac{b}{M} \right) \mathbb{E} \left\| e_m^k \right\|^2$$

$$\leq \left( 1 - \frac{b}{2\delta^{\text{dev}} M} \right) \cdot \frac{1}{b} \sum_{m=1}^{M} \left\| e_m^k \right\|^2 + 2\delta^{\text{dev}}\gamma^2 \tilde{L}^2 \cdot \left\| z^{k+1/2} - w^k \right\|^2$$

$$\leq 2\delta^{\text{dev}}\gamma^2 \tilde{L}^2 \sum_{j=0}^{k} \left( 1 - \frac{b}{2\delta^{\text{dev}} M} \right)^{k-j} \cdot \left\| z^{j+1/2} - w^j \right\|^2.$$

And then

$$\sum_{k=0}^{K-1} p^k \frac{1}{b} \sum_{m=1}^{M} \left\| e_m^k \right\|^2 \leq 16(\delta^{\text{dev}})^2 \gamma^2 \tilde{L}^2 \frac{M}{b} \sum_{k=0}^{K-1} p^k \left\| z^{k+1/2} - w^k \right\|^2. \tag{60}$$

Hence, (60) together with (59) gives

$$\sum_{k=0}^{K-1} p^k \mathbb{E} \| \hat{z}^{k+1} - z^* \|^2 + \sum_{k=0}^{K-1} p^k \mathbb{E} \| w^{k+1} - z^* \|^2$$

$$\leq \sum_{k=0}^{K-1} p^k \mathbb{E} \| \hat{z}^k - z^* \|^2 + \sum_{k=0}^{K-1} p^k \mathbb{E} \| w^k - z^* \|^2 - 2\gamma\mu \sum_{k=0}^{K-1} p^k \mathbb{E} \| z^{k+1/2} - z^* \|^2$$

$$- (1 - \tau - 2\gamma^2 \tilde{L}^2) \cdot \sum_{k=0}^{K-1} p^k \mathbb{E} \| w^k - z^{k+1/2} \|^2 - \left( \tau - \frac{1}{2} \right) \cdot \sum_{k=0}^{K-1} p^k \mathbb{E} \| z^{k+1/2} - z^k \|^2$$

$$+ 768(\delta^{\text{serv}})^2 \gamma^2 \tilde{L}^2 \sum_{k=0}^{K-1} p^k \mathbb{E} \left\| z^{k+1/2} - w^k \right\|^2$$

$$+ \left( 768(\delta^{\text{serv}})^2 + \frac{6M}{b} \right) \cdot 16(\delta^{\text{dev}})^2 \gamma^2 \tilde{L}^2 \frac{M}{b} \sum_{k=0}^{K-1} p^k \left\| z^{k+1/2} - w^k \right\|^2$$

$$\leq \sum_{k=0}^{K-1} p^k \mathbb{E}\|\hat{z}^k - z^*\|^2 + \sum_{k=0}^{K-1} p^k \mathbb{E}\|w^k - z^*\|^2 - 2\gamma\mu \sum_{k=0}^{K-1} p^k \mathbb{E}\|z^{k+1/2} - z^*\|^2$$

$$- \left( 1 - \tau - 768(\delta^{\text{serv}})^2 \gamma^2 \tilde{L}^2 - 12400\gamma^2 (\delta^{\text{serv}})^2 (\delta^{\text{dev}})^2 \frac{M}{b} \tilde{L}^2 - 96(\delta^{\text{dev}})^2 \gamma^2 \frac{M^2}{b^2} \tilde{L}^2 \right) \cdot \sum_{k=0}^{K-1} p^k \mathbb{E}\|w^k - z^{k+1/2}\|^2$$

$$- \left( \tau - \frac{1}{2} \right) \cdot \sum_{k=0}^{K-1} p^k \mathbb{E}\|z^{k+1/2} - z^k\|^2.$$

The same as in Theorem E.1 with $\tau \geq \frac{3}{4}$, $\gamma \leq \min\left[ \frac{1-\tau}{8\mu} ; \frac{\sqrt{1-\tau}}{(30\delta^{\text{serv}} + 10\delta^{\text{dev}} \frac{M}{b} + 165\delta^{\text{dev}} \delta^{\text{serv}} \sqrt{\frac{M}{b}})\tilde{L}} \right]$

**Monotone and Non-monotone cases**

The proof of the monotone and non-monotone cases repeat the techniques from Theorem F.4 (modifications of Theorem E.1) + techniques and estimates obtained in the proof of the strongly-monotone case in Theorem G.3.

# H  CEG: additional method

In this section, we present CEG with unbiased compression on devices. This is a very simple method. We prove its convergence only in the strongly monotone case and need this result to support the propositions in Sections 4.1 and B.

---

**Algorithm 9** CEG : Compressed Extra Gradient

---

1: **Parameters:** Stepsize $\gamma > 0$, number of iterations $K$.
2: **Initialization:** Choose $z^0 \in \mathcal{Z}$.
3: Server sends to devices $z^0$
4: **for** $k = 0, 1, 2, \ldots, K-1$ **do**
5:     **for** each device $m$ in parallel **do**
6:        Compute $F_m(z^k)$ & send $Q_m^{\text{dev}}(F_m(z^k))$ to server
7:     **end for**
8:     **for** server **do**
9:        Compute $\frac{1}{M} \sum\limits_{m=1}^{M} Q_m^{\text{dev}}(F_m(z^k))$ & send to devices
10:     **end for**
11:     **for** each device $m$ in parallel **do**
12:        $z^{k+1/2} = z^k - \gamma \cdot \frac{1}{M} \sum\limits_{m=1}^{M} Q_m^{\text{dev}}(F_m(z^k))$
13:        Compute $F_m(z^{k+1/2})$ & send $Q_m^{\text{dev}}(F_m(z^{k+1/2}))$ to server
14:     **end for**
15:     **for** server **do**
16:        Compute $\frac{1}{M} \sum\limits_{m=1}^{M} Q_m^{\text{dev}}(F_m(z^{k+1/2}))$ & send to devices
17:     **end for**
18:     **for** each device $m$ in parallel **do**
19:        $z^{k+1} = z^k - \gamma \cdot \frac{1}{M} \sum\limits_{m=1}^{M} Q_m^{\text{dev}}(F_m(z^{k+1/2}))$
20:     **end for**
21: **end for**

---

**Theorem H.1** *Let distributed variational inequality* (3) + (4) *is solved by Algorithm 9 with unbiased compressor operators* (1) *on devices with* $\{q_m^{dev} = q\}$. *Let Assumptions 3.4 and 3.5 (SM) are satisfied. Then the following estimates holds with* $\gamma \leq \min\left[ \frac{\mu}{48L^2} \cdot (1 + q/M)^{-1} ; \frac{1}{4\mu} \right]$:

$$\mathbb{E}\|z^K - z^*\|^2 \leq \left(1 - \frac{\mu\gamma}{2}\right)^K \cdot \|z^0 - z^*\|^2 + \frac{16q\gamma^2}{M^2} \sum_{m=1}^{M} \mathbb{E}\left[\|F_m(z^*)\|^2\right] ;$$

With $F_m(z^*) = 0$ we get the following estimates on iteration and bits complexities:

$$\mathcal{O}\left(\left[\left(1 + \frac{q}{M}\right)\frac{L^2}{\mu^2}\right] \log \frac{1}{\varepsilon}\right), \quad \mathcal{O}\left(\left[\left(\frac{1}{\beta} + \frac{q}{M\beta}\right)\frac{L^2}{\mu^2}\right] \log \frac{1}{\varepsilon}\right).$$

**Proof of Theorem H.1:** By a classical analysis of Extra Gradient in the strongly monotone (see [26] or [11]) case we get

$$\mathbb{E}\left[\|z^{k+1} - z^*\|^2\right] \leq \mathbb{E}\left[\|z^k - z^*\|^2\right] - \mathbb{E}\left[\|z^{k+1/2} - z^k\|^2\right]$$
$$- 2\gamma\mathbb{E}\left[\langle g^{k+1/2}, z^{k+1/2} - z^*\rangle\right] + \gamma^2 \mathbb{E}\left[\|g^{k+1/2} - g^k\|^2\right],$$

where $g^{k+1/2} = \frac{1}{M} \sum\limits_{m=1}^{M} Q\left(F_m(x^{k+1/2})\right)$ and $g^k = \frac{1}{M} \sum\limits_{m=1}^{M} Q\left(F_m(x^k)\right)$. With unbiasedness of compression we get

$$\mathbb{E}\left[\|z^{k+1} - z^*\|^2\right] \leq \mathbb{E}\left[\|z^k - z^*\|^2\right] - \mathbb{E}\left[\|z^{k+1/2} - z^k\|^2\right]$$

$$- 2\gamma\mathbb{E}\left[\langle F(z^{k+1/2}), z^{k+1/2} - z^*\rangle\right] + 2\gamma^2\mathbb{E}\left[\|g^{k+1/2} - F(z^*)\|^2\right] + 2\gamma^2\mathbb{E}\left[\|F(z^*) - g^k\|^2\right].$$

Then

$$\mathbb{E}\left[\left\|\frac{1}{M}\sum_{m=1}^{M}Q\left(F(z^k)\right) - F(z^*)\right\|^2\right] = \mathbb{E}\left[\left\|\frac{1}{M}\sum_{m=1}^{M}\left[Q\left(F_m(z^k)\right) - F_m(z^k) + F_m(z^k)\right] - F(z^*)\right\|^2\right]$$

$$\leq 2\mathbb{E}\left[\left\|\frac{1}{M}\sum_{m=1}^{M}Q\left(F_m(z^k)\right) - F_m(z^k)\right\|^2\right]$$

$$+ 2\mathbb{E}\left\|F(z^k) - F(z^*)\right\|^2$$

$$= \frac{2}{M^2}\sum_{m=1}^{M}\mathbb{E}\left[\left\|Q\left(F_m(z^k)\right) - F_m(z^k)\right\|^2\right]$$

$$+ \frac{2}{M^2}\sum_{i\neq j}^{M}\mathbb{E}\left[\langle Q\left(F_i(z^k)\right) - F_i(z^k), Q\left(F_j(z^k)\right) - F_j(z^k)\rangle\right]$$

$$+ 2\mathbb{E}\left[\left\|F(z^k) - F(z^*)\right\|^2\right].$$

Using definition of (1), we get

$$\mathbb{E}\left[\left\|\frac{1}{M}\sum_{m=1}^{M}Q\left(F(z^k)\right) - F(z^*)\right\|^2\right] \leq \frac{2q}{M^2}\sum_{m=1}^{M}\mathbb{E}\left[\left\|F_m(z^k)\right\|^2\right]$$

$$+ 2\mathbb{E}\left[\left\|F(z^k) - F(z^*)\right\|^2\right]$$

$$\leq \frac{4q}{M^2}\sum_{m=1}^{M}\mathbb{E}\left[\left\|F_m(z^k) - F_m(z^*)\right\|^2\right] + \frac{4q}{M^2}\sum_{m=1}^{M}\mathbb{E}\left[\left\|F_m(z^*)\right\|^2\right]$$

$$+ 2\left[\left\|F(z^k) - F(z^*)\right\|^2\right].$$

With Assumptions 3.4 we get

$$\mathbb{E}\left[\left\|\frac{1}{M}\sum_{m=1}^{M}Q\left(F(z^k)\right) - F(z^*)\right\|^2\right] \leq \frac{4qL^2}{M}\mathbb{E}\left[\left\|z^k - z^*\right\|^2\right] + \frac{4q}{M^2}\sum_{m=1}^{M}\mathbb{E}\left[\left\|F_m(z^*)\right\|^2\right]$$

$$+ 2L^2\mathbb{E}\left[\left\|z^k - z^*\right\|^2\right]$$

$$\leq 4L^2\left(1 + \frac{q}{M}\right)\mathbb{E}\left[\left\|z^k - z^*\right\|^2\right] + \frac{4q}{M^2}\sum_{m=1}^{M}\mathbb{E}\left[\left\|F_m(z^*)\right\|^2\right].$$

The same way we can get

$$\mathbb{E}\left[\left\|\frac{1}{M}\sum_{m=1}^{M}Q\left(F(z^{k+1/2})\right) - F(z^*)\right\|^2\right] \leq \frac{4qL^2}{M}\mathbb{E}\left[\left\|z^k - z^*\right\|^2\right] + \frac{4q}{M^2}\sum_{m=1}^{M}\mathbb{E}\left[\left\|F_m(z^*)\right\|^2\right]$$

$$+ 2L^2\mathbb{E}\left[\left\|z^k - z^*\right\|^2\right]$$

$$\leq 4L^2\left(1 + \frac{q}{M}\right)\mathbb{E}\left[\left\|z^{k+1/2} - z^*\right\|^2\right] + \frac{4q}{M^2}\sum_{m=1}^{M}\mathbb{E}\left[\left\|F_m(z^*)\right\|^2\right].$$

Finally, we obtain

$$\mathbb{E}\left[\|z^{k+1} - z^*\|^2\right] \leq \mathbb{E}\left[\|z^k - z^*\|^2\right] - \mathbb{E}\left[\|z^{k+1/2} - z^k\|^2\right]$$

$$- 2\gamma\mathbb{E}\left[\langle F(z^{k+1/2}), z^{k+1/2} - z^*\rangle\right] + 8\gamma^2L^2\left(1 + \frac{q}{M}\right)\mathbb{E}\left[\|z^k - z^*\|^2\right]$$

$$+ 8\gamma^2 L^2 \left(1 + \frac{q}{M}\right) \mathbb{E}\left[\left\|z^{k+1/2} - z^*\right\|^2\right] + \frac{16q\gamma^2}{M^2} \sum_{m=1}^{M} \mathbb{E}\left[\left\|F_m(z^*)\right\|^2\right].$$

With Assumption 3.5(SM) we obtain

$$
\begin{aligned}
\mathbb{E}\left[\|z^{k+1} - z^*\|^2\right] &\leq \mathbb{E}\left[\|z^k - z^*\|^2\right] - \mathbb{E}\left[\|z^{k+1/2} - z^k\|^2\right] \\
&\quad - 2\gamma\mu\mathbb{E}\left[\left\|z^{k+1/2} - z^*\right\|^2\right] + 8\gamma^2 L^2 \left(1 + \frac{q}{M}\right) \mathbb{E}\left[\left\|z^k - z^*\right\|^2\right] \\
&\quad + 8\gamma^2 L^2 \left(1 + \frac{q}{M}\right) \mathbb{E}\left[\left\|z^{k+1/2} - z^*\right\|^2\right] + \frac{16q\gamma^2}{M^2} \sum_{m=1}^{M} \mathbb{E}\left[\left\|F_m(z^*)\right\|^2\right] \\
&\leq \mathbb{E}\left[\|z^k - z^*\|^2\right] - \mathbb{E}\left[\|z^{k+1/2} - z^k\|^2\right] \\
&\quad - \gamma\mu\mathbb{E}\left[\left\|z^k - z^*\right\|^2\right] + 2\gamma\mu\mathbb{E}\left[\left\|z^{k+1/2} - z^k\right\|^2\right] + 8\gamma^2 L^2 \left(1 + \frac{q}{M}\right) \mathbb{E}\left[\left\|z^k - z^*\right\|^2\right] \\
&\quad + 16\gamma^2 L^2 \left(1 + \frac{q}{M}\right) \mathbb{E}\left[\left\|z^{k+1/2} - z^k\right\|^2\right] + 16\gamma^2 L^2 \left(1 + \frac{q}{M}\right) \mathbb{E}\left[\left\|z^k - z^*\right\|^2\right] \\
&\quad + \frac{16q\gamma^2}{M^2} \sum_{m=1}^{M} \mathbb{E}\left[\left\|F_m(z^*)\right\|^2\right].
\end{aligned}
$$

With $\gamma \leq \frac{1}{4\mu}; \frac{\mu}{48L^2(1+q/M)}$ we have

$$\mathbb{E}\left[\|z^{k+1} - z^*\|^2\right] \leq \left(1 - \gamma\mu + 24\gamma^2 L^2 \left(1 + \frac{q}{M}\right)\right) \mathbb{E}\left[\|z^k - z^*\|^2\right] + \frac{16q\gamma^2}{M^2} \sum_{m=1}^{M} \mathbb{E}\left[\left\|F_m(z^*)\right\|^2\right].$$

Choice $\gamma \leq \frac{\mu}{48L^2\left(1+\frac{q}{M}\right)}$ finishes the proof.

$\square$