# OpenReview forum: "Distributed Methods with Compressed Communication for Solving Variational Inequalities, with Theoretical Guarantees"
_NeurIPS.cc/2022/Conference — NeurIPS 2022 Accept_

### Official Review · Reviewer_SxRG · 2022-07-11

**Rating:** 8
**Confidence:** 3
**Soundness:** 4 excellent
**Presentation:** 3 good
**Contribution:** 4 excellent

**Summary:**

This paper designs and analyzes compression algorithms for solving three families of distributed VI/SPP problems.

**Questions:**

N/A

**Strengths And Weaknesses:**

The strengths of the work include extensive theoretical and experimental results highlighting that its designed algorithms outperform existing algorithms for solving VI/SPP. In particular, the results for strongly-monotone, monotone and non-monotone problems are provided.

I don't see any major weaknesses from this paper. However, I would like to point out to typo errors in the paper:

- Line 117-118: "But often ..." It is a run-on sentence.
- Line 125: "fro details" -> "for details"
- Line 241: "in str-monotone case" -> "in strongly-monotone case"

---

> ### Author Response · Authors · 2022-07-29
> **Response to Reviewer SxRG**
>
> We thank Reviewer **SxRG** for the work, and for the appreciation of our paper.
>
> We corrected all typos in the revision. If Reviewer has any further questions or comments on improving the paper, we will be happy to answer them.

---

### Official Review · Reviewer_1QcY · 2022-07-12

**Rating:** 7
**Confidence:** 3
**Soundness:** 3 good
**Presentation:** 3 good
**Contribution:** 3 good

**Summary:**

In this paper, the authors consider solving distributed variational inequalities (VIs) with compressed communication. For unbiased compressors, they use the variance reduction (VR) technique to control the variance; while for contractive compressors, they further incorporate the error compensation mechanism to handle the bias. They prove convergence rates for the cases where the operator is strongly-monotone, monotone, or non-monotone with Minty's condition, and demonstrate the speedup compared with the uncompressed counterparts. They further show how to extend their methods to the stochastic setting and the federated learning setting with partial participation.

**Questions:**

- Could you please articulate the connections between the proposed methods and the existing ones? In particular, both MASHA1 and MASHA2 seem similar to the FBF method with variance reduction in [2]. Also, the proof starting at line 786 in the Appendix also seems to be related to Lemma 2.4 in the mentioned paper.
- Minor issues:
    - The sentence from 39-42 is grammatically incorrect.
    - Line 53: "it often" -> "it is often"
    - Line 106: "second norm" -> "Euclidean norm"?
    - The fourth line from last in Algorithm 1: "$w^{k+1}=z^k$" -> "$w^{k+1}=z^{k+1}$"
    - Line 226: It should be $\mathcal{O}(1/\epsilon)$ instead of $\mathcal{O}(1/\epsilon^2)$?
    - The reference number in the appendix seems to be off by one.

**Limitations:**

Yes, the authors have mentioned the limitations in the paper.

**Strengths And Weaknesses:**

## Strengths

- This is the first work to study communication compression techniques in the setting of distributed VIs. Given the increasing importance of saddle point problems and the necessity of designing communication-efficient algorithms, I think this work should be of interest to the community.
- The presentation is mostly clear and easy to follow.
- The results are comprehensive and seem technically sound.

## Weaknesses

- I don't see any serious problem, but as a mostly theoretical paper I feel its algorithmic novelty is somehow limited. For instance, the idea of combining VR techniques and gradient compression seems not new; it has appeared in [1]. Also, the algorithm MASHA1 as well as its analysis resemble those in [2], with the stochastic oracle replaced by the compressed operator.

[1] Xun Qian, Peter Richtárik, and Tong Zhang. Error compensated distributed sgd can be accelerated. arXiv preprint arXiv:2010.00091, 2020.

[2] Ahmet Alacaoglu and Yura Malitsky. Stochastic variance reduction for variational inequality methods. arXiv preprint arXiv:2102.08352, 2021.

---

> ### Author Response · Authors · 2022-07-29
> **Response to Reviewer 1QcY**
>
> We thank Reviewer **1QcY** for the work! We are pleased that Reviewer appreciated our paper. We further provide answers to the questions and comments that Reviewer noted.
>
> > **Connections of MASHA and [1]** (here [1] is Alacaoglu and Malitsky's paper)
>
> As Reviewer correctly pointed out ( we point out it in line 207 (208 in the revision) and line 949 (995 in the revision) too), the idea of our method is related to the work [1]. But we borrow the momentum technique from there, everything else is either new facts or the use of already classical ones.
>
> 1) We have our own version of the proof. Please, compare for example Lemma 2.2 of [1] and our proof.
>
> 2) An interesting point: Reviewer noted that we have a typo in Algorithm 1: $w_{k+1}=z_{k}$. This is not a typo. Indeed in [1] the authors use $w_{k+1}=z_{k+1}$, but it is more convenient for us in our algorithm/proof to use $w_{k+1}=z_{k}$. This is most likely due to the fact that our and Alacaoglu and Malitsky's proofs follow several different ways.
>
> 3) Reviewer also pointed out Lemma 2.4 from [1]. It is quite classical, including in [1] the authors give a reference where it first appeared [2] (more than 10 years ago).
>
> 4) We give an analysis of Algorithm 1 and Algorithm 2 in the non-monotone case, which is not done in [1].
>
> 5) The analysis of Algorithm 2 is more complicated than the analysis of Algorithm 1 and the analysis of algorithms from [1]. This is due to the use of additional sequences for error feedback.
>
> > **Minor issues**
>
> Thanks! We have fixed all in the revision, but $w_{k+1}=z_k$ is not a typo (see above).
>
> [1] Ahmet Alacaoglu and Yura Malitsky. Stochastic variance reduction for variational inequality methods.
>
> [2] A. Nemirovski, A. Juditsky, G. Lan, and A. Shapiro. Robust stochastic approximation approach to stochastic programming.

---

> > ### Comment · Reviewer_1QcY · 2022-08-09
> > **After rebuttal**
> >
> > I thank the authors for the detailed responses. Now I see that MASHA and the variance-reduced FBF method in Alacaoglu and Malitsky's paper do differ in some ways and follow different lines of proof, though they share some similar high-level ideas. Below are a few further remarks:
> >
> > - Following the authors' response, in Line 220-221 of the revised paper $w^{k+1}=z^{k+1}$ should be $w^{k+1}=z^k$. Indeed, this is what led me to think that there is a typo in Algorithm 1 in the first place.
> > - I notice that the authors consider VIs in the unconstrained setting as shown in (3). On the other hand, in the example of adversarial training of Transformers we are dealing with a constrained min-max problem. Could the authors please comment on the possibility of extending the results to the constrained setting?

---

> > > ### Author Response · Authors · 2022-08-09
> > > **Response to Reviewer 1QcY (After rebuttal)**
> > >
> > > We thank Review **1QcY** for the response. See answers below.
> > >
> > > > **Following the authors' response, in Line 220-221 of the revised paper $w^{k+1} = z^{k+1}$ should be $w^{k+1} = z^k$. Indeed, this is what led me to think that there is a typo in Algorithm 1 in the first place.**
> > >
> > > Thanks very much! We fixed! See the new revision.
> > >
> > > > **I notice that the authors consider VIs in the unconstrained setting as shown in (3). On the other hand, in the example of adversarial training of Transformers we are dealing with a constrained min-max problem. Could the authors please comment on the possibility of extending the results to the constrained setting?**
> > >
> > > 1) For unbiased compressors, we say yes. Moreover, we checked all the proofs, and we can guarantee that the constrained setting can be done as well. For contractive compressors, the question is more tricky. We cannot guarantee it. Moreover, we have analyzed the literature on contractive compressors and on error compression techniques for minimization problems, and even for them the authors usually consider unconstrained setting [1,2,3,4,5,6,7]. Therefore, this is an interesting area for research not only for VIs/SPPs, but also for minimizations.
> > >
> > > 2) The problem from Section 5.2 (Transformer training) can be explored separately. This is due to the fact that the variables $\rho_n$ are unique for each data sample, they are stored locally and are not transmitted to the server (unlike the weights of the model $w$). It turns out that compression and error compensation are needed only for the minimization variable $w$. The variable $w$ is unconstrained, while the variables $\rho_n$ are constrained. It seems to us that for this particular problem, we can analyze the constrained setting.
> > >
> > >
> > > [1] Sebastian U Stich and Sai Praneeth Karimireddy. The error-feedback framework: Better rates for sgd with delayed gradients and compressed communication.
> > >
> > > [2] Xun Qian, Peter Richtárik, and Tong Zhang. Error compensated distributed sgd can be accelerated
> > >
> > > [3] Aleksandr Beznosikov, Samuel Horváth, Peter Richtárik, and Mher Safaryan. On biased compression for distributed learning.
> > >
> > > [4] Peter Richtárik, Igor Sokolov, and Ilyas Fatkhullin. EF21: A new, simpler, theoretically better, and practically faster error feedback.
> > >
> > > [5] Ilyas Fatkhullin, Igor Sokolov, Eduard Gorbunov, Zhize Li, and Peter Richtárik. Ef21 with bells & whistles: Practical algorithmic extensions of modern error feedback.
> > >
> > > [6] Hanlin Tang, Chen Yu, Xiangru Lian, Tong Zhang, and Ji Liu. Doublesqueeze: Parallel stochastic gradient descent with double-pass error-compensated compression.
> > >
> > > [7] Shuai Zheng, Ziyue Huang, and James Kwok. Communication-efficient distributed blockwise momentum sgd with error-feedback.

---

> > > > ### Comment · Reviewer_1QcY · 2022-08-09
> > > > **Thanks for your response**
> > > >
> > > > Thanks again for your thorough response. I think all my questions are resolved, and I decide to raise my score to 7.

---

> > > > > ### Author Response · Authors · 2022-08-09
> > > > > **Thank you!**
> > > > >
> > > > > We are grateful for raising the score!  Thanks again for the review, response, important comments, and positive final feedback!

---

### Official Review · Reviewer_CwZ6 · 2022-07-19

**Rating:** 6
**Confidence:** 3
**Soundness:** 2 fair
**Presentation:** 2 fair
**Contribution:** 3 good

**Summary:**

The paper proposes approaches to add compression to the communication steps involved in solving variational inequalities in a distributed fashion. The authors propose approaches for both unbiased and contractive compression along with theoretical analysis and two experiments to validate their approaches.

**Questions:**

1. Please address points 1 and 2 under Weaknesses above.

2. Please include a conclusion section in the paper.

3. The notation for local operators $F_m$ is used in Section 2.3 before it has been defined. Please define notation before using it.

**Strengths And Weaknesses:**

Strengths:

1. This appears to be the first work to address the issue of compressed communication in distributed methods for solving variaitional inequalities. Moreover it proposes and analyzes separate schemes for both unbiased and contractive compressors.

2. Results on the synthetic experiment of solving a bilinear saddle point problem clearly show that the proposed approaches require significantly fewer iterations than baselines for convergence.

Weaknesses:

1. The experiment on Adversarial Training of Transformers is not clear at all. Specifically it is unclear if any of the MASHA approaches have been implemented for this since the plots appear to contain only the baselines. Moreover  the size of the plots, and the font size of the text in them is extremely small and thus it is very hard to tell what is going on in them.

2. It is not clear if MASHA1 will always reduce the communication cost. Corollary 1 appears to show that MASHA1 will have fewer iterations than the uncompressed case, but each round of iteration can involve upto 2 rounds (each way) of communication between the server and the clients. As the authors state in line 235, the average communication cost per round is ($1/\beta + 1 - \tau$) times the communication cost in the uncompressed case. It is not clearly stated if $1/\beta + 1 - \tau$ is smaller or larger than 1. If it is larger, then the communication cost would be higher than that of the uncompressed case from what I can tell.

2. The paper appears to be incomplete with no conclusion sections and no discussion of limitations/future work.

---

> ### Author Response · Authors · 2022-07-29
> **Response to Reviewer CwZ6**
>
> We thank Reviewer **CwZ6** for the work! We are pleased that Reviewer appreciated our paper. We further provide answers to the questions and comments that Reviewer noted.
>
> > **The experiment on Adversarial Training of Transformers is not clear at all.** Specifically it is unclear if any of the MASHA approaches have been implemented for this since the plots appear to contain only the baselines. Moreover the size of the plots, and the font size of the text in them is extremely small and thus it is very hard to tell what is going on in them.
>
> 1) We have changed the size of figures and captions - see the revision.
>
> 2) The main goal of the second experiment is to show that for the large distributed VIs/SPPs, compression can play an important role.
>
> 3) As an optimizer, we use LAMB + MASHA.  From MASHA in this optimizer there is a negative momentum (in MASHA $z^{k+1/2} = z^k - (1 - \tau) ( z^k - w^k) - \gamma F(w^k)$, here $- (1 - \tau) ( z^k - w^k)$ is a negative momentum), as well as a compression and an error compensation technique. From LAMB (Adam type method) we add adaptivity/scaling to MASHA.
>
> 4) Reviewer is right that we don't exactly use MASHA. But it is a popular tendency in recent years in papers about VIs and SPPs, to make a theory for one method (without scaling/adaptivity), and then somehow modify Adam based on this theory. See papers [15, 26, 58, 13, 49] in our literature list.
>
> 5) We noted for ourselves that more recently (e.g., after the deadline for this  NeurIPS 2022 conference) there have been papers that attempt to analyze methods with scaling/adaptivity, like Adam, for SPPs and VIs. Then this would be an interesting direction for future research to connect MASHA and LAMB in theory. But it would be hard to get some additional facts into this paper.
>
> 6) Here we can only say that MASHA is good for monotone problems and really gives a win. And LAMB + MASHA are good for huge transformers.
>
> > **It is not clear if MASHA1 will always reduce the communication cost.** Corollary 1 appears to show that MASHA1 will have fewer iterations than the uncompressed case, but each round of iteration can involve upto 2 rounds (each way) of communication between the server and the clients. As the authors state in line 235, the average communication cost per round is  $(1/\beta+1−\tau)$ times the communication cost in the uncompressed case. It is not clearly stated if  $(1/\beta+1−\tau)$ is smaller or larger than 1. If it is larger, then the communication cost would be higher than that of the uncompressed case from what I can tell.
>
> 1) In line 235 (in the first version of the paper, in the revision it is line 238) we state that the average communication cost $(1/\beta+1−\tau)$ per iteration.
>
> 2) One can note that the Extra Gradient method (equality (6) after line 193 in the revision) has the communication cost $(1+1)$ per iteration, because we need to transfer the uncompressed operator $F$ twice per iteration. If we compare $(1/\beta+1−\tau)$ for MASHA1 and $(1+1)$ for Extra Gradient, we see that MASHA1 wins for any $\beta$ and $\tau$, since $\beta \geq 1$, $0\leq \tau \leq 1$ by definition.
>
> 3) If we want to compare compressed MASHA1 with uncompressed MASHA1 ($\beta = 1$), then one can note that uncompressed MASHA1 has the average communication cost $(1+1−\tau)$ per iteration. We can see that $(1/\beta+1−\tau) \leq  (1+1−\tau)$ for any $\beta$, because $\beta \geq 1$ by definition.
>
> 4) Reviewer asked to compare $(1/\beta+1−\tau)$ and $1$. In Corollary 4.1 we choose $1−\tau = 1/\beta$, then we need to compare $2/\beta$ and $1$. $2/\beta$ can sometimes be greater than $1$, but for practical compression operators usually $\beta >> 1$, and $2/\beta < 1$.
>
> > **Please include a conclusion section in the paper.**
>
> We have included, see the revision of our paper. For convenience, we give it here
>
> In this paper we present algorithms with unbiased and contractive compressions for solving distributed VIs and SPPs. Our algorithms are presented in deterministic, stochastic and federated versions. All basic algorithms and their modifications support bidirectional compression. Experiments confirm the efficiency of both our algorithms and the use of compression for solving large-scale VIs in general.
> In future works it is important to address the issue of the necessity to forward uncompressed information in some iterations. Although full packages are rarely transmitted, this is a slight limitation of our approach. Lower bounds for compression methods are also an interesting area of research. At the moment there are neither such results for VIs and SPPs, nor for minimizations. In Appendix C we only hypothesize the optimality of our methods and back it up with analogies, provable lower estimates could complete the story with compressed methods.
>
> > **The notation for local operators $F_m$ is used in Section 2.3 before it has been defined.** Please define notation before using it.
>
> Thanks, we have fixed it in the revision!

---

> > ### Comment · Reviewer_CwZ6 · 2022-08-08
> > **Re**
> >
> > Thank you for responding to my concerns. I am satisfied with the responses and have increased my score to 6. My only suggestions is that, if accepted, the paper should contain the clarifications you provided above regarding the adaptation of MASHA for adversarial training of Transformers since these points will make the scope and limitations of that experiment much clearer.

---

> > > ### Author Response · Authors · 2022-08-08
> > > **Thank you!**
> > >
> > > We greatly thank Reviewer **CwZ6** for the response, important comments, and positive final feedback!
> > >
> > > Of course we will include everything discussed with Reviewer in the final version, if we have extra space.

---

### Official Review · Reviewer_X46K · 2022-07-20

**Rating:** 6
**Confidence:** 3
**Soundness:** 3 good
**Presentation:** 3 good
**Contribution:** 3 good

**Summary:**

In this work the authors develop distributed communication-efficient methods for solving i)variational inequalities and ii)saddle point problems. The proposed algorithms (MASHA1 and MASHA2) utilize bidirectionally existing compression operators ( i) unbiased compression operator and ii) contractive compression operator) in order to reduce their communication cost in strongly monotone, monotone and non-monotone regimes. Convergence results are derived and presented for MASHA1 & 2 in all three regimes and numerical experiments showcase the merits suggested by the theory as well as the superior performance of the proposed methods compare to Extra Gradient. Stochastic variants (VR-MASHA1 and VR-MASHA2) with variance reduction are studied as well as connections to Federated Learning.

**Questions:**

The paper could certainly be improved substantially by more carefully revising the writing and the points mentioned in the 'Weaknesses section'.  Specifically the technical contributions and the description of algorithm 1 need to be more carefully rewritten.

Also as mentioned above the following parts need revising:
The sentence in lines 39-42 needs to be rephrased so that it has a clear meaning.
line 53 : In 'implementations it often advantageous' there is an 'is' missing.
line 56: Remove 'the' from 'lead to the comparable test error'.
line 60: Remove 'Since' from the beginning of the sentence.
line 79: Replace 'due to' with 'since'.
line 87: Remove ''provably
lines 130-131: Rephrase 'Moreover, devices can not only have a bad connection, in which one needs to compress data heavily, they can simply disconnect from the learning process'.
lines 140-143: Need rephrasing.
lines 209-221: The paragraph explaining MASHA1 needs to be rewritten. There are many typos and syntactic mistakes which make it hard to decipher the description of the algorithm.
line 228 : Replace 'listing' with 'description'.
line 230: 'an' is missing in 'Let us find optimal way to choose it'
line 247 : Rephrase 'Then once can note that MASHA1 O(􏰖1/M + 1/β · L/μ) better than the uncompressed extragradient method. We think that the factor O(􏰖1/M + 1/β · L/μ) is unimprovable and optimal – see Section A.'

**Limitations:**

The authors addressed the limitations of the paper.

**Strengths And Weaknesses:**

Strengths: This paper studies an interesting problem. Reducing the communication cost for distributed variational inequalities and saddle point problems is relevant in the area of machine learning. The paper provides convergence results for the proposed methods in many different regimes (strongly monotone, monotone and non-monotone operator regimes, deterministic, stochastic etc.). The theoretical results are also supported by illustrative experiments.

Weaknesses: In my perspective the paper has two main weaknesses.
1. Firstly, the writing of the paper requires a major revision. There are many typos, syntactic errors and sentences that do not make sense throughout the main body and the appendix. As a result many important points of the paper are not clearly conveyed such as the description of Algorithm 1. Further, the appendix needs also to be rewritten more carefully to make it easier to follow. Below I state a few examples that could be revised :

The sentence in lines 39-42 needs to be rephrased so that it has a clear meaning.
line 53 : In 'implementations it often advantageous' there is an 'is' missing.
line 56: Remove 'the' from 'lead to the comparable test error'.
line 60: Remove 'Since' from the beginning of the sentence.
line 79: Replace 'due to' with 'since'.
line 87: Remove ''provably
lines 130-131: Rephrase 'Moreover, devices can not only have a bad connection, in which one needs to compress data heavily, they can simply disconnect from the learning process'.
lines 140-143: Need rephrasing.
lines 209-221: The paragraph explaining MASHA1 needs to be rewritten. There are many typos and syntactic mistakes which make it hard to decipher the description of the algorithm.
line 228 : Replace 'listing' with 'description'.
line 230: 'an' is missing in 'Let us find optimal way to choose it'
line 247 : Rephrase 'Then once can note that MASHA1 O(􏰖1/M + 1/β · L/μ) better than the uncompressed extragradient method. We think that the factor O(􏰖1/M + 1/β · L/μ) is unimprovable and optimal – see Section A.'

2. While the theoretical results are extensive the originality and technical contribution appears to be limited. Similar results have been obtained with the same compression operators in optimization problems and certain techniques are repetitive.

---

> ### Author Response · Authors · 2022-07-29
> **Response to Reviewer X46K**
>
> We thank Reviewer **X46K** for the work! We are glad that Reviewer rated the contribution of our paper as “good”.
> We further provide answers to the questions and weaknesses that Reviewer noted. We hope we were able to solve the main problems.
>
> > **Firstly, the writing of the paper requires a major revision.** There are many typos, syntactic errors and sentences that do not make sense throughout the main body and the appendix. As a result many important points of the paper are not clearly conveyed such as the description of Algorithm 1. Further, the appendix needs also to be rewritten more carefully to make it easier to follow. Below I state a few examples that could be revised
>
> We are especially grateful to Reviewer for the work with the text of our paper. The comments and issues on the text have helped to improve our work. Please could Reviewer take a look at the revision of our article and see if Reviewer likes the changes? We have tried to solve all the troubles.
>
> > **While the theoretical results are extensive the originality and technical contribution appears to be limited.** Similar results have been obtained with the same compression operators in optimization problems and certain techniques are repetitive.
>
> 1) In Sections 4.1 and 4.2, we try to discuss that the new methods are not easy to obtain. In particular, known methods for minimization problems do not give the necessary result. Simple modifications of the Extra Gradient method do not give too. Our method is based on the negative momentum idea from [1] about non-distributed problems, but we give a different analysis, adding the non-monotone case (which is not in [1]). Moreover, Algorithm 2 is much more difficult to analyze than Algorithm 1 and algorithms from [1] because of the presence of the error compensation sequences.
>
> 2) Reviewer **1QcY** also noted this (one can see Reviewer **1QcY**‘s comment and our response to it in the corresponding review). But Reviewer **1QcY** gave 6.
>
> [1] Ahmet Alacaoglu and Yura Malitsky. Stochastic variance reduction for variational inequality methods.

---

> ### Comment · Reviewer_X46K · 2022-08-07
> **After rebuttal**
>
> I appreciate the efforts of the authors to address my concerns. I find that after revision the paper is written much more carefully and succinctly and the main results and ideas are conveyed with clarity (changes have been made throughout the whole paper and not only in the parts written in blue). Some important points are the description of Algorithm 1 as well as its comparison with uncompressed techniques which are now much better articulated. Further, the experiment section is now more carefully presented. In my opinion, the only weakness of the current version of the paper is that the novelty of their algorithms is somewhat limited. After carefully considering the reviewers' comments, the authors' responses and the latest version of this paper I have decided to improve my overall score to 6 "Weak Accept"(Soundness: 3 good, Presentation: 3 good, Contribution: 3 good).

---

> > ### Author Response · Authors · 2022-08-07
> > **Thank you!**
> >
> > We are very grateful for the raised score! We again want to say thanks for the very attentive reviewing of our paper, especially the work with the text!

---

### Official Review · Reviewer_9vze · 2022-07-20

**Rating:** 4
**Confidence:** 1
**Soundness:** 2 fair
**Presentation:** 2 fair
**Contribution:** 2 fair

**Summary:**

The authors propose communication-compressed methods for solving a particular class of problems that they refer to as "variational inequalities". The authors prevent theoretical guarantees for their method, claim that a naive compressed-gradient based approach will fail in general, and present experiments demonstrating benefit to their technique over a naive compressed-gradient based.

**Questions:**

I do not have any questions for the authors

**Limitations:**

Negative societal impact is not an issue here.

**Strengths And Weaknesses:**

### Major caveat to review
- I have not had time to go deeply through the mathematical content of this paper, therefore my review will not be able to address this aspect
- I apologise that my review thus addresses superficial aspects of the paper.

### Strengths
- the authors address a potentially practically useful problem, and produce both theoretical and experimental results on this topic.

### Weaknesses
- style violations. The authors greatly reduce the font-size of their figure captions, which as far as I am aware breaks the style rules, allowing them to fit more content into the 9 pages. This may sound a bit harsh, but I think that this is unfair to other authors who follow the rules and I would advocate for rejecting the paper for this reason.
- the paper has not been properly proof-read. Basic aspects of the technical introduction contain errors. For instance, the saddle point problem is introduced as min-min on line 158 when as far as I am aware it should be min-max. Also on line 221, the symbol q^serv is used twice instead of the proper notation.
- exposition. The exposition could be much clearer. For instance at line 154, no supporting intuition is offered about the mathematical form of the VI inequality, making it hard for an unfamiliar reader to build a working understanding of this topic.

### Overall
I want to apologise to the authors that my review is superficial. However, I am being clear about the fact that it is superficial. I want to add that I had a limited amount of time to review the work and ultimately needing to comment on things like style violations reduced the amount of time available to deal with the content.

---

> ### Author Response · Authors · 2022-07-29
> **Response to Reviewer 9vze**
>
> We thank Reviewer **9vze** for the work! We are very sorry that Reviewer couldn't spend much time on our paper, but Reviewer helped us make our paper better.
>
> Other Reviewers rate our contribution as “good” or “excellent”. We ask Reviewer not to judge our paper harshly because of typos and captions size. We have corrected all these problems in the revision. In more detail below.
>
> > **style violations.** The authors greatly reduce the font-size of their figure captions, which as far as I am aware breaks the style rules, allowing them to fit more content into the 9 pages. This may sound a bit harsh, but I think that this is unfair to other authors who follow the rules and I would advocate for rejecting the paper for this reason.
>
> Our paper revision is now 9 pages long, but we have returned the caption size to normal.
>
> > **the paper has not been properly proof-read.** Basic aspects of the technical introduction contain errors. For instance, the saddle point problem is introduced as min-min on line 158 when as far as I am aware it should be min-max. Also on line 221, the symbol q^serv is used twice instead of the proper notation.
>
> We have uploaded the revision of the paper in which the typos have been corrected. We have also tried to make an independent proof-read.
>
> > **exposition.** The exposition could be much clearer. For instance at line 154, no supporting intuition is offered about the mathematical form of the VI inequality, making it hard for an unfamiliar reader to build a working understanding of this topic.
>
> In the revision, we gave additional information about this. In line 157 there is a link to Appendix B. In this section we give examples of VIs formalism breadth. For the convenience, we include it here
>
> **Example 1 [Minimization]** Consider the minimization problem:
> \begin{align}
> \min_{z \in R^d} f(z).
> \end{align}
> Suppose that $F(z) = \nabla f(z)$. Then, if $f$ is convex, it can be proved that $z^* \in R^d$ is a solution for the VI problem if and only if $z^* \in R^d$ is a solution for the minimization problem. And if the function $f$ is non-convex, then $z^* \in R^d$ is a solution for the VI problem if and only if $\nabla f(z^*) = 0$, i.e. $z^*$ is a stationary point.
>
>
> **Example 2 [Saddle point problem]** Consider the saddle point problem:
> \begin{align}
> \min_{x \in R^{d_x}} \max_{y \in R^{d_y}} g(x,y).
> \end{align}
> Suppose that $F(z) = F(x,y) = [\nabla_x g(x,y), -\nabla_y g(x,y)]$ and $Z = R^{d_x} \times R^{d_y}$. Then, if $g$ is convex-concave, it can be proved that $z^* \in Z$ is a solution for the VI problem if and only if $z^* \in Z$ is a solution for the saddle point problem. And if the function $g$ is non-convex-non-concave, then $z^* \in Z$ is a solution for the VI problem if and only if $\nabla_x g(x^*, y^*) = 0$ and $\nabla_y g(x^*, y^*) = 0$, i.e. $z^*$ is a stationary point.
>
> If minimization problems are widely researched separately from variational inequalities. The study of saddle point problems often is associated with variational inequalities, therefore saddle point problems are strongly related to variational inequalities.
>
> **Example 3 [Fixed point problem]**
> Consider the fixed point problem:
> \begin{align}
>  \text{Find} ~~ z^* &\in R^d ~~ \text{such that} ~~
>     T(z^*) = z^*,
> \end{align}
> where $T: R^d \to R^d $ is an operator. With $F(z) = z - T(z)$, it can be proved that $z^* \in R^d$ is a solution for the VI problem if and only if $F(z^*) = 0$, i.e. $z^* \in R^d$ is a solution for the fixed point problem.
>
> Since this section takes up half of the page, we are ready to include it in the main part if we have extra space.

---

### Author Response · Authors · 2022-07-29
**Rebuttal Revision**

Dear Reviewers, Area Chairs and Senior Area Chairs!

Thank you very much for your work! You really helped make our paper better.

We have published a revision of our paper in which we have tried to solve most of the issues related to work. All changes are highlighted in blue. What is new:

1) Conclusion. Reviewer **CwZ6** asked to include a conclusion, where we, among other things, have described directions for future works.

2) Typos, rephrases etc. Reviewers **9vze**, **X46K**, **CwZ6**, **1QcY**, **SxRG** found typos or/and gave comments to improve the text. We have taken all comments into account in the revision.

3) Size of figures and captions. We have enlarged the figures and their captions at the request of Reviewers **9vze** and **CwZ6**.

4) Examples of VIs Formalism Breadth (Appendix B). Reviewer **9vze** asked for an additional explanation of the VI formalism for people who are not very familiar with this topic. We made a separate small section - Appendix B (a link to this section is given in line 157 of the revision). Since this section takes up half of the page, we are ready to include it in the main part if we have extra space.

---

### Author Response · Authors · 2022-08-07
**A kind reminder about rebuttals**

With this message, we would just like to kindly remind Reviewers that we would be happy if Reviewers would participate in the rebuttal discussion process. We are looking forward to hearing from Reviewers **9vze**, **CwZ6**, **1QcY** and **SxRG**. We thank Reviewer **X46K** for the responses to the rebuttal.

---

### Meta-Review · Area_Chair_VxwJ · 2022-08-27

**Recommendation:** Accept
**Confidence:** Certain

**Metareview:**

Dear Authors,

We had a long discussion about this paper. Overall, the reviews are positive. Several reviewers raised their scores after the rebuttal phase, and they found the response by the authors satisfactory.

However, there were some concerns about the novelty of the paper that I summarize here:

This paper combines some standard techniques and ideas from decentralized optimization and minimax optimization to obtain the presented results. Hence, the algorithmic novelty of the paper is limited. Perhaps the major contribution of the paper is in the vector that they decide to quantize, but still, the main idea of the paper is very similar to the single-loop variance reduction techniques that were first proposed in stochastic optimization and later used for distributed optimization. The main theoretical challenge that the authors had to face was combining quantization with the Extra Gradient method as highlighted in the first paragraphs of section 4.1. Indeed, similar quantization ideas have been extensively studied in the distributed optimization literature and thus the algorithmic novelty seems to be very limited. Similarly, excluding the aforementioned challenge (Extra Gradient + compression) the derivation of the theoretical results appears to be tedious but based on standard techniques.

Considering the above points, the AC and one of the reviewers found the paper below the bar as its novelty is limited. However, four reviewers voted in favor of accepting this paper, as they believe the technical novelty of the paper and its proof techniques are significant enough.

I respect the majority vote and hence recommend this paper to be accepted.

**Award:**

No

---

### Decision · Program_Chairs · 2022-09-14

Accept